# Variational Regularized Unbalanced Optimal Transport: Single Network, Least Action

**Yuhao Sun**[1,*]**, Zhenyi Zhang**[2,*]**, Zihan Wang**[3,*]**, Tiejun Li**[1,2,4,†] **and Peijie Zhou**[1,3,4,5,†]

[1]Center for Machine Learning Research, Peking University.
[2]LMAM and School of Mathematical Sciences, Peking University.
[3]Center for Quantitative Biology, Peking University.
[4]NELBDA, Peking University. [5]AI for Science Institute, Beijing.
Emails: {2501111524,zhenyizhang,jackwzh}@stu.pku.edu.cn,
{tieli,pjzhou}@pku.edu.cn

## Abstract

Recovering the dynamics from a few snapshots of a high-dimensional system is a challenging task in statistical physics and machine learning, with important applications in computational biology. Many algorithms have been developed to tackle this problem, based on frameworks such as optimal transport and the Schrödinger bridge. A notable recent framework is Regularized Unbalanced Optimal Transport (RUOT), which integrates both stochastic dynamics and unnormalized distributions. However, since many existing methods do not explicitly enforce optimality conditions, their solutions often struggle to satisfy the principle of least action and meet challenges to converge in a stable and reliable way. To address these issues, we propose Variational RUOT (Var-RUOT), a new framework to solve the RUOT problem. By incorporating the optimal necessary conditions for the RUOT problem into both the parameterization of the search space and the loss function design, Var-RUOT only needs to learn a scalar field to solve the RUOT problem and can search for solutions with lower action. We also examined the challenge of selecting a growth penalty function in the widely used Wasserstein-Fisher-Rao metric and proposed a solution that better aligns with biological priors in Var-RUOT. We validated the effectiveness of Var-RUOT on both simulated data and real single-cell datasets. Compared with existing algorithms, Var-RUOT can find solutions with lower action while exhibiting faster convergence and improved training stability. Our code is available at `https://github.com/ZerooVector/VarRUOT`.

## 1 Introduction

Inferring continuous dynamics from finite observations is crucial when analyzing systems with many particles (Chen et al. 2018). However, in many important applications such as single-cell RNA sequencing (scRNA-seq) experiments, only a few snapshot measurements are available, which makes recovering the underlying continuous dynamics a challenging task (Ding et al. 2022). Such a task of reconstructing dynamics from sparse snapshots is commonly referred to as *trajectory inference* in time-series scRNA-seq modeling (Zhang et al. 2025b; Ding et al. 2022; Heitz et al. 2024; Yeo et al. 2021a; Schiebinger et al. 2019a; Bunne et al. 2023b; Zhang et al. 2021) or the mathematical problem of *ensemble regression* (Yang et al. 2022).

A number of frameworks have been proposed to address this problem. For example, in dynamical optimal transport (OT), particles evolve according to the ordinary differential equations (ODEs) with

---

*These authors contributed equally. [†]Corresponding authors.

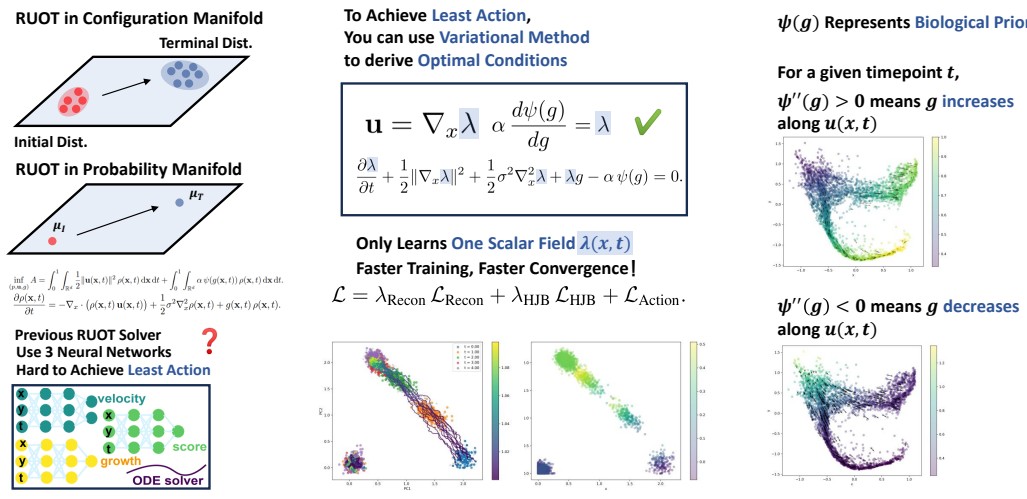

Figure 1: Overview of Variational RUOT (Var-RUOT).

the objective of minimizing the total action required to transport the initial distribution to the terminal distribution (Benamou and Brenier 2000). Unbalanced dynamical OT further extends this framework by adding a penalty term $\psi(g)$ on the particle growth or death processes in total transport energy (namely the Wasserstein–Fisher–Rao metric or WFR metric) to handle unnormalized distributions (Chizat et al. 2018a; Chizat et al. 2018b). Moreover, stochastic methods such as the Schrödinger Bridge adopt similar action principles while governing particle evolution via stochastic differential equations (SDEs) (Gentil et al. 2017; Léonard 2014). Recently, the Regularized Unbalanced Optimal Transport (RUOT) framework generalizes these ideas by incorporating both stochasticity and particle birth-death processes (Lavenant et al. 2024; Ventre et al. 2023; Chizat et al. 2022; Pariset et al. 2023; Zhang et al. 2025a). In machine learning, generative models such as diffusion models (Ho et al. 2020; Song et al. 2021; Sohl-Dickstein et al. 2015; Song et al. 2020) and flow matching techniques (Lipman et al. 2023; Tong et al. 2024a; Liu et al. 2022) have also been adapted to solve transport problems. However, these approaches face two major challenges: 1) They usually do not explicitly enforce optimality conditions, leading to solutions that violate the principle of least action, and they meet challenges to converge reliably; 2) Selecting an appropriate penalty function $\psi(g)$ that aligns with underlying biological priors remains challenging.

To overcome these challenges, we propose **Variational-RUOT (Var-RUOT)**. Our algorithm employs variational methods to derive the necessary conditions for action minimization within the RUOT framework. By parameterizing a single scalar field with a neural network and incorporating these optimality conditions directly into our loss design, Var-RUOT learns dynamics with lower action. Experiments on both simulated and real datasets demonstrate that our approach achieves competitive performance with fewer training epochs and improved stability. Furthermore, we show that different choices of the penalty function for the growth rate $g$ yield distinct biologically relevant priors in single-cell dynamics modeling. Our contributions are summarized as follows:

- We introduce a new method for solving RUOT problems by incorporating the first-order optimality conditions directly into the solution parameterization. This reduces the learning task to a single scalar potential function, which significantly simplifies the model space.

- We show how incorporating these necessary conditions into the loss function and architecture enables Var-RUOT to consistently discover transport paths with lower action, providing a more efficient and stable training process for RUOT problem.

- We address a key limitation in the classical Wasserstein-Fisher-Rao metric, which can yield biologically implausible solutions due to its quadratic growth penalty term. We propose the criterion and practical solution to modify such a penalty term, therefore enhancing the more realistic modeling of single-cell dynamics.

## 2 Related Works

**Deep Learning Solver for Trajectory Inference Problem**    There are a large number of deep learning-based trajectory inference problem solvers. For example, there are solvers for Optimal Transport using static OT solver, Neural ODE or Flow matching techniques (Tong et al. 2020; Huguet et al. 2022; Wan et al. 2023; Zhang et al. 2024a; Tong et al. 2024a; Albergo et al. 2023; Palma et al. 2025; Rohbeck et al. 2025; Petrović et al. 2025; Schiebinger et al. 2019b; Klein et al. 2025; Wang et al. 2025), as well as solvers for the Schrödinger bridge that utilize either its relation to entropy regularized OT or its optimal control formulation(Shi et al. 2024; De Bortoli et al. 2021; Gu et al. 2025; Koshizuka and Sato 2023; Neklyudov et al. 2024; Zhang et al. 2024b; Bunne et al. 2023a; Chen et al. 2022a; Zhou et al. 2024; Zhu et al. 2024; Maddu et al. 2024; Yeo et al. 2021b; Jiang and Wan 2024; Lavenant et al. 2024; Ventre et al. 2023; Chizat et al. 2022; Tong et al. 2024b; Atanackovic et al. 2025; Yang 2025; You et al. 2024). However, these methods typically **employ separate neural networks** to parameterize the velocity and growth functions, without leveraging their **optimality conditions** or the **inherent relationship** between them(Zhang et al. 2025d) with few exceptions such as Action Matching (Neklyudov et al. 2023; Neklyudov et al. 2024)) . This poses challenges in achieving optimal solutions that minimize the action energy. Our work generalizes Action Matching (Neklyudov et al. 2023; Neklyudov et al. 2024) to handle the setting where unbalanced distributions and stochastic effects coexist.

**HJB equations in optimal transport**    Methods that leverage the optimality conditions (e.g., Hamilton-Jacobi-Bellman (HJB) equations) of dynamic OT and its variants, have been proposed (Neklyudov et al. 2024; Zhang et al. 2024b; Chen et al. 2016; Benamou and Brenier 2000; Neklyudov et al. 2023; Wu et al. 2025; Chow et al. 2020). However, these approaches typically do not simultaneously address both **unbalanced and stochastic** dynamics.

**WFR metric in time-series scRNA-seq modeling**    In computational biology, several existing works model both cell state transitions and growth dynamics simultaneously in temporal scRNA-seq datasets by minimizing the action in the WFR metric i.e.solving the dynamical unbalanced optimal transport problem (Sha et al. 2024; Tong et al. 2023; Peng et al. 2024; Eyring et al. 2024b) or its variants (Pariset et al. 2023; Lavenant et al. 2024; Zhang et al. 2025a; Zhang et al. 2025c). However, these works usually adopt the default growth penalty function $\psi(g) = \frac{1}{2}g^2$ in the WFR metric and have not investigated the **biological implications of different choices** for $\psi(g)$.

## 3 Preliminaries and Backgrounds

**Dynamical Optimal Transport**    The Dynamical Optimal Transport, also known as the Benamou–Brenier formulation, requires minimizing the following action functional (Benamou and Brenier 2000):

$$\inf_{\rho,\mathbf{u}} \int_0^1 \int_{\mathbb{R}^d} \frac{1}{2}\|\mathbf{u}(\mathbf{x},t)\|^2 \, \rho(\mathbf{x},t) \, \mathrm{d}\mathbf{x} \, \mathrm{d}t,$$

where $\rho$ and $\mathbf{u}$ are subject to the continuity equation constraint:

$$\frac{\partial \rho(\mathbf{x},t)}{\partial t} + \nabla_{\mathbf{x}} \cdot \Big(\mathbf{u}(\mathbf{x},t)\rho(\mathbf{x},t)\Big) = 0, \quad \rho(\cdot,0) = \mu_0, \quad \rho(\cdot,1) = \mu_1.$$

**Unbalanced Dynamical OT and Wasserstein–Fisher–Rao (WFR) metric**    In order to handle unnormalized probability densities in practical problems (for example, to account for cell proliferation and death in computational biology), one can modify the form of the continuity equation by adding a birth-death term, and accordingly include a corresponding penalty term in the action. This leads to the optimal transport problem under the Wasserstein–Fisher–Rao (WFR) metric (Chizat et al. 2018a; Chizat et al. 2018b).

$$\inf_{\rho,\mathbf{u},g} \int_0^1 \int_{\mathbb{R}^d} \left(\frac{1}{2}\|\mathbf{u}(\mathbf{x},t)\|^2 + \alpha \, g^2(\mathbf{x},t)\right) \rho(\mathbf{x},t) \, d\mathbf{x} \, \mathrm{d}t,$$

with $\rho$, $\mathbf{u}$, and $g$ subject to the unnormalized continuity equation constraint

$$\frac{\partial \rho(\mathbf{x},t)}{\partial t} + \nabla_{\mathbf{x}} \cdot \Big(\mathbf{u}(\mathbf{x},t)\rho(\mathbf{x},t)\Big) - g(\mathbf{x},t)\rho(\mathbf{x},t) = 0, \quad \rho(\cdot,0) = \mu_0, \quad \rho(\cdot,1) = \mu_1.$$

**Schrödinger Bridge Problem and Dynamical Formulation** Schrödinger bridges aims to find the most likely way for a stochastic system to evolve from an initial distribution $\mu_0$ to a terminal distribution $\mu_1$. Formally, let $\mu_{[0,1]}^{\mathbf{X}}$ denote the probability measure induced by the stochastic process $\mathbf{X}(t)$, $0 \le t \le 1$, and let $\mu_{[0,1]}^{\mathbf{Y}}$ denote the probability measure induced by a given reference process $\mathbf{Y}(t)$, $0 \le t \le 1$. The Schrödinger bridge seeks to solve $\min_{\mu_{[0,1]}^{\mathbf{X}}} \mathcal{D}_{\mathrm{KL}}\left( \mu_{[0,1]}^{\mathbf{X}} \,\big\|\, \mu_{[0,1]}^{\mathbf{Y}} \right)$. In particular, if $\mathbf{X}_t$ follows the SDE $\mathrm{d}\mathbf{X}_t = \mathbf{u}(\mathbf{X}_t, t)\,\mathrm{d}t + \boldsymbol{\sigma}(\mathbf{X}_t, t)\,\mathrm{d}\mathbf{W}_t$, where $\mathbf{W}_t \in \mathbb{R}^d$ is a standard Brownian motion, $\boldsymbol{\sigma}(\mathbf{x}, t) \in \mathbb{R}^{d \times d}$ is a given diffusion matrix, and the reference process is defined as $\mathrm{d}\mathbf{Y}_t = \boldsymbol{\sigma}(\mathbf{Y}_t, t)\,\mathrm{d}\mathbf{W}_t$, then the Schrödinger bridge problem is equivalent to the following stochastic optimal control problem (Chen et al. 2016; Gentil et al. 2017):

$$\inf_{\rho, \mathbf{u}} \int_0^1 \int_{\mathbb{R}^d} \left( \frac{1}{2}\, \mathbf{u}^T(\mathbf{x}, t)\, \mathbf{a}^{-1}(\mathbf{x}, t)\, \mathbf{u}(\mathbf{x}, t) \right) \rho(\mathbf{x}, t)\, \mathrm{d}\mathbf{x}\, \mathrm{d}t,$$

where $\rho$ and $\mathbf{u}$ are subject to the Fokker–Planck equation constraint

$$\frac{\partial \rho(\mathbf{x}, t)}{\partial t} + \nabla_{\mathbf{x}} \cdot \Big( \mathbf{u}(\mathbf{x}, t)\, \rho(\mathbf{x}, t) \Big) - \frac{1}{2}\nabla_{\mathbf{x}}^2 : \Big( \mathbf{a}(\mathbf{x}, t)\, \rho(\mathbf{x}, t) \Big) = 0, \quad \rho(\cdot, 0) = \mu_0, \quad \rho(\cdot, 1) = \mu_1.$$

Here, $\mathbf{a}(\mathbf{x}, t) = \boldsymbol{\sigma}(\mathbf{x}, t)\boldsymbol{\sigma}^T(\mathbf{x}, t)$ and $\nabla_{\mathbf{x}}^2 : \Big( \mathbf{a}(\mathbf{x}, t)\, \rho(\mathbf{x}, t) \Big) = \sum_{ij} \partial_{ij}\big( a_{ij}\, \rho(\mathbf{x}, t) \big)$.

**Regularized Unbalanced Optimal Transport** If we consider both unnormalized probability densities and stochasticity simultaneously, we arrive at the Regularized Unbalanced Optimal Transport (RUOT) problem (Chen et al. 2022b; Baradat and Lavenant 2021; Zhang et al. 2025a).

**Definition 3.1** (Regularized Unbalanced Optimal Transport (RUOT) Problem). *Consider minimizing the following action:*

$$\inf_{\rho, \mathbf{u}, g} \int_0^1 \int_{\mathbb{R}^d} \left( \frac{1}{2}\, \mathbf{u}^T(\mathbf{x}, t)\, \mathbf{a}^{-1}(\mathbf{x}, t)\, \mathbf{u}(\mathbf{x}, t) + \alpha\, \psi(g) \right) \rho(\mathbf{x}, t)\, \mathrm{d}\mathbf{x}\, \mathrm{d}t,$$

*where $\psi : \mathbb{R} \to [0, +\infty)$ is a growth penalty function, $\alpha$ is a penalty coefficient, and the quantities $\rho$, $\mathbf{u}$ and $g$ are subject to the following constraint, which is an unnormalized continuity equation:*

$$\frac{\partial \rho}{\partial t} + \nabla_{\mathbf{x}} \cdot \Big( \mathbf{u}(\mathbf{x}, t)\, \rho \Big) - \frac{1}{2}\nabla_{\mathbf{x}}^2 : \Big( \mathbf{a}(\mathbf{x}, t)\, \rho \Big) - g(\mathbf{x}, t)\, \rho = 0, \quad \rho(\cdot, 0) = \mu_0, \quad \rho(\cdot, 1) = \mu_1.$$

**Remark 3.1.** *When modeling the particle dynamics in these OT variants, one can use either autonomous systems (where $v$ or $g$ does not explicitly depend on $t$) or non-autonomous systems (where $v$ or $g$ explicitly depends on $t$). Here, we follow methods such as DeepRUOT(Zhang et al. 2025a) to choose non-autonomous systems. Indeed non-autonomous systems offer greater expressive power from a learning perspective and are less prone to underfitting the data.*

## 4 Necessary Optimality Conditions for RUOT

To simplify our problem we adopt the assumption of isotropic time-invariant diffusion, i.e., $\mathbf{a}(\mathbf{x}, t) = \sigma^2 \mathbf{I}$. We refer to RUOT problem in this scenario as the *isotropic time-invariant RUOT problem*.

**Definition 4.1** (Isotropic Time-Invariant (ITI) RUOT Problem). *Consider the following minimum-action problem with the action functional given by*

$$\inf_{(\rho, \mathbf{u}, g)} \mathscr{T} = \int_0^1 \int_{\mathbb{R}^d} \frac{1}{2}\|\mathbf{u}(\mathbf{x}, t)\|^2\, \rho(\mathbf{x}, t)\, \mathrm{d}\mathbf{x}\, \mathrm{d}t + \int_0^1 \int_{\mathbb{R}^d} \alpha\, \psi(g(\mathbf{x}, t))\, \rho(\mathbf{x}, t)\, \mathrm{d}\mathbf{x}\, \mathrm{d}t. \qquad (1)$$

*Here, $\psi : \mathbb{R} \to [0, +\infty)$ is the growth penalty function, and the triplet $(\rho, \mathbf{u}, g)$ is subject to the constraint of the Fokker–Planck equation*

$$\frac{\partial \rho(\mathbf{x}, t)}{\partial t} = -\nabla_{\mathbf{x}} \cdot \big( \rho(\mathbf{x}, t)\, \mathbf{u}(\mathbf{x}, t) \big) + \frac{1}{2}\sigma^2 \nabla_{\mathbf{x}}^2 \rho(\mathbf{x}, t) + g(\mathbf{x}, t)\, \rho(\mathbf{x}, t). \qquad (2)$$

*Additionally, $p$ satisfies the initial and terminal conditions $\rho(\cdot, 0) = \mu_0, \quad \rho(\cdot, 1) = \mu_1$.*

In particular, if $\psi(g(\mathbf{x}, t)) = \frac{1}{2}g^2(\mathbf{x}, t)$, then this problem is referred to as the *unbalanced dynamic optimal transport with WFR metric*. We can derive the necessary conditions for the action functional to achieve a minimum using variational methods.

**Theorem 4.1** (Necessary Conditions for Achieving the Optimal Solution in the ITI-RUOT Problem)**.** *In the problem defined in* Definition 4.1, *the necessary conditions for the action $\mathscr{T}$ to attain a minimum are*

$$\mathbf{u} = \nabla_{\mathbf{x}}\lambda, \quad \alpha \frac{\mathrm{d}\psi(g)}{\mathrm{d}g} = \lambda, \quad \frac{\partial\lambda}{\partial t} + \frac{1}{2}\|\nabla_{\mathbf{x}}\lambda\|^2 + \frac{1}{2}\sigma^2\nabla_{\mathbf{x}}^2\lambda + \lambda g - \alpha\,\psi(g) = 0. \tag{3}$$

Here, $\lambda(\mathbf{x}, t)$ is a scalar field. The proof of this theorem can be found in Appendix A.1.1.

**Remark 4.1.** *Substituting the necessary conditions satisfied by $\mathbf{u}$ and $g$ into the Fokker–Planck equation, the evolution of the probability density $\rho(x, t)$ is determined by $\frac{\partial\rho(\mathbf{x},t)}{\partial t} = -\nabla_{\mathbf{x}} \cdot \left(\rho(\mathbf{x}, t)\nabla_{\mathbf{x}}\lambda(\mathbf{x}, t)\right) + \frac{1}{2}\sigma^2\nabla_{\mathbf{x}}^2\rho(\mathbf{x}, t) + (\psi')^{-1}\left(\frac{\lambda(\mathbf{x},t)}{\alpha}\right)\rho(\mathbf{x}, t)$, where $\psi' = \frac{\mathrm{d}\psi(g)}{\mathrm{d}g}$, and $(\psi')^{-1}$ denotes the inverse function of $\psi'$.*

**Remark 4.2.** *If we choose the growth penalty function to take the form used in the WFR metric, i.e., $\psi(g) = \frac{1}{2}g^2$, and set $\alpha = 1$, $\sigma = 0$, then the above optimal necessary conditions immediately degenerate to $\mathbf{u} = \nabla_{\mathbf{x}}\lambda$, $g = \lambda$, $\frac{\partial\lambda}{\partial t} + \frac{1}{2}\|\nabla_{\mathbf{x}}\lambda\|^2 + \frac{1}{2}\lambda^2 = 0$, which is same as the form derived in (Neklyudov et al. 2024) under the WFR metric. If we let $g = 0$ and $\psi(0) = 0$ it becomes*
$$\mathbf{u} = \nabla_{\mathbf{x}}\lambda, \quad \frac{\partial\lambda}{\partial t} + \frac{1}{2}\|\nabla_{\mathbf{x}}\lambda\|^2 + \frac{1}{2}\sigma^2\nabla_{\mathbf{x}}^2\lambda = 0,$$ *which is same as the form derived in (Neklyudov et al. 2024; Zhang et al. 2024b; Chen et al. 2016) under the Schrödinger Bridge problem.*

From Theorem 4.1 and Remark 4.1, the vector field $\mathbf{u}(\mathbf{x}, t)$ and the growth rate $g(\mathbf{x}, t)$ can be directly obtained from the scalar field $\lambda(\mathbf{x}, t)$. Moreover, since the initial density $\rho(\cdot, 0)$ is known, once the necessary conditions are satisfied, the evolution equation (i.e., the Fokker–Planck equation) is completely determined by $\lambda(\mathbf{x}, t)$. Thus, the scalar field $\lambda(\mathbf{x}, t)$ fully determines the system's evolution, and we only need to solve for one $\lambda(\mathbf{x}, t)$, which simplifies the problem.

However, these necessary conditions introduce a coupling between $\mathbf{u}(\mathbf{x}, t)$ and $g(\mathbf{x}, t)$, and this coupling could contradict biological prior knowledge. In biological data, it is generally believed that cells located at the upstream of a trajectory are stem cells with the highest proliferation and differentiation capabilities, and thus the corresponding $g$ values should be maximal. Along the trajectory, as the cells gradually lose their "stemness," the $g$ values would decrease. Under the necessary conditions, however, whether $g(\mathbf{x}, t)$ increases or decreases along $\mathbf{u}(x, t)$ at a given time $t$ depends on the form of the growth penalty function.

**Theorem 4.2** (The relationship between $\mathbf{u}$ and $g$ ; Biological prior)**.** *At a fixed time $t$, if $\frac{\mathrm{d}\psi^2(g)}{\mathrm{d}g^2} > 0$, then $g(\mathbf{x}, t)$ ascends in the direction of the velocity field $\mathbf{u}(\mathbf{x}, t)$ (i.e., $\mathbf{u}(\mathbf{x}, t)^T(\nabla_{\mathbf{x}}g(\mathbf{x}, t)) > 0$); otherwise, it descends.*

The proof is given in Appendix A.1.2. According to this theorem, to ensure the solution complies with biological prior, i.e., that at a given time the cells upstream in the trajectory exhibit the higher $g$ value, it is necessary to ensure that that $\frac{\mathrm{d}^2\psi(g)}{\mathrm{d}g^2} < 0$.

## 5 Solving ITI ROUT Problem Through Neural Network

Given samples from distributions $\rho_t$ at $K$ discrete time points, $t \in \{T_1, \cdots, T_K\}$, we aim to recover the continuous evolution process of the distributions by solving the ITI RUOT problem, that is, by minimizing the action functional while ensuring that $\rho(\mathbf{x}, t)$ matches the distributions $\rho_t$ at the corresponding time points. Since the values of $\mathbf{u}(\mathbf{x}, t)$ and $g(\mathbf{x}, t)$, as well as the evolution of $\rho(\mathbf{x}, t)$ over time, are fully determined by the scalar field $\lambda(\mathbf{x}, t)$ in variational form (Appendix A.1.1), we approximate this scalar field using a single neural network. Specifically, we parameterize $\lambda(\mathbf{x}, t)$ as $\lambda_\theta(\mathbf{x}, t)$, where $\theta$ represents the neural network parameters.

### 5.1 Simulating SDEs Using the Weighted Particle Method

Directly solving the high-dimensional RUOT with PDE constraints is challenging. Therefore, we reformulate the problem by simulating the trajectories of a number of weighted particles.

**Theorem 5.1.** *Consider a weighted particle system consisting of $N$ particles, where the position of particle $i$ at time $t$ is given by $\mathbf{X}_i^t \in \mathbb{R}^d$ and its weight by $w_i(t) > 0$. The dynamics of each particle are described by*

$$
\mathrm{d}\mathbf{X}_i^t = \mathbf{u}(\mathbf{X}_i^t, t)\,\mathrm{d}t + \sigma(t)\,\mathrm{d}\mathbf{W}_t,
$$

$$
\mathrm{d}w_i = g(\mathbf{X}_i^t, t)\,w_i\,\mathrm{d}t,
$$

(4)

*where $\mathbf{u} : \mathbb{R}^d \times [0, T] \to \mathbb{R}^d$ is a time-varying vector field, $g : \mathbb{R}^d \times [0, T] \to \mathbb{R}$ is a growth rate function, $\sigma : [0, T] \to [0, +\infty)$ is a time-varying diffusion coefficient, and $\mathbf{W}_t$ is an $N$-dimensional standard Brownian motion with independent components in each coordinate. The initial conditions are $\mathbf{X}_i^0 \sim \rho(\mathbf{x}, 0)$ and $w_i(0) = 1$. In the limit as $N \to \infty$, the empirical measure $\mu^N(\mathbf{x}, t) = \frac{1}{N}\sum_{i=1}^N w_i(t)\,\delta(\mathbf{x} - \mathbf{X}_i^t)$ converges to the solution of the following Fokker–Planck equation:*

$$
\frac{\partial \rho(\mathbf{x}, t)}{\partial t} = -\nabla_{\mathbf{x}} \cdot \Big( \mathbf{u}(\mathbf{x}, t)\,\rho(\mathbf{x}, t) \Big) + \frac{1}{2}\sigma^2(t)\nabla_{\mathbf{x}}^2 \rho(\mathbf{x}, t) + g(\mathbf{x}, t)\,\rho(\mathbf{x}, t),
$$

(5)

*with the initial condition $\rho(\mathbf{x}, 0) = \rho_0(\mathbf{x})$.*

The proof is provided in Appendix A.1.3. This theorem implies that we can approximate the evolution of $\rho(\mathbf{x}, t)$ by simulating $N$ particles, where each particle's weight $w_i$ is governed by an ODE and its position $\mathbf{X}_i$ is governed by an SDE. The evolution of the empirical measure $\mu^N(\mathbf{x}, t)$ thereby approximates the evolution of $\rho(\mathbf{x}, t)$.

## 5.2 Reformulating the Loss in Weighted Particle Form

The total loss function consists of three components such that $\mathcal{L} = \mathcal{L}_{\text{Recon}} + \gamma_{\text{HJB}}\,\mathcal{L}_{\text{HJB}} + \gamma_{\text{Action}}\mathcal{L}_{\text{Action}}$. Here, $\mathcal{L}_{\text{Recon}}$ ensures that the distribution generated by the model closely matches the true data distribution, $\mathcal{L}_{\text{HJB}}$ enforces that the learned $\lambda_\theta(\mathbf{x}, t)$ satisfies the HJB equation in the necessary conditions, and $\mathcal{L}_{\text{Action}}$ minimizes the action as much as possible.

**Reconstruction Loss**  Minimizing the reconstruction loss guarantees that the distribution generated by the model is consistent with the real data distribution. Since in the ITI RUOT problem the probability density $\rho(\mathbf{x}, t)$ is not normalized, we need to match both the total mass and the discrepancy between the two distributions. Our reconstruction loss is given by $\mathcal{L}_{\text{Recon}} = \gamma_{\text{Mass}}\,\mathcal{L}_{\text{Mass}} + \mathcal{L}_{\text{OT}}$, where, at time point $k$, the true mass is $\int_{\mathbb{R}^d} \rho(\mathbf{x}, T_k)\,\mathrm{d}\mathbf{x} = M(T_k)$, and the weight of particle $i$ is $w_i(T_k)$. Thus, the total mass of the model-generated distribution is $\hat{M}(T_k) = \frac{1}{N}\sum_{i=1}^N w_i(T_k)$. The mass reconstruction loss is then defined as $\mathcal{L}_{\text{Mass}} = \sum_{k=1}^K \Big( M(T_k) - \hat{M}(T_k) \Big)^2$. Let the true distribution at time point $k$ be $\rho(\mathbf{x}, T_k)$. Its normalized version is given by $\tilde{\rho}(\mathbf{x}, T_k) = \frac{\rho(\mathbf{x}, T_k)}{\int_{\mathbb{R}^d} \rho(\mathbf{x}, T_k)\,\mathrm{d}\mathbf{x}}$, while the normalized model-generated distribution is $\hat{\tilde{\rho}}(\mathbf{x}, T_k) = \frac{\frac{1}{N}\sum_{i=1}^N w_i(T_k)\,\delta(\mathbf{x} - \mathbf{X}_i)}{\hat{M}(T_k)}$. The distribution reconstruction loss is then defined as $\mathcal{L}_{\text{OT}} = \sum_{k=1}^K \mathcal{W}_2\Big( \tilde{\rho}(\cdot, T_k),\ \hat{\tilde{\rho}}(\cdot, T_k) \Big)$, where $\gamma_{\text{Mass}}$ is a hyperparameter that controls the importance of the mass reconstruction loss.

**HJB Loss**  Minimizing the HJB loss ensures that the learned $\lambda_\theta(\mathbf{x}, t)$ obeys the HJB equation constraints specified in the necessary conditions. Since the gradient operator in the HJB equation is a local operator, we compute the HJB loss by integrating the extent to which $\lambda_\theta(\mathbf{x}, t)$ violates the HJB equation along the trajectories. When using $N$ particles, the HJB loss is given by: $\mathcal{L}_{\text{HJB}}^N = \sum_{i=1}^N \left[ \int_0^{T_K} \frac{w_i(t)}{\sum_{i=1}^N w_i(t)} \left( \frac{\partial \lambda_\theta(\mathbf{X}_i^t, t)}{\partial t} + \frac{1}{2}\|\nabla_{\mathbf{x}}\lambda_\theta\|^2 + \frac{1}{2}\sigma^2 \nabla_{\mathbf{x}}^2 \lambda_\theta + g_\theta(\mathbf{X}_i^t, t) - \alpha\,\psi(g_\theta) \right)^2 \mathrm{d}t \right]$. Here $g_\theta$ is obtained from the necessary condition $\alpha\,\frac{\mathrm{d}\psi(g_\theta)}{\mathrm{d}g_\theta} = \lambda_\theta$.

**Remark 5.1.** *The expectation of HJB Loss is $\mathbb{E}[\mathcal{L}_{\text{HJB}}^N] = \int_0^{T_K} \int_{\mathbb{R}^d} \hat{\rho}(\mathbf{x}, t) \left( \frac{\partial \lambda(\mathbf{x}, t)}{\partial t} + \frac{1}{2}\|\nabla_{\mathbf{x}}\lambda\|^2 + \frac{1}{2}\sigma^2 \nabla_{\mathbf{x}}^2 \lambda + g(\mathbf{x}, t) - \alpha\,\psi(g) \right)^2 \mathrm{d}\mathbf{x}\mathrm{d}t$ where $\hat{\rho}(\mathbf{x}, t) = \frac{\rho(\mathbf{x}, t)}{\int_{\mathbb{R}^d} \rho(\mathbf{x}, t)\mathrm{d}\mathbf{x}}$ is the probability density by normalizing $\rho(\mathbf{x}, t)$. The proof is left in Appendix A.1.4.*

**Action Loss**  Since the variational method provides only the necessary conditions for achieving minimal action rather than sufficient ones, we need to incorporate the action into the loss so that the action is minimized as much as possible.  The action loss is also computed via simulating weighted particles.  When using $N$ particles, it is given by $\mathcal{L}_{\text{Action}}^N =$
$$\frac{1}{N}\sum_{i=1}^{N}\left(\int_0^1 \frac{1}{2}\|\mathbf{u}_\theta(\mathbf{X}_i^t,t)\|^2 w_i(t)\mathrm{d}t + \int_0^1 \alpha\psi(g_\theta(\mathbf{X}_i^t,t))w_i(t)\mathrm{d}t\right).$$ Here $\mathbf{u}$ and $g$ is obtained from the necessary condition $\mathbf{u}_\theta = \nabla_{\mathbf{x}}\lambda_\theta(\mathbf{x},t), \alpha\frac{\mathrm{d}\psi(g_\theta)}{\mathrm{d}g_\theta} = \lambda_\theta$.

**Remark 5.2.** *The expectation of action loss is exactly the action defined in the ITI RUOT problem (Definition 4.1): $\mathbb{E}[\mathcal{L}_{Action}^N] = \mathscr{T}$. The proof is left in Appendix A.1.5.*

Overall, the training process of Var-RUOT involves minimizing the total of three loss terms described above to fit $\lambda_\theta$. The training procedure is provided in Algorithm 1.

### 5.3  Adjusting the Growth Penalty Function to Match Biological Priors

As discussed in Theorem 4.2, the second-order derivative of $\psi(g)$ encodes the biological prior: if $\psi''(g) > 0$, then at any given time $t$, $g$ increases in the direction of the velocity field, and vice versa. Therefore, we choose two representative forms of $\psi(g)$ for our solution. Given that $\psi(g)$ penalizes nonzero $g$, it should satisfy the following properties: (1) The further $g$ deviates from $0$, the larger $\psi(g)$ becomes, i.e., $\frac{\mathrm{d}\psi(g)}{\mathrm{d}|g|} > 0$ . (2) Birth and death are penalized equally when prior knowledge is absent, i.e., $\psi(g) = \psi(-g)$.

**Case 1: $\psi''(g) > 0$**  In the case where $\psi''(g) > 0$, a typical form that meets the requirements is $\psi(g) = Cg^{2p}, \quad p \in \mathbb{Z}^+, \quad C > 0$. We select the form used in the WFR Metric, namely, $\psi_1(g) = \frac{1}{2}g^2$. The optimality conditions are presented in Appendix A.2.

**Case 2: $\psi''(g) < 0$**  For the case where $\psi''(g) < 0$, a typical form that meets the conditions is $\psi(g) = C\,g^{(2p)/(2q+1)}$ , where $p, q \in \mathbb{Z}^+$ and $2p < 2q + 1$. In order to obtain a smoother $g(\lambda)$ relationship from the necessary conditions, and *as a illustrative example*, we choose $\psi_2(g) = g^{2/15}$. The optimality conditions are also presented in Appendix A.2.

## 6  Numerical Results

In the experiments presented below, unless the use of the modified metric is explicitly stated, we utilize the standard WFR metric, namely, $\psi_1(g) = \frac{1}{2}g^2$.

### 6.1  Var-RUOT Minimizes Path Action

To evaluate the ability of Var-RUOT to capture the minimum-action trajectory, we first conducted experiments on a three-gene simulation dataset (Zhang et al. 2025a). The dynamics of the three-gene simulation data are governed by stochastic differential equations that incorporate self-activation, mutual inhibition, and external activation. The detailed specifications of the dataset are provided in Appendix B.1. The trajectories learned by DeepRUOT and Var-RUOT are illustrated in Fig. 2, and the $\mathcal{W}_1$ and $\mathcal{W}_2$ losses between the generated distributions and the ground truth distribution, as well as the corresponding action values, are reported in Table 1. In the table, we report the action of the method that utilizes the WFR Metric. The experimental results demonstrate that Var-RUOT achieves a lower path action while maintaining distribution matching accuracy. To further assess the performance of Var-RUOT on high-dimensional data, we also conducted experiments on an Epithelial Mesenchymal Transition (EMT) dataset(Sha et al. 2024; Cook and Vanderhyden 2020). This dataset was reduced to a 10-dimensional feature space, and the trajectories obtained after applying PCA for dimensionality reduction are shown in Fig. 3. Both Var-RUOT and DeepRUOT learn dynamics that can transform the distribution at $t = 0$ into the distributions at $t = 1, 2, 3$. Var-RUOT learns the nearly straight-line trajectory corresponding to the minimum action, whereas DeepRUOT learns a curved trajectory. The results of $\mathcal{W}_1, \mathcal{W}_2$ distance and action, summarized in Table 2, Var-RUOT also learns trajectories with smaller action while achieving matching accuracy comparable to that of other algorithms.

Table 1: On the three-gene simulated dataset, the Wasserstein distances ($\mathcal{W}_1$ and $\mathcal{W}_2$) between the predicted distributions of each algorithm and the true distribution at various time points. Each experiment was run five times to compute the mean and standard deviation. ($\mathcal{W}_1$ and $\mathcal{W}_2$ are relative metrics and should therefore only be compared on the same dataset.)

| Model | $t=1$ $\mathcal{W}_1$ | $\mathcal{W}_2$ | $t=2$ $\mathcal{W}_1$ | $\mathcal{W}_2$ | $t=3$ $\mathcal{W}_1$ | $\mathcal{W}_2$ | $t=4$ $\mathcal{W}_1$ | $\mathcal{W}_2$ | Path Action |
|---|---|---|---|---|---|---|---|---|---|
| SF2M (Tong et al. 2024b) | $0.1914_{\pm0.0051}$ | $0.3253_{\pm0.0059}$ | $0.4706_{\pm0.0200}$ | $0.7648_{\pm0.0059}$ | $0.7648_{\pm0.0260}$ | $1.0750_{\pm0.0267}$ | $2.1879_{\pm0.0451}$ | $2.8830_{\pm0.0741}$ | – |
| PISDE (Jiang and Wan 2024) | $0.1313_{\pm0.0023}$ | $0.3232_{\pm0.0013}$ | $0.2311_{\pm0.0015}$ | $0.5356_{\pm0.0015}$ | $0.4103_{\pm0.0006}$ | $0.7913_{\pm0.0035}$ | $0.5418_{\pm0.0015}$ | $0.9579_{\pm0.0037}$ | – |
| MIO Flow (Huguet et al. 2022) | $0.1290_{\pm0.0000}$ | $0.2087_{\pm0.0000}$ | $0.2963_{\pm0.0000}$ | $0.4565_{\pm0.0000}$ | $0.6461_{\pm0.0000}$ | $1.0165_{\pm0.0000}$ | $1.1473_{\pm0.0000}$ | $1.7827_{\pm0.0000}$ | – |
| Action Matching (Neklyudov et al. 2023) | $0.3801_{\pm0.0000}$ | $0.5033_{\pm0.0000}$ | $0.5028_{\pm0.0000}$ | $0.5637_{\pm0.0000}$ | $0.6288_{\pm0.0000}$ | $0.6822_{\pm0.0000}$ | $0.8480_{\pm0.0000}$ | $0.9034_{\pm0.0000}$ | 1.5491 |
| OTCFM (Tong et al. 2024a) | $0.1035_{\pm0.0000}$ | $0.3043_{\pm0.0000}$ | $0.2078_{\pm0.0000}$ | $0.4923_{\pm0.0000}$ | $0.2898_{\pm0.0000}$ | $0.5867_{\pm0.0000}$ | $0.3107_{\pm0.0000}$ | $0.4358_{\pm0.0000}$ | – |
| UOTCFM (Eyring et al. 2024a) | $0.1002_{\pm0.0000}$ | $0.2911_{\pm0.0000}$ | $0.1653_{\pm0.0000}$ | $0.3578_{\pm0.0000}$ | $0.1711_{\pm0.0000}$ | $0.2620_{\pm0.0000}$ | $0.4129_{\pm0.0000}$ | $0.7583_{\pm0.0000}$ | – |
| WLF (Neklyudov et al. 2024) | $0.4983_{\pm0.0000}$ | $0.5273_{\pm0.0000}$ | $0.8346_{\pm0.0000}$ | $0.8357_{\pm0.0000}$ | $0.8046_{\pm0.0000}$ | $0.8815_{\pm0.0000}$ | $0.4493_{\pm0.0000}$ | $0.8571_{\pm0.0000}$ | – |
| TIGON (Sha et al. 2024) | $0.0519_{\pm0.0000}$ | $\mathbf{0.0731}_{\pm0.0000}$ | $0.0763_{\pm0.0000}$ | $0.1559_{\pm0.0000}$ | $0.1387_{\pm0.0000}$ | $0.2436_{\pm0.0000}$ | $0.1908_{\pm0.0000}$ | $0.2203_{\pm0.0000}$ | 1.2442 |
| DeepRUOT (Zhang et al. 2025a) | $0.0569_{\pm0.0019}$ | $0.1125_{\pm0.0033}$ | $0.0811_{\pm0.0037}$ | $0.1578_{\pm0.0079}$ | $0.1246_{\pm0.0040}$ | $0.2158_{\pm0.0081}$ | $0.1538_{\pm0.0056}$ | $0.2588_{\pm0.0088}$ | 1.4058 |
| Var-RUOT (Ours) | $\mathbf{0.0452}_{\pm0.0024}$ | $0.1181_{\pm0.0064}$ | $\mathbf{0.0385}_{\pm0.0022}$ | $\mathbf{0.1270}_{\pm0.0121}$ | $\mathbf{0.0445}_{\pm0.0033}$ | $\mathbf{0.1144}_{\pm0.0160}$ | $\mathbf{0.0572}_{\pm0.0034}$ | $\mathbf{0.2140}_{\pm0.0067}$ | $\mathbf{1.1105}_{\pm0.0515}$ |

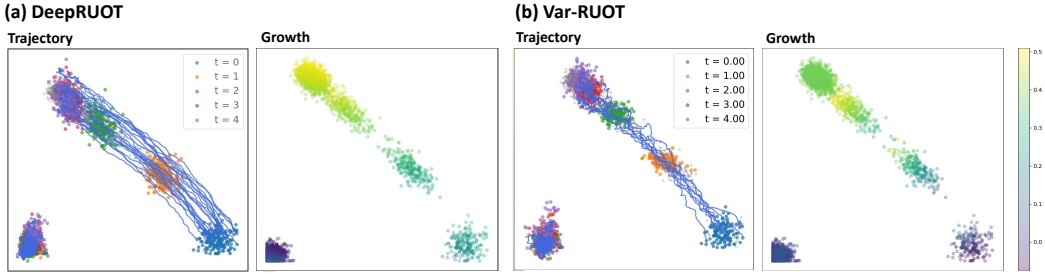

**(a) DeepRUOT**  Trajectory  Growth  **(b) Var-RUOT**  Trajectory  Growth

Figure 2: **Comparison between DeepRUOT and Var-RUOT on dynamics reconstruction on three-gene simulation dataset.** a) The trajectory and growth rate learned by DeepRUOT. b) The trajectory and growth rate learned by Var-RUOT on the same dataset. In the figure, the circles and '+' signs denote the generated data and the original data points, respectively.

Table 2: On the EMT dataset, the Wasserstein distances ($\mathcal{W}_1$ and $\mathcal{W}_2$) between the predicted distributions of each algorithm and the true distribution at various time points. Each experiment was run five times to compute the mean and standard deviation.

| Model | $t=1$ $\mathcal{W}_1$ | $\mathcal{W}_2$ | $t=2$ $\mathcal{W}_1$ | $\mathcal{W}_2$ | $t=3$ $\mathcal{W}_1$ | $\mathcal{W}_2$ | Path Action |
|---|---|---|---|---|---|---|---|
| SF2M (Tong et al. 2024b) | $0.2566_{\pm0.0016}$ | $0.2646_{\pm0.0016}$ | $0.2811_{\pm0.0016}$ | $0.2897_{\pm0.0012}$ | $0.2900_{\pm0.0010}$ | $0.3005_{\pm0.0010}$ | – |
| PISDE (Jiang and Wan 2024) | $0.2694_{\pm0.0016}$ | $0.2785_{\pm0.0016}$ | $0.2860_{\pm0.0013}$ | $0.2954_{\pm0.0012}$ | $0.2790_{\pm0.0015}$ | $0.2920_{\pm0.0016}$ | – |
| MIO Flow (Huguet et al. 2022) | $\mathbf{0.2439}_{\pm0.0000}$ | $\mathbf{0.2529}_{\pm0.0000}$ | $\mathbf{0.2665}_{\pm0.0000}$ | $0.2770_{\pm0.0000}$ | $0.2841_{\pm0.0000}$ | $0.2984_{\pm0.0000}$ | – |
| Action Matching (Neklyudov et al. 2023) | $0.4723_{\pm0.0000}$ | $0.4794_{\pm0.0000}$ | $0.6382_{\pm0.0000}$ | $0.6454_{\pm0.0000}$ | $0.8453_{\pm0.0000}$ | $0.8524_{\pm0.0000}$ | 0.8583 |
| OTCFM (Tong et al. 2024a) | $0.2557_{\pm0.0000}$ | $0.2649_{\pm0.0000}$ | $0.2701_{\pm0.0000}$ | $0.2799_{\pm0.0000}$ | $0.2799_{\pm0.0000}$ | $0.2914_{\pm0.0000}$ | – |
| UOTCFM (Eyring et al. 2024a) | $0.2538_{\pm0.0000}$ | $0.2629_{\pm0.0000}$ | $0.2696_{\pm0.0000}$ | $0.2797_{\pm0.0000}$ | $0.2771_{\pm0.0000}$ | $0.2912_{\pm0.0000}$ | – |
| WLF (Neklyudov et al. 2024) | $0.3901_{\pm0.0000}$ | $0.3977_{\pm0.0000}$ | $0.4381_{\pm0.0000}$ | $0.4466_{\pm0.0000}$ | $0.2848_{\pm0.0000}$ | $0.2976_{\pm0.0000}$ | – |
| TIGON (Sha et al. 2024) | $0.2433_{\pm0.0000}$ | $0.2523_{\pm0.0000}$ | $0.2661_{\pm0.0000}$ | $0.2766_{\pm0.0000}$ | $0.2847_{\pm0.0000}$ | $0.2989_{\pm0.0000}$ | 0.4672 |
| DeepRUOT (Zhang et al. 2025a) | $0.2902_{\pm0.0009}$ | $0.2987_{\pm0.0012}$ | $0.3193_{\pm0.0006}$ | $0.3293_{\pm0.0008}$ | $0.3291_{\pm0.00018}$ | $0.3410_{\pm0.0023}$ | 0.4857 |
| Var-RUOT (Ours) | $0.2540_{\pm0.0016}$ | $0.2623_{\pm0.0017}$ | $0.2670_{\pm0.0013}$ | $\mathbf{0.2756}_{\pm0.0014}$ | $\mathbf{0.2683}_{\pm0.0014}$ | $\mathbf{0.2796}_{\pm0.0015}$ | $\mathbf{0.3544}_{\pm0.0019}$ |

## 6.2 Var-RUOT Stabilizes and Accelerates Training Process

To demonstrate that Var-RUOT converges faster and exhibits improved training stability, we further tested it on both the simulated and the EMT dataset. We trained the neural networks for the various algorithms using the same learning rate and optimizer, running each dataset five times. For each training, we recorded the number of epochs and wall-clock time required for the OT loss related to the distribution matching accuracy to decrease below a specified threshold (set to 0.30 in this study). Each training session was capped at a maximum of 500 epochs. If an algorithm's OT loss did not reach the threshold within 500 epochs, the required epoch was recorded as 500, and the wall-clock time was noted as the total duration of the training session. The experimental results are summarized in Table 3, which lists the mean and standard deviation of both the epochs and wall-clock times required for each algorithm on each dataset. The mean values reflect the convergence speed, while the standard deviations indicate the training stability. Our algorithm demonstrated both a faster convergence speed and better stability compared to the other methods. In Appendix C.1, we further demonstrate our training speed and stability by plotting the loss decay curves.

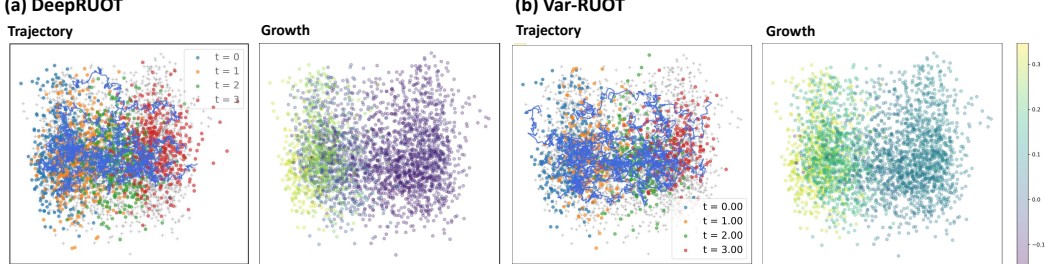

Figure 3: **Comparison between DeepRUOT and Var-RUOT on dynamics reconstruction on EMT dataset.** a) The trajectory and growth rate learned by DeepRUOT. b) The trajectory and growth rate learned by Var-RUOT on the same dataset. In the figure, the circles and '+' signs denote the generated data and the original data points, respectively.

Table 3: The number of epochs and wall time required for the OT Loss to drop below the threshold for each algorithm. We trained each algorithm five times to compute the mean and standard deviation.

| | Simulation Gene | | EMT | |
|---|---|---|---|---|
| Model | Epoch | Wall Time(Sec.) | Epoch | Wall Time(Sec.) |
| TIGON (Sha et al. 2024) | $228.40_{\pm223.71}$ | $1142.79_{\pm1345.21}$ | $110.40_{\pm193.37}$ | $365.54_{\pm639.86}$ |
| RUOT w/o Pretraining (Zhang et al. 2025a) | $172.00_{\pm229.11}$ | $578.67_{\pm768.52}$ | $228.20_{\pm223.88}$ | $819.31_{\pm804.05}$ |
| RUOT with 3 Epoch Pretraining (Zhang et al. 2025a) | $204.40_{\pm238.29}$ | $653.33_{\pm761.35}$ | $221.60_{\pm226.52}$ | $801.18_{\pm819.46}$ |
| Var-RUOT (Ours) | $\mathbf{27.60}_{\pm5.75}$ | $\mathbf{33.98}_{\pm6.37}$ | $\mathbf{5.20}_{\pm1.26}$ | $\mathbf{7.37}_{\pm1.89}$ |

## 6.3 Different $\psi(g)$ Represents Different Biological Prior

To illustrate that the choice of $\psi(g)$ represents different biological priors, we present the learned dynamics under two selections of $\psi(g)$. We apply our algorithm on the Mouse Blood Hematopoiesis dataset (Weinreb et al. 2020; Sha et al. 2024). In Fig. 4(a), the standard WFR metric is applied, i.e., $\psi_1(g) = \frac{1}{2}g^2$, from which it can be observed that at time points $t = 0, 1, 2$, along the direction of the drift vector field $\mathbf{u}(\mathbf{x}, t)$, $g(\mathbf{x}, t)$ gradually increases. In Fig. 4(b), on the other hand, the alternative selection $\psi_2(g) = g^{2/15}$ mentioned in Section 5.3 is used, and it is evident that at each time point, $g(\mathbf{x}, t)$ gradually decreases along the direction of $\mathbf{u}(\mathbf{x}, t)$. The distribution matching accuracy and the action are reported in Table 4. When employing the modified metric, the corresponding action quantity is not directly comparable to those obtained using the WFR metric. Therefore we do not report its action here.

Table 4: Wasserstein distances ($\mathcal{W}_1$ and $\mathcal{W}_2$) between the predicted distributions of each algorithm and the true distribution at various time points on mouse blood hematopoiesis. Each experiment was run five times to compute the mean and standard deviation.

| | $t = 1$ | | $t = 2$ | | Path Action |
|---|---|---|---|---|---|
| Model | $\mathcal{W}_1$ | $\mathcal{W}_2$ | $\mathcal{W}_1$ | $\mathcal{W}_2$ | |
| Action Matching (Neklyudov et al. 2023) | $0.4719_{\pm0.0000}$ | $0.5673_{\pm0.0000}$ | $0.8350_{\pm0.0000}$ | $0.8936_{\pm0.0000}$ | 4.3517 |
| MIOFLOW (Huguet et al. 2022) | $0.3546_{\pm0.0000}$ | $0.4083_{\pm0.0000}$ | $0.2772_{\pm0.0000}$ | $0.3459_{\pm0.0000}$ | - |
| SF2M (Tong et al. 2024b) | $0.1706_{\pm0.0043}$ | $0.2150_{\pm0.0062}$ | $0.1602_{\pm0.0029}$ | $0.2013_{\pm0.0039}$ | - |
| PISDE (Jiang and Wan 2024) | $0.3124_{\pm0.0065}$ | $0.3499_{\pm0.0062}$ | $0.2531_{\pm0.0142}$ | $0.2983_{\pm0.0175}$ | - |
| OTCFM (Tong et al. 2024a) | $0.3674_{\pm0.0000}$ | $0.4814_{\pm0.0000}$ | $0.3222_{\pm0.0000}$ | $0.3737_{\pm0.0000}$ | - |
| UOTCFM (Eyring et al. 2024a) | $0.3301_{\pm0.0000}$ | $0.3950_{\pm0.0000}$ | $0.2051_{\pm0.0000}$ | $0.2606_{\pm0.0000}$ | - |
| WLF (Neklyudov et al. 2024) | $0.3302_{\pm0.0000}$ | $0.3950_{\pm0.0000}$ | $0.2051_{\pm0.0000}$ | $0.2606_{\pm0.0000}$ | - |
| TIGON (Sha et al. 2024) | $0.4498_{\pm0.0000}$ | $0.5139_{\pm0.0000}$ | $0.4368_{\pm0.0000}$ | $0.4852_{\pm0.0000}$ | 3.7438 |
| DeepRUOT (Zhang et al. 2025a) | $0.1456_{\pm0.0016}$ | $0.1807_{\pm0.0019}$ | $0.1469_{\pm0.0046}$ | $0.1791_{\pm0.0061}$ | 5.5887 |
| Var-RUOT (Standard WFR) | $\mathbf{0.1200}_{\pm0.0038}$ | $\mathbf{0.1459}_{\pm0.0038}$ | $\mathbf{0.1431}_{\pm0.0092}$ | $\mathbf{0.1764}_{\pm0.0135}$ | $\mathbf{3.1491}_{\pm0.0837}$ |
| Var-RUOT (Modified Metric) | $0.2953_{\pm0.0357}$ | $0.3117_{\pm0.0323}$ | $0.1917_{\pm0.0140}$ | $0.2226_{\pm0.0170}$ | - |

In addition to the three experiments presented in the main text, we conducted several ablation studies and supplementary experiments to further validate our method. First, we performed ablation studies on the weights of the HJB and action losses to verify their effectiveness in learning dynamics with a minimal action. Additionally, we explored alternative modeling choices, including a comparison

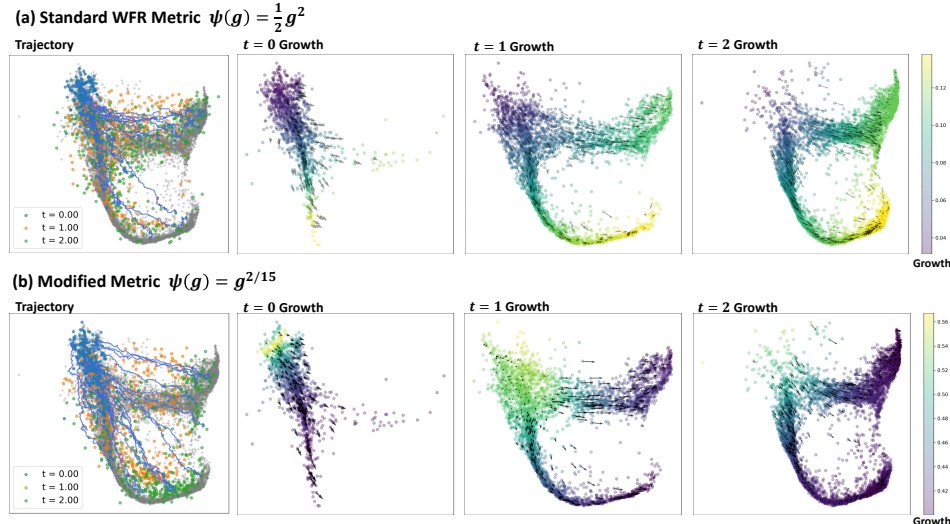

Figure 4: **Comparison of Var-RUOT using different growth metric on mouse blood hematopoiesis dataset.** a) The trajectory and growth at time points $t = 0, 1, 2$ learned using the standard WFR metric. b) The trajectory and growth at time points $t = 0, 1, 2$ learned using the modified metric. In the figure, the circles and '+' signs denote the generated data and the original data points, respectively.

between using $L_1$ and $L_2$ norms for the loss terms, and an analysis of the impact of different growth penalty functions beyond the $\psi_2(g) = g^{\frac{2}{15}}$ example where $\psi''(g) < 0$. The detailed results of these studies are provided in Appendix C.2. To assess the model's generalization capabilities, we performed hold-one-out and long-term extrapolation experiments. The results indicate that the Var-RUOT algorithm can effectively perform both interpolation and extrapolation, with the learned minimum-action dynamics leading to more accurate extrapolation outcomes (Appendix C.3). Furthermore, we carried out experiments on several high-dimensional datasets, which further validate the effectiveness of Var-RUOT in high-dimensional settings (Appendix C.4). Finally, we compared the path action computed by Var-RUOT against that from a traditional OT solver(Appendix C.5). The experimental results show that the path actions obtained by both methods are similar, which helps to validate our approach.

## 7 Conclusion

In this paper, we propose a new algorithm for solving the RUOT problem called Variational RUOT. By employing variational methods to derive the necessary conditions for the minimum action solution of RUOT, we solve the problem by learning a single scalar field. Compared to other algorithms, Var-RUOT can find solutions with lower action values while achieving the same level of fitting performance, and it offers faster training and convergence speeds. Finally, we emphasize that the selection of $\psi(g)$ in the action is crucial and directly linked to biological priors.

Although the Var-RUOT algorithm presented in this paper offers new insights for solving the RUOT problem, it is subject to several key limitations. Firstly, because the algorithm is based on neural network optimization, it can only converge to a local minimum rather than guaranteeing the attainment of the global minimum of the action. In practice, the HJB loss functions more as a regularization term than as a hard constraint that converges to zero. In addition, when using the modified metric, the goodness-of-fit for the distribution deteriorates, which warrants further investigation. Finally, the choice of the function $\psi(g)$ within the action is dependent on biological priors and is not automated. Future work could address these limitations by systematically investigating more canonical problems, which would provide clearer insight into the algorithm's fundamental behavior and limitations, and by automating the selection of $\psi(g)$ either by approximating it with a neural network or by deriving it from first-principle-based microscopic dynamics, such as a branching Wiener (Baradat and Lavenant 2021). We discussed more limitations of our work and potential broader impacts in Appendix E.1.

## Acknowledgments and Disclosure of Funding

This work was supported by the National Key R&D Program of China (No. 2021YFA1003301 to T.L.), National Natural Science Foundation of China (NSFC No. 12288101 to T.L. & P.Z., and 8206100646, T2321001 to P.Z.) and The Fundamental Research Funds for the Central Universities, Peking University. We acknowledge the support from the High-performance Computing Platform of Peking University for computation.

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

# A Technical Details

## A.1 Proof of Theorems

### A.1.1 Proof for Theorem 4.1

**Theorem A.1.** *The RUOT problem with isotropic and time-invariant diffusion intensity is formulated as:*

$$\min_{\rho,g} A = \int_0^1 \int_{\mathbb{R}^d} \left( \frac{1}{2} \|\mathbf{u}(\mathbf{x},t)\|^2 + \alpha\,\psi(g(\mathbf{x},t)) \right) \rho(\mathbf{x},t)\,\mathrm{d}\mathbf{x}\,\mathrm{d}t \tag{6}$$

$$s.t. \quad \frac{\partial \rho(\mathbf{x},t)}{\partial t} = -\nabla \cdot \left( \rho(\mathbf{x},t)\,\mathbf{u}(\mathbf{x},t) \right) + \frac{1}{2}\sigma^2 \nabla^2 \rho(\mathbf{x},t) + g(\mathbf{x},t)\rho(\mathbf{x},t) \tag{7}$$

*In this problem, the necessary conditions for the action $A$ to achieve a minimum are given by:*

$$\mathbf{u} = \nabla_{\mathbf{x}}\lambda, \quad \alpha\,\frac{\mathrm{d}\psi(g)}{\mathrm{d}g} = \lambda, \quad \frac{\partial \lambda}{\partial t} + \frac{1}{2}\|\nabla_{\mathbf{x}}\lambda\|^2 + \frac{1}{2}\sigma^2 \nabla_{\mathbf{x}}^2 \lambda + \lambda g - \alpha\,\psi(g) = 0, \tag{8}$$

*where $\lambda(\mathbf{x},t)$ is a scalar field.*

*Proof.* In order to incorporate the constraints of the Fokker–Planck equation, we construct an augmented action functional:

$$A^{\dagger} = \int_0^1 \int_{\mathbb{R}^d} \left( \frac{1}{2} \|\mathbf{u}(\mathbf{x},t)\|^2 + \alpha\,\psi(g(\mathbf{x},t)) \right) \rho(\mathbf{x},t)\,\mathrm{d}\mathbf{x}\,\mathrm{d}t$$
$$+ \int_0^1 \int_{\mathbb{R}^d} \lambda(\mathbf{x},t) \left( \frac{\partial \rho(\mathbf{x},t)}{\partial t} + \nabla \cdot \left( \rho(\mathbf{x},t)\,\mathbf{u}(\mathbf{x},t) \right) - \frac{1}{2}\sigma^2 \nabla^2 \rho(\mathbf{x},t) - g(\mathbf{x},t)\rho(\mathbf{x},t) \right) \mathrm{d}\mathbf{x}\mathrm{d}t$$

We take variations with respect to $\mathbf{u}$, $g$, and $\rho$. At the stationary point of the functional, the variation of the augmented action functional must vanish.

**Step 1: Variation with respect to $\mathbf{u}$.**
Let $\mathbf{u} \to \mathbf{u} + \delta\mathbf{u}$. The variation of the augmented action is

$$\delta A^{\dagger} = \int_0^1 \int_{\mathbb{R}^d} (\mathbf{u}^T \delta\mathbf{u})\rho + \lambda \nabla_{\mathbf{x}} \cdot (\rho\delta\mathbf{u})\mathrm{d}\mathbf{x}\mathrm{d}t$$
$$= \int_0^1 \int_{\mathbb{R}^d} (\mathbf{u}^T \delta\mathbf{u})\rho + \int_0^1 \int_{\mathbb{R}^d} (\nabla_{\mathbf{x}} \cdot (\lambda\rho\delta\mathbf{u}) - \rho(\nabla_{\mathbf{x}}\lambda)^T \delta\mathbf{u})\mathrm{d}\mathbf{x}\mathrm{d}t$$
$$= \int_0^1 \int_{\mathbb{R}^d} (\mathbf{u}^T \delta\mathbf{u})\rho + \cancel{\int_0^1 \int_{S^{\infty}} \lambda\rho(\delta\mathbf{u})^T \mathrm{d}\mathbf{S}} - \int_0^1 \int_{\mathbb{R}^d} \rho(\nabla_{\mathbf{x}}\lambda)^T \delta\mathbf{u}\mathrm{d}\mathbf{x}\mathrm{d}t$$
$$= \int_0^1 \int_{\mathbb{R}^d} (\mathbf{u}^T - (\nabla_{\mathbf{x}}\lambda)^T)\rho\delta\mathbf{u}\mathrm{d}\mathbf{x}\mathrm{d}t$$

Here, $S^{\infty}$ denotes the boundary at infinity in $\mathbb{R}^d$ and $\mathrm{d}\mathbf{S}$ is the surface element. Based on the assumption that

$$\int_{S^{\infty}} \lambda\rho\,(\delta\mathbf{u})^T\,\mathrm{d}\mathbf{S} = 0,$$

and using the arbitrariness of $\delta\mathbf{u}$, we obtain the optimality condition

$$\mathbf{u} = \nabla_{\mathbf{x}}\lambda.$$

**Step 2: Variation with respect to $g$.**
Let $g \to g + \delta g$, then the variation of the augmented action becomes

$$\delta A^{\dagger} = \int_0^1 \int_{\mathbb{R}^d} \left( \alpha\,\frac{\mathrm{d}\psi(g)}{\mathrm{d}g} - \lambda \right) \rho\,\delta g\,\mathrm{d}\mathbf{x}\,\mathrm{d}t.$$

Since $\delta g$ is arbitrary, we immediately obtain the optimality condition

$$\alpha \frac{\mathrm{d}\psi(g)}{\mathrm{d}g} = \lambda.$$

**Step 3: Variation with respect to $\rho$.**
Let $\rho \to \rho + \delta\rho$. Then the variation of the augmented action is given by

$$\delta A^{\dagger} = \int_0^1 \int_{\mathbb{R}^d} \left[ \left( \frac{1}{2}\|\mathbf{u}\|^2 + \alpha\,\psi(g) \right)\delta\rho + \lambda\left( \frac{\partial\delta\rho}{\partial t} + \nabla_{\mathbf{x}} \cdot (\mathbf{u}\,\delta\rho) - \frac{1}{2}\sigma^2\,\nabla_{\mathbf{x}}^2(\delta\rho) - g\,\delta\rho \right) \right] \mathrm{d}\mathbf{x}\,\mathrm{d}t$$

$$= \int_0^1 \int_{\mathbb{R}^d} \left( \frac{1}{2}\|\mathbf{u}\|^2 + \alpha\,\psi(g) - \lambda g \right)\delta\rho\,\mathrm{d}\mathbf{x}\,\mathrm{d}t + \int_{\mathbb{R}^d} \int_0^1 \left( \frac{\partial(\lambda\,\delta\rho)}{\partial t} - \delta\rho\,\frac{\partial\lambda}{\partial t} \right)\mathrm{d}t\,\mathrm{d}\mathbf{x}$$

$$+ \int_0^1 \int_{\mathbb{R}^d} \left( \nabla_{\mathbf{x}} \cdot (\mathbf{u}\,\lambda\,\delta\rho)\,\mathrm{d}\mathbf{x}\,\mathrm{d}t - \mathbf{u}^T(\nabla_{\mathbf{x}}\lambda)\,\delta\rho \right)\,\mathrm{d}\mathbf{x}\,\mathrm{d}t$$

$$- \frac{1}{2}\sigma^2 \int_0^1 \int_{\mathbb{R}^d} \left( \nabla_{\mathbf{x}} \cdot (\lambda\,\nabla_{\mathbf{x}}\delta\rho) - (\nabla_{\mathbf{x}}\lambda)^T(\nabla_{\mathbf{x}}\delta\rho) \right)\mathrm{d}\mathbf{x}\,\mathrm{d}t$$

$$= \int_0^1 \int_{\mathbb{R}^d} \left( \frac{1}{2}\|\mathbf{u}\|^2 + \alpha\,\psi(g) - \lambda g - \frac{\partial\lambda}{\partial t} - \mathbf{u}^T(\nabla_{\mathbf{x}}\lambda) \right)\delta\rho\,\mathrm{d}\mathbf{x}\,\mathrm{d}t + \cancel{\int_{\mathbb{R}^d}[\lambda\,\delta\rho]_{t=0}^{t=1}\mathrm{d}\mathbf{x}}$$

$$+ \cancel{\int_0^1 \int_{S^\infty} \nabla_{\mathbf{x}} \cdot (\mathbf{u}\,\lambda\,\delta\rho)^T\,\mathrm{d}\mathbf{S}\,\mathrm{d}t} - \cancel{\frac{1}{2}\sigma^2 \int_0^1 \int_{S^\infty} (\lambda\,\nabla_{\mathbf{x}}\delta\rho)^T\,\mathrm{d}\mathbf{S}\,\mathrm{d}t}$$

$$+ \frac{1}{2}\sigma^2 \int_0^1 \int_{\mathbb{R}^d} (\nabla_{\mathbf{x}}\lambda)^T(\nabla_{\mathbf{x}}\delta\rho)\mathrm{d}\mathbf{x}\,\mathrm{d}t$$

$$= \int_0^1 \int_{\mathbb{R}^d} \left( \frac{1}{2}\|\mathbf{u}\|^2 + \alpha\,\psi(g) - \lambda g - \frac{\partial\lambda}{\partial t} - \mathbf{u}^T(\nabla_{\mathbf{x}}\lambda) \right)\delta\rho\,\mathrm{d}\mathbf{x}\,\mathrm{d}t$$

$$+ \frac{1}{2}\sigma^2 \int_0^1 \int_{\mathbb{R}^d} \left( \nabla_{\mathbf{x}} \cdot ((\nabla_{\mathbf{x}}\lambda)\,\delta\rho) - (\nabla_{\mathbf{x}}^2\lambda)\,\delta\rho \right)\mathrm{d}\mathbf{x}\,\mathrm{d}t$$

$$= \int_0^1 \int_{\mathbb{R}^d} \left( \frac{1}{2}\|\mathbf{u}\|^2 + \alpha\,\psi(g) - \lambda g - \frac{\partial\lambda}{\partial t} - \mathbf{u}^T(\nabla_{\mathbf{x}}\lambda) - \frac{1}{2}\sigma^2\,\nabla_{\mathbf{x}}^2\lambda \right)\delta\rho\,\mathrm{d}\mathbf{x}\,\mathrm{d}t$$

$$- \cancel{\frac{1}{2}\sigma^2 \int_0^1 \int_{S^\infty} ((\nabla_{\mathbf{x}}\lambda)\,\delta\rho)^T\,\mathrm{d}\mathbf{S}\,\mathrm{d}t}$$

$$= \int_0^1 \int_{\mathbb{R}^d} \left( \frac{1}{2}\|\mathbf{u}\|^2 + \alpha\,\psi(g) - \lambda g - \frac{\partial\lambda}{\partial t} - \mathbf{u}^T(\nabla_{\mathbf{x}}\lambda) - \frac{1}{2}\sigma^2\,\nabla_{\mathbf{x}}^2\lambda \right)\delta\rho\,\mathrm{d}\mathbf{x}\,\mathrm{d}t.$$

Since $\delta\rho$ is arbitrary, the corresponding optimality condition is

$$\frac{1}{2}\|\mathbf{u}\|^2 + \alpha\,\psi(g) - \lambda g - \frac{\partial\lambda}{\partial t} - \mathbf{u}^T(\nabla_{\mathbf{x}}\lambda) - \frac{1}{2}\sigma^2\nabla_{\mathbf{x}}^2\lambda = 0.$$

Substituting the previously obtained condition $\mathbf{u} = \nabla_{\mathbf{x}}\lambda$, we arrive at the final optimality condition:

$$\frac{\partial\lambda}{\partial t} + \frac{1}{2}\|\nabla_{\mathbf{x}}\lambda\|^2 + \frac{1}{2}\sigma^2\nabla_{\mathbf{x}}^2\lambda + \lambda g - \alpha\,\psi(g) = 0.$$

$\square$

### A.1.2 Proof for Theorem 4.2

**Theorem A.2.** *The choice of $\psi(g)$ affects whether $g$ ascends or descends along the direction of the velocity field $\mathbf{u}$ at a given time. Specifically, at a fixed time $t$, if*

$$\frac{\mathrm{d}\psi^2(g)}{\mathrm{d}g^2} > 0, \tag{9}$$

*then $g(\mathbf{x}, t)$ ascends in the direction of the velocity field $\mathbf{u}(\mathbf{x}, t)$ (i.e., $\mathbf{u}(\mathbf{x}, t)^T(\nabla_{\mathbf{x}} g(\mathbf{x}, t)) > 0$); otherwise, it descends.*

*Proof.* Let

$$\frac{\mathrm{d}\psi(g)}{\mathrm{d}g} = \psi'(g) \quad \text{and} \quad \frac{\mathrm{d}^2\psi(g)}{\mathrm{d}g^2} = \psi''(g).$$

Using the optimality condition for $g$ from Appendix A.1.1,

$$\alpha \frac{\mathrm{d}\psi(g)}{\mathrm{d}g} = \lambda,$$

taking the gradient with respect to $x$ on both sides yields

$$\nabla_{\mathbf{x}} g(\mathbf{x}, t) = \frac{1}{\alpha\,\psi''(g)}\,\nabla_{\mathbf{x}}\lambda(\mathbf{x}, t).$$

The condition for $g$ to increase along the velocity field is that the inner product between $\nabla_{\mathbf{x}} g(\mathbf{x}, t)$ and $u(\mathbf{x}, t)$ is positive everywhere. Using the optimality condition for the velocity,

$$\mathbf{u}(\mathbf{x}, t) = \nabla_{\mathbf{x}}\lambda(\mathbf{x}, t),$$

we have

$$\mathbf{u}(\mathbf{x}, t)^T \nabla_{\mathbf{x}} g(\mathbf{x}, t) = \frac{1}{\alpha\,\psi''(g)}\,\|\nabla_{\mathbf{x}}\lambda(\mathbf{x}, t)\|^2.$$

Since $\alpha > 0$, the condition

$$\mathbf{u}(\mathbf{x}, t)^T \nabla_{\mathbf{x}} g(\mathbf{x}, t) > 0$$

is equivalent to requiring that

$$\psi''(g) > 0 \quad \forall g.$$

$\square$

### A.1.3 Proof for Theorem 5.1

**Theorem A.3.** *Consider a weighted particle system consisting of $N$ particles, where the position of particle $i$ is given by $\mathbf{X}_i^t \in \mathbb{R}^d$ and its weight by $w_i(t) > 0$. The dynamics of each particle are described by*

$$\frac{\mathrm{d}\mathbf{X}_i^t}{\mathrm{d}t} = \mathbf{u}(\mathbf{X}_i^t, t)\,\mathrm{d}t + \sigma(t)\,\mathrm{d}\mathbf{W}_t,$$

$$\frac{\mathrm{d}w_i}{\mathrm{d}t} = g(\mathbf{X}_i^t, t)\,w_i\,\mathrm{d}t, \tag{10}$$

*where $\mathbf{u} : \mathbb{R}^d \times [0, T] \to \mathbb{R}^d$ is a time-varying vector field, $g : \mathbb{R}^d \times [0, T] \to \mathbb{R}$ is a growth rate function, $\sigma : [0, T] \to [0, +\infty)$ is a time-varying diffusion coefficient, and $\mathbf{W}_t$ is an $N$-dimensional standard Brownian motion with independent components in each coordinate. The initial conditions are $\mathbf{X}_i^0 \sim \rho(\mathbf{x}, 0)$ and $w_i(0) = 1$. In the limit as $N \to \infty$, the empirical measure*

$$\mu^N(\mathbf{x}, t) = \frac{1}{N}\sum_{i=1}^N w_i(t)\,\delta\big(\mathbf{x} - \mathbf{X}_i^t\big) \tag{11}$$

*converges to the solution of the following Fokker–Planck equation:*

$$\frac{\partial\rho(\mathbf{x}, t)}{\partial t} = -\nabla_{\mathbf{x}} \cdot \Big(\mathbf{u}(\mathbf{x}, t)\,\rho(\mathbf{x}, t)\Big) + \frac{1}{2}\sigma^2(t)\nabla_{\mathbf{x}}^2\rho(\mathbf{x}, t) + g(\mathbf{x}, t)\,\rho(\mathbf{x}, t), \tag{12}$$

*with the initial condition $\rho(\mathbf{x}, 0) = \rho_0(\mathbf{x})$.*

*Proof.* Consider a smooth test function $\phi : \mathbb{R}^d \to \mathbb{R}$. We study the evolution of the expectation

$$\int_{\mathbb{R}^d} \phi(\mathbf{x})\, \mu^N(\mathbf{x}, t) \mathrm{d}\mathbf{x} = \frac{1}{N} \sum_{i=1}^{N} w_i(t)\, \phi(\mathbf{X}_i^t).$$

By applying Itô's formula, we have

$$\mathrm{d}\Big( w_i(t)\, \phi(\mathbf{X}_i^t)\Big) = w_i(t)\, \mathrm{d}\phi(\mathbf{X}_i^t) + \phi(\mathbf{X}_i^t)\, \mathrm{d}w_i(t) + \mathrm{d}w_i(t)\, \mathrm{d}\phi(\mathbf{X}_i^t).$$

Using Itô's formula to compute $\mathrm{d}\phi(\mathbf{X}_i^t)$, we obtain

$$\mathrm{d}\phi(\mathbf{X}_i^t) = (\nabla_{\mathbf{x}}\phi(\mathbf{X}_i^t))^T \, \mathrm{d}\mathbf{X}_i^t + \frac{1}{2}\nabla_{\mathbf{x}}^2\phi(\mathbf{X}_i^t)\, \sigma^2(t)\, \mathrm{d}t.$$

Since $\mathrm{d}w_i(t) = g(\mathbf{X}_i^t, t)\, w_i(t)\, \mathrm{d}t$ contains no stochastic term (i.e., there is no $\mathrm{d}\mathbf{W}$), the term $\mathrm{d}w_i(t)\, \mathrm{d}\phi(\mathbf{X}_i^t)$ is of higher order and can be neglected. Therefore, we have:

$$\begin{aligned}
\mathrm{d}\Big( w_i(t)\, \phi(\mathbf{X}_i^t)\Big) = {}& \phi(\mathbf{X}_i^t)\, g(\mathbf{X}_i^t, t)\, w_i(t)\, \mathrm{d}t \\
& + w_i(t)\, (\nabla_{\mathbf{x}}\phi(\mathbf{X}_i^t))^T \Big( \mathbf{u}(\mathbf{X}_i^t, t)\, \mathrm{d}t + \sigma(t)\, \mathrm{d}\mathbf{W}_t \Big) \\
& + \frac{1}{2}\nabla_{\mathbf{x}}^2\phi(\mathbf{X}_i^t)\, \sigma^2(t)\, \mathrm{d}t.
\end{aligned}$$

Next, we compute

$$\begin{aligned}
\mathbb{E}\left[ \frac{\mathrm{d}}{\mathrm{d}t}\Big( \frac{1}{N}\sum_{i=1}^{N} w_i(t)\, \phi(\mathbf{X}_i^t)\Big)\right] = \mathbb{E}\Bigg[ \frac{1}{N}\sum_{i=1}^{N}\Big( & \phi(\mathbf{X}_i^t)\, g(\mathbf{X}_i^t, t)\, w_i(t) \\
& + w_i(t)\, (\nabla_{\mathbf{x}}\phi(\mathbf{X}_i^t))^T \mathbf{u}(\mathbf{X}_i^t, t) \\
& + \frac{1}{2}\nabla_{\mathbf{x}}^2\phi(\mathbf{X}_i^t)\, \sigma^2(t)\Big)\Bigg].
\end{aligned}$$

Thus, in the limit as $N \to \infty$, and let $\rho(\mathbf{x}, t) = \mu^\infty(\mathbf{x}, t)$, we have

$$\begin{aligned}
\frac{\mathrm{d}}{\mathrm{d}t}\int_{\mathbb{R}^d} \phi(\mathbf{x})\, \rho(\mathbf{x}, t)\, \mathrm{d}\mathbf{x} = \int_{\mathbb{R}^d} \Big( & g(\mathbf{x}, t)\, \rho(\mathbf{x}, t)\phi(\mathbf{x}) + \rho(\mathbf{x}, t)(\nabla_{\mathbf{x}}\phi(\mathbf{x}))^T\mathbf{u}(\mathbf{x}, t) \\
& + \frac{1}{2}\sigma^2(t)\rho(\mathbf{x}, t)\nabla_{\mathbf{x}}^2\phi(\mathbf{x})\Big)\mathrm{d}\mathbf{x}.
\end{aligned}$$

By integrating by parts, we obtain

$$\int_{\mathbb{R}^d} \rho(\mathbf{x}, t)\, (\nabla_{\mathbf{x}}\phi(\mathbf{x}))^T\mathbf{u}(\mathbf{x}, t)\, \mathrm{d}\mathbf{x} = -\int_{\mathbb{R}^d} \phi(\mathbf{x}, t)\, \nabla_{\mathbf{x}} \cdot \Big( \mathbf{u}(\mathbf{x}, t)\, \rho(\mathbf{x}, t)\Big)\mathrm{d}\mathbf{x},$$

and

$$\int_{\mathbb{R}^d} \rho(\mathbf{x}, t)\, \nabla_{\mathbf{x}}^2\phi(\mathbf{x})\, \mathrm{d}\mathbf{x} = \int_{\mathbb{R}^d} \phi(\mathbf{x}, t)\, \nabla_{\mathbf{x}}^2\rho(\mathbf{x}, t)\, \mathrm{d}\mathbf{x}.$$

Hence, we deduce that

$$\frac{\mathrm{d}}{\mathrm{d}t}\int_{\mathbb{R}^d} \phi(\mathbf{x}, t)\, \rho(\mathbf{x}, t)\, \mathrm{d}x = \int_{\mathbb{R}^d} \phi(\mathbf{x}, t)\Big[ -\nabla_{\mathbf{x}}\cdot\big(\mathbf{u}(\mathbf{x}, t)\rho(\mathbf{x}, t)\big) + \frac{1}{2}\sigma^2(t)\nabla_{\mathbf{x}}^2\rho(\mathbf{x}, t) + g(\mathbf{x}, t)\rho(\mathbf{x}, t)\Big]\mathrm{d}x.$$

Since $\phi(\mathbf{x})$ is arbitrary, we obtain the Fokker–Planck equation:

$$\frac{\partial \rho(\mathbf{x}, t)}{\partial t} = -\nabla_{\mathbf{x}} \cdot \Big( \mathbf{u}(\mathbf{x}, t)\, \rho(\mathbf{x}, t)\Big) + \frac{1}{2}\sigma^2(t)\nabla_{\mathbf{x}}^2\rho(\mathbf{x}, t) + g(\mathbf{x}, t)\, \rho(\mathbf{x}, t).$$

$$\square$$

### A.1.4 Proposition : the Expectation of HJB Loss

**Proposition A.1.** *Consider the following HJB loss:*

$$\mathcal{L}_{HJB}^N = \sum_{i=1}^N \left[ \int_0^{T_K} \frac{w_i(t)}{\sum_{i=1}^N w_i(t)} \left( \frac{\partial \lambda(\mathbf{X}_i^t, t)}{\partial t} + \frac{1}{2} \|\nabla_{\mathbf{x}} \lambda\|^2 + \frac{1}{2} \sigma^2 \nabla_{\mathbf{x}}^2 \lambda + g(\mathbf{X}_i^t, t) - \alpha\,\psi(g) \right)^2 dt \right] \tag{13}$$

*where*

$$d\mathbf{X}_i^t = \mathbf{u}(\mathbf{X}_i^t, t)\,dt + \sigma(t)\,d\mathbf{W}_t, \quad dw_i = g(\mathbf{X}_i^t, t)\,w_i\,dt,$$

*The expectation of HJB loss is*

$$\mathbb{E}[\mathcal{L}_{HJB}^N] = \int_0^{T_K} \int_{\mathbb{R}^d} \hat{\rho}(\mathbf{x}, t) \left( \frac{\partial \lambda(\mathbf{x}, t)}{\partial t} + \frac{1}{2} \|\nabla_{\mathbf{x}} \lambda\|^2 + \frac{1}{2} \sigma^2 \nabla_{\mathbf{x}}^2 \lambda + g(\mathbf{x}, t) - \alpha\,\psi(g) \right)^2 d\mathbf{x}dt \tag{14}$$

*where* $\hat{\rho}(\mathbf{x}, t) = \dfrac{\rho(\mathbf{x}, t)}{\int_{\mathbb{R}^d} \rho(\mathbf{x}, t)d\mathbf{x}}$ *is the normalized probability density.*

*Proof.* Taking the expectation of $\mathcal{L}_{HJB}^N$ is equivalent to drawing $N$ particles each time to obtain $\mathcal{L}_{HJB}^N$, repeating this process infinitely many times, and computing the average of the $\mathcal{L}_{HJB}^N$ obtained in each instance. Since the particles are independent, this operation is directly equivalent to taking the number of particles $N \to \infty$, thus:

$$\mathbb{E}[\mathcal{L}_{HJB}^N] = \lim_{N \to \infty} \sum_{i=1}^N \left[ \int_0^{T_K} \frac{w_i(t)}{\sum_{j=1}^N w_i(t)} \left( \frac{\partial \lambda}{\partial t} + \frac{1}{2} \|\nabla_{\mathbf{x}} \lambda\|^2 + \frac{1}{2} \sigma^2 \nabla_{\mathbf{x}}^2 \lambda + g(\mathbf{X}_i^t, t) - \alpha\,\psi(g) \right)^2 dt \right]$$

$$= \lim_{N \to \infty} \frac{1}{N} \sum_{i=1}^N \left[ \int_0^{T_K} \frac{w_i(t)}{\frac{1}{N}\sum_{j=1}^N w_i(t)} \left( \frac{\partial \lambda}{\partial t} + \frac{1}{2} \|\nabla_{\mathbf{x}} \lambda\|^2 + \frac{1}{2} \sigma^2 \nabla_{\mathbf{x}}^2 \lambda + g - \alpha\,\psi(g) \right)^2 dt \right]$$

$$= \lim_{N \to \infty} \left[ \int_0^{T_K} \int_{\mathbb{R}} \frac{1}{\int_{\mathbb{R}^d} \mu^N d\mathbf{x}} \mu^N(\mathbf{x}, t) \left( \frac{\partial \lambda}{\partial t} + \frac{1}{2} \|\nabla_{\mathbf{x}} \lambda\|^2 + \frac{1}{2} \sigma^2 \nabla_{\mathbf{x}}^2 \lambda + g - \alpha\,\psi(g) \right)^2 d\mathbf{x}dt \right]$$

$$= \int_0^{T_K} \int_{\mathbb{R}} \frac{1}{\lim\limits_{N \to \infty} \int_{\mathbb{R}^d} \mu^N d\mathbf{x}} \lim_{N \to \infty} \left( \mu^N \left( \frac{\partial \lambda}{\partial t} + \frac{1}{2} \|\nabla_{\mathbf{x}} \lambda\|^2 + \frac{1}{2} \sigma^2 \nabla_{\mathbf{x}}^2 \lambda + g - \alpha\,\psi(g) \right)^2 d\mathbf{x}dt \right)$$

$$= \int_0^{T_K} \int_{\mathbb{R}^d} \hat{\rho}(\mathbf{x}, t) \left( \frac{\partial \lambda(\mathbf{x}, t)}{\partial t} + \frac{1}{2} \|\nabla_{\mathbf{x}} \lambda\|^2 + \frac{1}{2} \sigma^2 \nabla_{\mathbf{x}}^2 \lambda + g(\mathbf{x}, t) - \alpha\,\psi(g) \right)^2 d\mathbf{x}dt$$

In the final equality of the proof, we employed the previously proven Appendix A.1.3. $\qquad\square$

### A.1.5 Proposition : the Expectation of Action Loss

**Proposition A.2.** *Consider the following action loss:*

$$\mathcal{L}_{Action}^N = \frac{1}{N} \sum_{i=1}^N \left( \int_0^1 \frac{1}{2} \|\mathbf{u}(\mathbf{X}_i^t, t)\|^2 w_i(t)dt + \int_0^1 \alpha\psi(g(\mathbf{X}_i^t, t))w_i(t)dt \right) \tag{15}$$

*where*

$$d\mathbf{X}_i^t = \mathbf{u}(\mathbf{X}_i^t, t)\,dt + \sigma(t)\,d\mathbf{W}_t, \quad dw_i = g(\mathbf{X}_i^t, t)\,w_i\,dt,$$

*The expectation of action loss is equal to the action defined in the RUOT formulation, namely,*

$$\mathbb{E}[\mathcal{L}_{Action}^N] = \int_0^{T_K} \int_{\mathbb{R}^d} \left( \frac{1}{2} \|\mathbf{u}(\mathbf{x}, t)\|^2 + \alpha\,\psi(g(\mathbf{x}, t)) \right) \rho(\mathbf{x}, t)\,d\mathbf{x}\,dt. \tag{16}$$

*Proof.* Taking the expectation of $\mathcal{L}_{\text{Action}}^{N}$ is equivalent to drawing $N$ particles each time to obtain $\mathcal{L}_{\text{Action}}^{N}$, repeating this process infinitely many times, and computing the average of the $\mathcal{L}_{\text{Action}}^{N}$ obtained in each instance. Since the particles are independent, this operation is directly equivalent to taking the number of particles $N \to \infty$, thus:

$$
\begin{aligned}
\mathbb{E}[\mathcal{L}_{\text{Action}}^{N}] &= \lim_{N\to\infty} \frac{1}{N} \sum_{i=1}^{N} \left[ \int_{0}^{T_K} \frac{1}{2} \|\mathbf{u}(\mathbf{X}_i^t, t)\|^2 \, w_i(t)\, dt + \int_{0}^{T_K} \alpha\, \psi\big(g(\mathbf{X}_i^t, t)\big)\, w_i(t)\, dt \right] \\
&= \lim_{N\to\infty} \frac{1}{N} \sum_{i=1}^{N} \left[ \int_{0}^{T_K} \int_{\mathbb{R}^d} \frac{1}{2} \|\mathbf{u}(\mathbf{x}, t)\|^2 \, \mu^N(\mathbf{x})\, dt \right. \\
&\qquad\qquad\qquad \left. + \int_{0}^{T_K} \int_{\mathbb{R}^d} \alpha\, \psi\big(g(\mathbf{x}, t)\big)\, \mu^N(\mathbf{x})\, dt \right] \\
&= \int_{0}^{T_K} \int_{\mathbb{R}^d} \left( \frac{1}{2}\|\mathbf{u}(\mathbf{x}, t)\|^2 + \alpha\, \psi(g(\mathbf{x}, t)) \right) \rho(\mathbf{x}, t)\, d\mathbf{x}\, dt.
\end{aligned}
$$

In the final equality of the proof, we employed the previously proven Appendix A.1.3. $\qquad\square$

## A.2 Optimality Conditions Under Different $\psi(g)$

In our Experiment, we use two different $\psi(g)$ as examples. When $\psi_1(g) = \frac{1}{2}g^2$, the optimality conditions are :

$$
\mathbf{u} = \nabla_{\mathbf{x}} \lambda, \quad g = \frac{\lambda}{\alpha}, \quad \frac{\partial \lambda}{\partial t} + \frac{1}{2}\|\nabla_{\mathbf{x}} \lambda\|^2 + \frac{1}{2}\sigma^2 \nabla_{\mathbf{x}}^2 \lambda + \frac{1}{2}\frac{\lambda^2}{\alpha} = 0.
$$

When $\psi_2(g) = g^{2/15}$, the optimality conditions are :

$$
\mathbf{u} = \nabla_{\mathbf{x}} \lambda, \quad g = \left( \frac{2\alpha}{15\,\lambda} \right)^{\frac{15}{13}}, \quad \frac{\partial \lambda}{\partial t} + \frac{1}{2}\|\nabla_{\mathbf{x}} \lambda\|^2 + \frac{1}{2}\sigma^2 \nabla_{\mathbf{x}}^2 \lambda - \frac{13}{15}\alpha \left( \frac{2\alpha}{15\,\lambda} \right)^{\frac{2}{13}} = 0.
$$

Note that in this case the function $g(\lambda)$ exhibits a singularity at $\lambda = 0$. In fact, given the two properties we imposed on $\psi(g)$ ($\frac{d\psi(g)}{d|g|} > 0$ and $\psi(g) = \psi(-g)$) along with the constraint $\psi''(g) < 0$, it follows that $\psi'(g)$ must be discontinuous at 0, and hence $g(\lambda) = (\psi')^{-1}\left(\frac{\lambda(\mathbf{x},t)}{\alpha}\right)$ necessarily has a singularity at $\lambda = 0$. For the sake of training stability, we slightly modify $g(\lambda)$ to remove this singularity. We redefine $g(\lambda)$ as:

$$
g^{\dagger}(\lambda) = \begin{cases} g(\lambda), & \lambda \geq \delta, \\ \frac{1}{2\delta}(g(\delta) - g(-\delta))(\lambda + \delta) + g(-\delta), & -\delta \leq \lambda < \delta, \\ g(-\lambda), & \lambda < -\delta, \end{cases}
$$

where $\delta$ is a small positive constant. In our computations, we set $\delta = 0.1$. Another possible treatment for this singularity involves fitting the reciprocal of $\lambda$, $\mu$, thereby canceling the singularity at $\lambda = 0$. In this case, $g = \left(\frac{2\alpha\mu}{15}\right)^{\frac{15}{13}}$, and thus the singularity near zero points no longer exists. We leave how to eliminate this singularity more systematically in numerical computation as the future work.

## A.3 Training Algorithm and Computation of Losses

For the algorithms mentioned above, we use the particle method to compute the losses. In Appendix D.2, we discuss the convergence rates of these losses: both $\mathcal{L}$HJB and $\mathcal{L}$Action have a convergence rate of $\mathcal{O}\left(\frac{1}{\sqrt{N}}\right)$, while for data with high dimensionality ($d \gg 1$), the convergence

---
**Algorithm 1** Training Var-RUOT
---
**Require:** Datasets $D_1, \ldots, D_K$, batch size $N$, training epochs $N_{\text{Epoch}}$, initialized network $\lambda_\theta(\mathbf{x}, t)$.
**Ensure:** Trained scalar field $\lambda_\theta(\mathbf{x}, t)$.

1: **for** $i = 1$ to $N_{\text{Epoch}}$ **do**
2:   From the data at the first time point, $D_1$, sample $N$ particles and set all their weights to $w_i(0) = 1$, for $i = \{1, 2, \cdots, N\}$.
3:   **for** $j = 1$ to $K$ **do**
4:    use optimality conditions $\mathbf{u}_\theta(\mathbf{x}, t) = \nabla_{\mathbf{x}} \lambda_\theta(\mathbf{x}, t)$, $\alpha \dfrac{\mathrm{d}\psi(g_\theta)}{\mathrm{d}g_\theta} = \lambda_\theta(\mathbf{x}, t)$ to calculate $\mathbf{u}_\theta(\mathbf{x}, t)$ and $g_\theta(\mathbf{x}, t)$, where $t \in [T_i, T_{i+1})$

5:    $\mathcal{L}_{\text{Action}}^N \leftarrow \mathcal{L}_{\text{Action}}^N +$
$$\frac{1}{N} \sum_{i=1}^{N} \left( \int_{T_i}^{T_{i+1}} \frac{1}{2} \|\mathbf{u}_\theta(\mathbf{X}_i^t, t)\|^2 w_i(t)\mathrm{d}t + \int_{T_i}^{T_{i+1}} \alpha\psi(g_\theta(\mathbf{X}_i^t, t))w_i(t)\mathrm{d}t \right)$$

6:    $\mathcal{L}_{\text{HJB}}^N \leftarrow \mathcal{L}_{\text{HJB}}^N + \sum_{i=1}^{N} \left[ \int_{T_i}^{T_{i+1}} \frac{w_i(t)}{\sum_{j=1}^{N} w_j(t)} \right.$
$$\left. \times \left( \frac{\partial \lambda_\theta(\mathbf{X}_i^t, t)}{\partial t} + \frac{1}{2} \|\nabla_{\mathbf{x}} \lambda_\theta\|^2 + \frac{1}{2} \sigma^2 \nabla_{\mathbf{x}}^2 \lambda_\theta + g_\theta(\mathbf{X}_i^t, t) - \alpha\,\psi(g_\theta) \right)^2 \mathrm{d}t \right]$$

7:    $\mathcal{L}_{\text{Recon}} \leftarrow \mathcal{L}_{\text{Recon}} + (M(T_{i+1}) - \hat{M}(T_{i+1}))^2 + \mathcal{W}_2(\tilde{\rho}(\cdot, T_{i+1}),\, \hat{\tilde{\rho}}(\cdot, T_{i+1}))$
8:   $\mathcal{L}_{\text{Total}} = \gamma_{\text{Action}}\mathcal{L}_{\text{Action}} + \gamma_{\text{HJB}}\mathcal{L}_{\text{HJB}} + \mathcal{L}_{\text{Recon}}$
9:   update $\lambda_\theta(\mathbf{x}, t)$ w.r.t. $\mathcal{L}_{\text{Total}}$

---

rate of $\mathcal{L}_{\text{OT}}$ is $\mathcal{O}(N^{-\frac{2}{d}})$. To balance training speed, memory consumption, and the algorithm's convergence rate, we recommend using approximately $N = 10^3$ to $10^4$ particles to compute $\mathcal{L}_{\text{HJB}}$, $\mathcal{L}_{\text{Action}}$, and $\mathcal{L}_{\text{Recon}}$. Furthermore, to ensure high accuracy in the integration for $\mathcal{L}_{\text{HJB}}$ and $\mathcal{L}_{\text{Action}}$, we advise using a high-order integrator (Berman et al. 2024) (e.g., a Stochastic Runge-method (Li et al. 2020)) and setting the integration step size to approximately $\Delta t = 0.1$. $\mathcal{L}_{\text{Mass}}$ measures the discrepancy between the mass learned by the model and the true mass. The true mass at time point $i$ (for $i > 1$) is defined as the ratio of the number of data points at that time, $N_i$, to the number of data points at the initial time, $N_1$.

In particular, computing $\mathcal{L}_{\text{HJB}}$ requires second-order derivatives to calculate the Laplacian $\nabla_x^2 \lambda(\boldsymbol{x}, t)$. This step can be computationally expensive. However, we only need the trace of the Hessian (i.e., the Laplacian), not the full matrix. This can be efficiently estimated using Hutchinson's randomized trace estimator. For a Hessian matrix $H$, the trace is given by $\text{trace}(H) = \mathbb{E}_{\boldsymbol{v}}[\boldsymbol{v}^T H \boldsymbol{v}]$, where the vector $\boldsymbol{v}$ is drawn from a distribution with zero mean and unit covariance, such as $\mathcal{N}(0, I)$.

## B   Experiential Details

### B.1   Additional Information for Datasets

**Simulation Dataset**   In the main text, we utilize a simulated dataset derived from a three-gene regulatory network (Zhang et al. 2025a). The dynamics of this system are governed by stochastic differential equations that incorporate self-activation, mutual inhibition, and external activation. The dynamics of the three genes are described by the following equations:

$$\frac{\mathrm{d}X_1^i}{\mathrm{d}t} = \frac{\alpha_1 (X_1^i)^2 + \beta}{1 + \gamma_1 (X_1^i)^2 + \alpha_2 (X_2^i)^2 + \gamma_3 (X_3^i)^2 + \beta} - \delta_1 X_1^i + \eta_1 \xi_t,$$

$$\frac{\mathrm{d}X_2^i}{\mathrm{d}t} = \frac{\alpha_2 (X_2^i)^2 + \beta}{1 + \gamma_1 (X_1^i)^2 + \alpha_2 (X_2^i)^2 + \gamma_3 (X_3^i)^2 + \beta} - \delta_2 X_2^i + \eta_2 \xi_t,$$

$$\frac{\mathrm{d}X_3^i}{\mathrm{d}t} = \frac{\alpha_3 (X_3^i)^2}{1 + \alpha_3 (X_3^i)^2} - \delta_3 X_3^i + \eta_3 \xi_t,$$

where $\mathbf{X}^i(t)$ represents the gene expression levels of the $i$th cell at time $t$. The coefficients $\alpha_i$, $\gamma_i$, and $\beta$ control the strengths of self-activation, inhibition, and the external stimulus, respectively.

The parameters $\delta_i$ indicate the rates of gene degradation, and the terms $\eta_i \xi_t$ account for stochastic influences using additive white noise.

The probability of cell division is linked to the expression level of $X_2$ and is given by

$$g = \alpha_g \frac{X_2^2}{1 + X_2^2}.$$

When a cell divides, the resulting daughter cells are created with each gene perturbed by an independent random noise term, $\eta_d N(0, 1)$, around the parent cell's gene expression profile $(X_1(t), X_2(t), X_3(t))$. Detailed hyper-parameters are provided in Table 5. The initial population of cells is independently sampled from two normal distributions, $\mathcal{N}([2, 0.2, 0], 0.1)$ and $\mathcal{N}([0, 0, 2], 0.1)$. At every time step, any negative expression values are set to zero.

Table 5: Simulation parameters for the gene regulatory network.

| Parameter | Value | Description |
|---|---|---|
| $\alpha_1$ | 0.5 | Self-activation strength for $X_1$. |
| $\gamma_1$ | 0.5 | Inhibition strength exerted by $X_3$ on $X_1$. |
| $\alpha_2$ | 1 | Self-activation strength for $X_2$. |
| $\gamma_2$ | 1 | Inhibition strength exerted by $X_3$ on $X_2$. |
| $\alpha_3$ | 1 | Self-activation strength for $X_3$. |
| $\gamma_3$ | 10 | Half-saturation constant in the inhibition term. |
| $\delta_1$ | 0.4 | Degradation rate for $X_1$. |
| $\delta_2$ | 0.4 | Degradation rate for $X_2$. |
| $\delta_3$ | 0.4 | Degradation rate for $X_3$. |
| $\eta_1$ | 0.05 | Noise intensity for $X_1$. |
| $\eta_2$ | 0.05 | Noise intensity for $X_2$. |
| $\eta_3$ | 0.01 | Noise intensity for $X_3$. |
| $\eta_d$ | 0.014 | Noise intensity for perturbations during cell division. |
| $\beta$ | 1 | External signal activating $X_1$ and $X_2$. |
| $dt$ | 1 | Time step size. |
| Time Points | [0, 8, 16, 24, 32] | Discrete time points when data is recorded. |

**Other Datasets Used in Main Text** In addition to the three-gene simulated dataset, our main text also utilizes the EMT dataset and the Mouse Blood Hematopoiesis dataset. The EMT dataset is sourced from (Sha et al. 2024; Cook and Vanderhyden 2020) and is derived from A549 cancer cells undergoing TGFB1-induced epithelial-mesenchymal transition (EMT). It comprises data from four distinct time points, containing a total of 3133 cells, with each cell represented by 10 features obtained through PCA dimensionality reduction. Meanwhile, the Mouse Blood Hematopoiesis dataset covers 3 time points and includes 10,998 cells in total (Weinreb et al. 2020; Sha et al. 2024) and was reduced to 2-dimensional space using nonlinear dimensionality reduction .

**Gaussian Datasets** To examine the differences between the optimal transport paths solved by our algorithm and those obtained from traditional OT solvers, and to validate the scalability of our model (i.e., its learning capability on high-dimensional data), we constructed two 2D Gaussian datasets (a balanced version and an unbalanced version) and three high-dimensional Gaussian datasets.The 2D Gaussian dataset consists of two time points, where the distribution at each time point is a mixture of multiple Gaussian distributions, as illustrated in Figure 5.The three high-dimensional Gaussian datasets have dimensions of 50, 100, and 150, respectively. The 100-dimensional dataset was adapted from the experiments in DeepRUOT (Zhang et al. 2025a) The results of these high-dimensional datasets, after being reduced to two dimensions using PCA for visualization, are shown in Figure 5.

**Other High Dimensional Datasets** In addition, we employed four real-world datasets for our evaluation. The first is the Mouse Blood Hematopoiesis dataset from (Weinreb et al. 2020), which comprises 49,302 cells collected at three time points. We reduced its dimensionality to 50 using PCA, and a subset of this processed data was used for the experiments in our main text. The second is the Pancreatic $\beta$-cell Differentiation dataset from (Veres et al. 2019), consisting of 51,274 cells sampled across eight time points. We subsequently reduced its dimensionality to 30 via PCA. The third dataset, sourced from (Moon et al. 2019), profiles 16,819 cells sampled at five time points from human embryoid bodies (EBs), which developed via the spontaneous aggregation of stem cells in

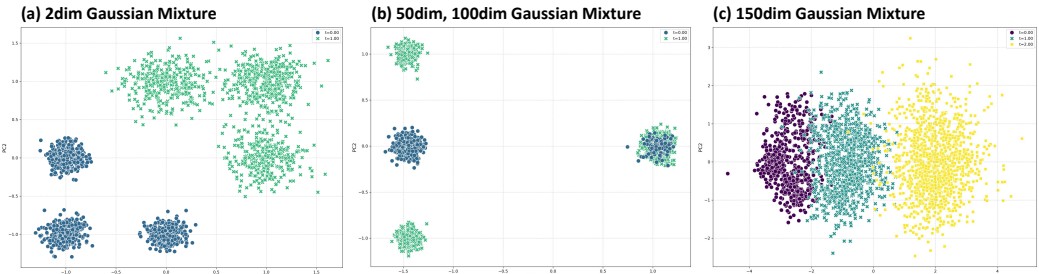

Figure 5: Diagram of the Gaussian mixture datasets. PCA is used to reduce the high dimensional data to 2 dimensions.

3D culture throughout a 27-day differentiation process. We reduced its dimensionality to 100 using PCA and then standardized each feature. The fourth dataset is from the NeurIPS Challenge (Luecken et al. 2021). From this, we utilized the gene expression data for the donor with DonorID 28483. We extracted the 1,079 cells from this donor that were annotated with pseudotime values. To create discrete time points, we performed K-Means clustering on these cells based on their pseudotime. The resulting clusters were then ordered by their centroid pseudotime values and assigned discrete labels $t = 0, 1, 2, 3, 4$. This data was also standardized on a per-feature basis.

## B.2 Evaluation Metrics

To assess the fitting accuracy of the learned dynamics to the data distribution, we compute the $\mathcal{W}_1$ and $\mathcal{W}_2$ distances between the data points generated by the model and the real data points. They are defined as

$$\mathcal{W}_1(p, q) = \min_{\pi \in \Pi(p,q)} \int \|\mathbf{x} - \mathbf{y}\|_2 \, \mathrm{d}\pi(\mathbf{x}, \mathbf{y}),$$

and

$$\mathcal{W}_2(p, q) = \left( \min_{\pi \in \Pi(p,q)} \int \|\mathbf{x} - \mathbf{y}\|_2^2 \, \mathrm{d}\pi(\mathbf{x}, \mathbf{y}) \right)^{1/2}.$$

We compute these two metrics using the `emd` function from the `pot` library.

To evaluate the action of the dynamics learned by the model, we directly compute the action loss. Appendix A.1.5 guarantees that the expectation of the loss is equal to the action defined in the RUOT problem. The action loss is:

$$\mathcal{L}_{\text{Action}}^N = \frac{1}{N} \sum_{i=1}^{N} \left( \int_0^1 \frac{1}{2} \|\mathbf{u}(\mathbf{X}_i^t, t)\|^2 w_i(t) \mathrm{d}t + \int_0^1 \alpha \psi(g(\mathbf{X}_i^t, t)) w_i(t) \mathrm{d}t \right)$$

where

$$\mathrm{d}\mathbf{X}_i^t = \mathbf{u}(\mathbf{X}_i^t, t) \, \mathrm{d}t + \sigma(t) \, \mathrm{d}\mathbf{W}_t,$$

$$\mathrm{d}w_i = g(\mathbf{X}_i^t, t) \, w_i \, \mathrm{d}t,$$

with initial conditions $\mathbf{X}_i^0 \sim \rho(\mathbf{x}, 0)$ and $w_i(0) = 1$. We run our model 5 times on each dataset, to calculate the mean and standard deviation of $\mathcal{W}_1, \mathcal{W}_2$ and action.

To evaluate the training speed of the model, we use the `SamplesLoss` class from the `geomloss` library to compute the OT loss at each epoch during the training process for each method, with the blur parameter set to $0.10$. We sum the OT losses at all time points to obtain the total OT loss. For each model, we perform 5 training runs, recording the number of epochs and the time required for the OT loss to drop below a specified threshold. We then compute the mean and standard deviation of these values, with the mean reflecting the training/convergence speed and the standard deviation reflecting the training stability.

For models whose dynamics are governed by stochastic differential equations, the choice of $\sigma$ directly affects the results (both the OT loss and the path action). Therefore, when running the RUOT and our Var-RUOT models on each dataset, $\sigma$ is set to $0.10$.

## C  Additional Experiment Results

### C.1  Additional Results on Training Speed and Stability

We plotted the average loss per epoch across five training runs in Fig. 6. Experimental results show that on the Simulation Gene dataset, our algorithm converges approximately 10 times faster than the fastest among the other algorithms (RUOT with 3-epoch pretraining), and on the EMT dataset, our algorithm converges roughly 20 times faster than the fastest alternative (TIGON).

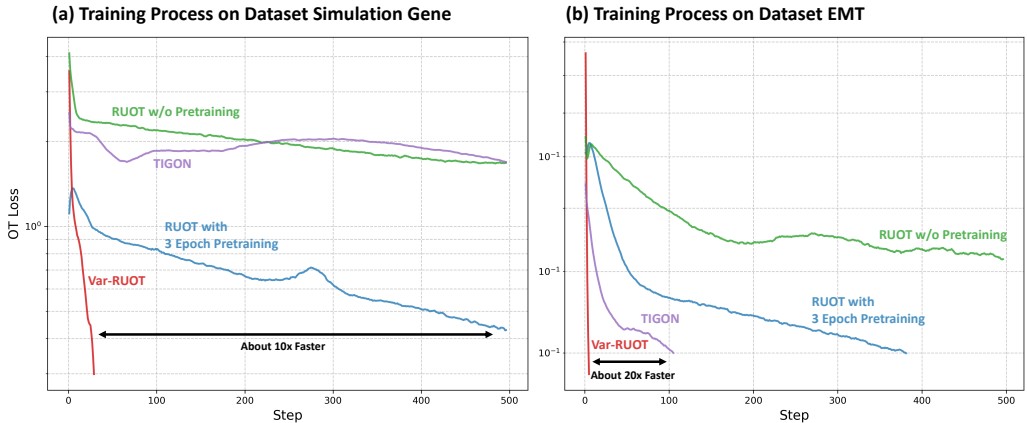

Figure 6: Loss curves of each algorithm over five training runs. The curves are obtained by averaging the losses from the five runs.

### C.2  Hyperparameter Selection and Ablation Study

**Hyperparameter Selection**  We used NVIDIA A100 GPUs (with 40G memory) and 128-core CPUs to conduct the experiments described in this paper. The neural network used to fit $\lambda(\mathbf{x}, t)$ is a fully connected network augmented with layer normalization and residual connections. It consists of 2 hidden layers, each with 512 dimensions. In our algorithm, the main hyperparameters that need tuning include the penalty coefficient $\alpha$ for growth in the action, and the weights $\gamma_{\text{HJB}}$ and $\gamma_{\text{Action}}$ for the two regularization losses, $\mathcal{L}_{\text{HJB}}$ and $\mathcal{L}_{\text{Action}}$, respectively. Here, $\alpha$ represents our prior regarding the strength of cell birth and death in the data: a larger $\alpha$ imposes a greater penalty on cell birth and death, thereby making it easier for the model to learn solutions with lower birth and death intensities. Meanwhile, the HJB loss and the action loss, serving as regularizers, are both designed to ensure that the solution obtained by the algorithm has as low an action as possible—the HJB equation being a necessary condition for the action to reach its minimum, and the inclusion of the action loss ensuring that the model learns a solution with an even smaller action under those necessary conditions.

To ensure that our algorithm generalizes well across a wide range of real-world datasets, we only used two sets of parameters: one for the standard WFR Metric ($\psi(g) = \frac{1}{2}g^2$) and one for the Modified Metric ($\psi(g) = g^{2/15}$). The parameters used in each case are listed in Table 6. The primary reason for using two sets is that different metrics yield different scales for the HJB loss.

Table 6: Parameter settings for standard WFR metric and modified metric.

| Parameter | $\gamma_{\text{HJB}}$ | $\gamma_{\text{Action}}$ | $\alpha$ | Learning Rate | Optimizer |
|---|---|---|---|---|---|
| Standard WFR Metric ($\psi_1(g) = \frac{1}{2}g^2$) | $6.25 \times 10^{-2}$ | $6.25 \times 10^{-2}$ | 2.00 | $1 \times 10^{-4}$ | AdamW |
| Modified Metric ($\psi_2(g) = g^{2/15}$) | $6.25 \times 10^{-3}$ | $6.25 \times 10^{-2}$ | 7.00 | $2 \times 10^{-5}$ | AdamW |

**Sensitivity Analysis of** $\alpha$  To demonstrate the robustness of our algorithm with respect to hyperparameter selection, we first varied the penalty coefficient $\alpha$ for growth and examined the resulting

changes in model performance. This sensitivity analysis was conducted on the 2D Mouse Blood Hematopoiesis dataset, and we performed the analysis for both the Standard WFR metric and the modified metric. The performance of the model under different values of $\alpha$ is shown in Table 7. The experimental results indicate that our algorithm is not sensitive to $\alpha$, as similar performance can be achieved across multiple different values of $\alpha$. In comparison with the Standard WFR metric, however, the algorithm appears to be somewhat more sensitive to $\alpha$ when the modified metric is used.

Table 7: On the Mouse Blood Hematopoiesis dataset, the Wasserstein distances ($\mathcal{W}_1$ and $\mathcal{W}_2$) between the predicted distributions of Var-RUOT with different $\alpha$ and the true distribution at each time points. Each experiment was run five times to compute the mean and standard deviation.

| Model | $t = 1$ | | $t = 2$ | |
|---|---|---|---|---|
| | $\mathcal{W}_1$ | $\mathcal{W}_2$ | $\mathcal{W}_1$ | $\mathcal{W}_2$ |
| Var-RUOT (Standard WFR, $\alpha = 1$) | $0.1622_{\pm 0.0072}$ | $0.2027_{\pm 0.0097}$ | $0.1280_{\pm 0.0123}$ | $0.1522_{\pm 0.0178}$ |
| Var-RUOT (Standard WFR, $\alpha = 2$) | $0.1203_{\pm 0.0060}$ | $0.1498_{\pm 0.0043}$ | $0.1389_{\pm 0.0068}$ | $0.1701_{\pm 0.0096}$ |
| Var-RUOT (Standard WFR, $\alpha = 3$) | $0.1402_{\pm 0.0054}$ | $0.1704_{\pm 0.0077}$ | $0.1350_{\pm 0.0100}$ | $0.1655_{\pm 0.0132}$ |
| Var-RUOT (Modified Metric, $\alpha = 5$) | $0.3783_{\pm 0.0194}$ | $0.3326_{\pm 0.0128}$ | $0.2110_{\pm 0.0164}$ | $0.2226_{\pm 0.0219}$ |
| Var-RUOT (Modified Metric, $\alpha = 7$) | $0.2953_{\pm 0.0357}$ | $0.3117_{\pm 0.0323}$ | $0.1917_{\pm 0.0140}$ | $0.2226_{\pm 0.0170}$ |
| Var-RUOT (Modified Metric, $\alpha = 9$) | $0.2737_{\pm 0.0095}$ | $0.3116_{\pm 0.0072}$ | $0.1970_{\pm 0.0072}$ | $0.2224_{\pm 0.0075}$ |

**Sensitivity Analysis of Growth Penalty Function**

In this work, we select $\psi_2(g) = g^{2/15}$ to illustrate a case where the conditions $\psi'(g) > 0$ and $\psi''(g) < 0$ hold. As established in Theorem 4.2, these conditions ensure that the growth rate $g(\boldsymbol{x}, t)$ increases along the direction of the velocity field $\boldsymbol{u}(\boldsymbol{x}, t)$ at each time step.

The specific choice of $\psi_2(g) = g^{2/15}$ is motivated by its simplicity and, more importantly, by considerations for training stability. For a general penalty of the form $\psi(g) = g^{(2p)/(2q+1)}$, the first-order optimality condition $\alpha \frac{\mathrm{d}\psi(g)}{\mathrm{d}g} = \lambda$ yields an analytical solution for $g$ in terms of $\lambda$:

$$g(\lambda) = \left( \frac{\lambda}{\alpha} \frac{2q+1}{2p} \right)^{\frac{2q+1}{2p-(2q+1)}} \propto \left( \frac{1}{\lambda} \right)^{\frac{2q+1}{(2q+1)-2p}}.$$

Since $g$ is computed directly from $\lambda$ during training, we must avoid an excessively steep $g(\lambda)$ curve, especially near $\lambda = 0$, to ensure stability. This requires the exponent $\frac{2q+1}{(2q+1)-2p}$ to be small. We achieve this by setting $p = 1$ (the smallest positive integer) and choosing a large value for $q$. Selecting $q = 7$ results in our final penalty function, $\psi_2(g) = g^{2/15}$, which balances simplicity and stability.

To empirically validate this choice, we conducted a sensitivity study on the 2D Mouse Blood Hematopoiesis dataset using different parameter choices for $\psi(g)$. The results, shown in Table 8, compare the Wasserstein distances ($\mathcal{W}_1$ and $\mathcal{W}_2$) at different time points.

Table 8: Sensitivity analysis on the choice of $\psi(g)$.

| Penalty Function $\psi(g)$ | $t = 1$ | | $t = 2$ | |
|---|---|---|---|---|
| | $\mathcal{W}_1$ | $\mathcal{W}_2$ | $\mathcal{W}_1$ | $\mathcal{W}_2$ |
| $\psi(g) = g^{2/9}$ ($p = 1, q = 4$) | $0.9476_{\pm 0.0835}$ | $1.0824_{\pm 0.0887}$ | $1.2197_{\pm 0.1450}$ | $1.3414_{\pm 0.1449}$ |
| $\psi(g) = g^{2/15}$ ($p = 1, q = 7$) | $0.2953_{\pm 0.0357}$ | $0.3117_{\pm 0.0323}$ | $0.1917_{\pm 0.0140}$ | $0.1764_{\pm 0.0135}$ |
| $\psi(g) = g^{2/21}$ ($p = 1, q = 10$) | $0.4239_{\pm 0.0181}$ | $0.5009_{\pm 0.0162}$ | $0.2638_{\pm 0.0146}$ | $0.2194_{\pm 0.0175}$ |

The results demonstrate that $\psi(g) = g^{2/15}$ achieves the best performance. Its performance is comparable to that of $\psi(g) = g^{2/21}$, whereas the choice of $\psi(g) = g^{2/9}$ (with a smaller $q$) leads to poorer reconstruction quality due to training instability. This confirms that selecting an exponent within a reasonable range (i.e., with a sufficiently large $q$) is effective and stable.

**Ablation Study of $\gamma_{\textbf{HJB}}$ and $\gamma_{\textbf{Action}}$** In order to verify whether $\mathcal{L}_{\text{HJB}}$ and $\mathcal{L}_{\text{Action}}$ facilitate the algorithm find solutions with lower action, we conducted ablation studies. These experiments were carried out on the EMT data and 2D Mouse Blood Hematopoiesis Data, since in this dataset the transition from the initial distribution to the terminal distribution can be achieved through relatively

simple dynamics (each particle moving in a roughly straight line). Therefore, if the HJB loss and the action loss are effective, the model will learn this simple dynamics rather than a more complex one. We varied the HJB loss weight $\gamma_{\text{HJB}}$ to the following values:$[0, 6.25 \times 10^{-3}, 3.125 \times 10^{-2}, 6.25 \times 10^{-2}, 6.25 \times 10^{-1}, 3.125]$, while keeping the action loss weight $\gamma_{\text{Action}}$ fixed at 1. We then shown both the mean $\mathcal{W}_1$ distances between the predicted and true distributions at four different time points and the trajectory action (as shown in Table 9 ). Similarly, we fixed $\gamma_{\text{HJB}} = 1$ and varied $\lambda_{\text{Action}}$ over the same set of values:$[0, 6.25 \times 10^{-3}, 3.125 \times 10^{-2}, 6.25 \times 10^{-2}, 6.25 \times 10^{-1}, 3.125]$, with the corresponding results illustrated in Table 10. The figures indicate that as $\gamma_{\text{HJB}}$ and $\gamma_{\text{Action}}$ increase, the action of the learned trajectories decreases monotonically, demonstrating that both loss terms are effective. However, as these weights increase, the model's ability to fit the distribution deteriorates. Therefore, we recommend that in practical applications, both $\gamma_{\text{HJB}}$ and $\gamma_{\text{Action}}$ should be set to values below $0.1$, as configured in this paper.

Table 9: Comparison of $\mathcal{W}_1$ and Path Action for different values of the parameter $\gamma_{\text{HJB}}$ on a specific dataset.

| Dataset | Metric | Value of $\gamma_{\text{HJB}}$ | | | | | |
|---|---|---|---|---|---|---|---|
| | | 0 | $6.25 \times 10^{-3}$ | $3.125 \times 10^{-2}$ | $6.25 \times 10^{-2}$ | $6.25 \times 10^{-1}$ | 3.125 |
| Mouse Blood Hematopoiesis | $\mathcal{W}_1$ | 0.1573 | 0.1609 | 0.1550 | 0.1316 | 0.2109 | 0.2539 |
| | Path Action | 3.3002 | 3.2165 | 3.2397 | 3.1491 | 3.1182 | 3.1200 |
| EMT | $\mathcal{W}_1$ | 0.2702 | 0.2701 | 0.2686 | 0.2719 | 0.2722 | 0.2748 |
| | Path Action | 0.4512 | 0.4519 | 0.3867 | 0.3641 | 0.2865 | 0.2796 |

Table 10: Comparison of $\mathcal{W}_1$ and Path Action for different values of the parameter $\gamma_{\text{Action}}$ on a specific dataset.

| Dataset | Metric | Value of $\gamma_{\text{Action}}$ | | | | | |
|---|---|---|---|---|---|---|---|
| | | 0 | $6.25 \times 10^{-3}$ | $3.125 \times 10^{-2}$ | $6.25 \times 10^{-2}$ | $6.25 \times 10^{-1}$ | 3.125 |
| Mouse Blood Hematopoiesis | $\mathcal{W}_1$ | 0.1454 | 0.1344 | 0.1674 | 0.1316 | 0.4001 | 0.7914 |
| | Path Action | 3.4043 | 3.4497 | 3.2737 | 3.1491 | 2.4716 | 1.2059 |
| EMT | $\mathcal{W}_1$ | 0.2706 | 0.2681 | 0.2656 | 0.2774 | 0.3043 | 0.3832 |
| | Path Action | 0.4192 | 0.4171 | 0.3934 | 0.3288 | 0.1686 | 0.0709 |

### The Effectiveness of different types of Loss Norms

In the experiments above, we used the $L_2$ loss for computing both $\mathcal{L}_{\text{HJB}}$ and $\mathcal{L}_{\text{Action}}$. To investigate the effect of the loss function type, we present the results of using the $L_1$ loss. On the Simulation and EMT datasets, the experimental results with the $L_1$ loss are shown in Tables 11 and 12. The results indicate that with our hyperparameter values ($\gamma_{\text{HJB}} = 6.25 \times 10^{-2}, \gamma_{\text{Action}} = 6.25 \times 10^{-2}$), using either the $L_1$ loss or the $L_2$ loss when computing the HJB loss achieves similar reconstruction performance and Path Action values.

Table 11: Performance comparison of Var-RUOT with $L_2$ and $L_1$ loss functions on Simulation Dataset. The table shows $\mathcal{W}_1$ and $\mathcal{W}_2$ distances, as well as the final path action.

| | $t = 1$ | | $t = 2$ | | $t = 3$ | | $t = 4$ | | |
|---|---|---|---|---|---|---|---|---|---|
| Model | $\mathcal{W}_1$ | $\mathcal{W}_2$ | $\mathcal{W}_1$ | $\mathcal{W}_2$ | $\mathcal{W}_1$ | $\mathcal{W}_2$ | $\mathcal{W}_1$ | $\mathcal{W}_2$ | Path Action |
| Var-RUOT ($L_2$ Loss) | $0.0452_{\pm 0.0024}$ | $0.1181_{\pm 0.0064}$ | $0.0385_{\pm 0.0022}$ | $0.1270_{\pm 0.0121}$ | $0.0445_{\pm 0.0033}$ | $0.1144_{\pm 0.0160}$ | $0.0572_{\pm 0.0034}$ | $0.2140_{\pm 0.0067}$ | $1.1105_{\pm 0.0515}$ |
| Var-RUOT ($L_1$ Loss) | $0.0489_{\pm 0.0024}$ | $0.1112_{\pm 0.0135}$ | $0.0556_{\pm 0.0038}$ | $0.1541_{\pm 0.0175}$ | $0.0589_{\pm 0.0062}$ | $0.1372_{\pm 0.0226}$ | $0.0664_{\pm 0.0040}$ | $0.2181_{\pm 0.0130}$ | $1.1249_{\pm 0.0035}$ |

Table 12: Performance comparison of Var-RUOT with $L_2$ and $L_1$ loss functions on EMT Dataset for time steps $t = 1, 2, 3$.

| | $t = 1$ | | $t = 2$ | | $t = 3$ | | |
|---|---|---|---|---|---|---|---|
| Model | $\mathcal{W}_1$ | $\mathcal{W}_2$ | $\mathcal{W}_1$ | $\mathcal{W}_2$ | $\mathcal{W}_1$ | $\mathcal{W}_2$ | Path Action |
| Var-RUOT ($L_2$ Loss) | $0.2540_{\pm 0.0016}$ | $0.2623_{\pm 0.0017}$ | $0.2670_{\pm 0.0013}$ | $0.2756_{\pm 0.0014}$ | $0.2683_{\pm 0.0014}$ | $0.2796_{\pm 0.0015}$ | $0.3544_{\pm 0.0019}$ |
| Var-RUOT ($L_1$ Loss) | $0.2631_{\pm 0.0016}$ | $0.2717_{\pm 0.0014}$ | $0.2796_{\pm 0.0013}$ | $0.2886_{\pm 0.0014}$ | $0.2809_{\pm 0.0025}$ | $0.2916_{\pm 0.0025}$ | $0.3528_{\pm 0.0016}$ |

### C.3 Hold-one-out Experiment

Table 13: Performance comparison of Var-RUOT with $L_2$ and $L_1$ loss functions on 2D Mouse Blood Hematopoiesis Dataset for time steps $t = 1, 2$.

| Model | $t = 1$ | | $t = 2$ | | |
| --- | --- | --- | --- | --- | --- |
| | $\mathcal{W}_1$ | $\mathcal{W}_2$ | $\mathcal{W}_1$ | $\mathcal{W}_2$ | Path Action |
| Var-RUOT ($L_2$ Loss) | $0.1200_{\pm 0.0038}$ | $0.1459_{\pm 0.0038}$ | $0.1431_{\pm 0.0092}$ | $0.1764_{\pm 0.0135}$ | $3.1491_{\pm 0.0837}$ |
| Var-RUOT ($L_1$ Loss) | $0.1747_{\pm 0.0095}$ | $0.2097_{\pm 0.0172}$ | $0.1690_{\pm 0.0156}$ | $0.2102_{\pm 0.0242}$ | $3.1889_{\pm 0.0946}$ |

In order to validate whether our algorithm can learn the correct dynamical equations from a limited set of snapshot data, we conducted hold-one-out experiments on the three-gene simulated data, the EMT data, and the 2D Mouse Blood Hematopoiesis data. This experiment is designed to test the interpolation and extrapolation capabilities of the algorithm. For a dataset with $n$ time points, we perform $n$ experiments. In each experiment, one time point is removed from the $n$ time points, and the model is trained using the remaining time points. Afterwards, we compute the $W_1$ and $W_2$ distances between the predicted distribution and the true distribution at the missing time point. When a time point from $\{1, 2, \cdots, n - 1\}$ is removed, the model is performing interpolation; when the time point $n$ is removed, the model is performing extrapolation. The results of these experiments are shown in Table 14, Table 15, and Table 16. The experimental results indicate that our model's interpolation performance is superior to that of TIGON and comparable to that of DeepRUOT: in the EMT data and the Mouse Blood Hematopoiesis data, our model achieves the superior extrapolation performance. To further assess the long-term extrapolation capabilities of Var-RUOT and investigate whether minor errors from early time points are amplified over the longer time period, we conducted an additional extrapolation experiment. For this, we extended the three-gene simulation dataset to nine time points. The initial five time points ($t = 0, 1, 2, 3, 4$) were designated for training, while the final four ($t = 5, 6, 7, 8$) were reserved for evaluating extrapolation performance. We compared the performance of TIGON, DeepRUOT, and Var-RUOT, with the results presented in Table 17. These results indicate that Var-RUOT maintains performance superior to TIGON and comparable to DeepRUOT, even in this long-term extrapolation setting. A degradation in performance over longer extrapolation horizons is expected. This is primarily because we employ a non-autonomous system, where the parameter $\lambda$ is a function of both the state $\boldsymbol{x}$ and time $t$. Consequently, the model learns the mapping $\lambda(\boldsymbol{x}, t)$ only within the temporal domain of the training data, and its behavior beyond this range cannot be accurately inferred.

From a physical viewpoint, the dynamical equations governing the biological processes of cells can be formulated in the form of a minimum action principle (in this work, ITI RUOT Problem is a surrogate model whose action is not the true action derived from studying the biological process, but rather a simple and numerically convenient form of action). Compared to other algorithms, our method can find trajectories with lower action, i.e., It is more capable of learning dynamics that conform to the prior prescribed by the action functional. These dynamics yield better extrapolation performance, which indicates that the design of the action in the RUOT problem is at least partially reasonable. From a machine learning perspective, forcing the model to learn trajectories corresponding to the minimum action serves as a form of regularization that enhances the model's generalization capability.

In addition, we separately illustrate the learned trajectories and growth profiles on the three-gene simulated dataset after removing four different time points, as shown in Fig. 7 and Fig. 8, respectively. The consistency in the learned results indirectly demonstrates that the model is still able to learn the correct dynamics and perform effective interpolation and extrapolation, even when snapshots at certain time points are missing. We further illustrate the interpolated and extrapolated trajectories of both the DeepRUOT and Var-RUOT algorithms on the Mouse Blood Hematopoiesis dataset, as shown in Fig. 9 and Fig. 10, respectively. This dataset comprises only three time points, $t = 0, 1, 2$. When one time point is removed, Var-RUOT tends to favor a straight-line trajectory connecting the remaining two time points (since such a trajectory represents the minimum-action path), which serves as an effective prior and leads to a reasonably accurate interpolation. In contrast, because DeepRUOT does not explicitly incorporate the minimum-action objective into its model, the trajectories it learns tend to be more intricate and curved. These more complex trajectories might present challenges for generalization, making accurate interpolation or extrapolation more difficult.

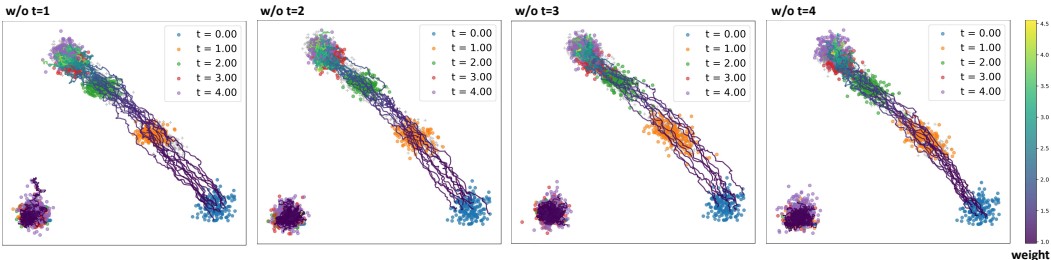

Figure 7: Trajectories learned on the three-gene simulated dataset after individually removing $t = 1, 2, 3, 4$.

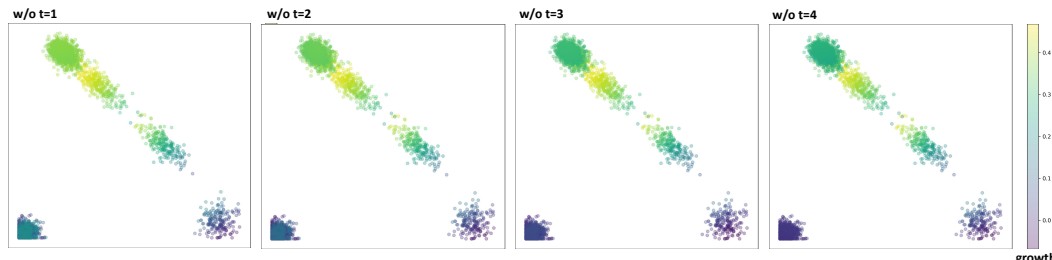

Figure 8: Growth learned on the three-gene simulated dataset after individually removing $t = 1, 2, 3, 4$.

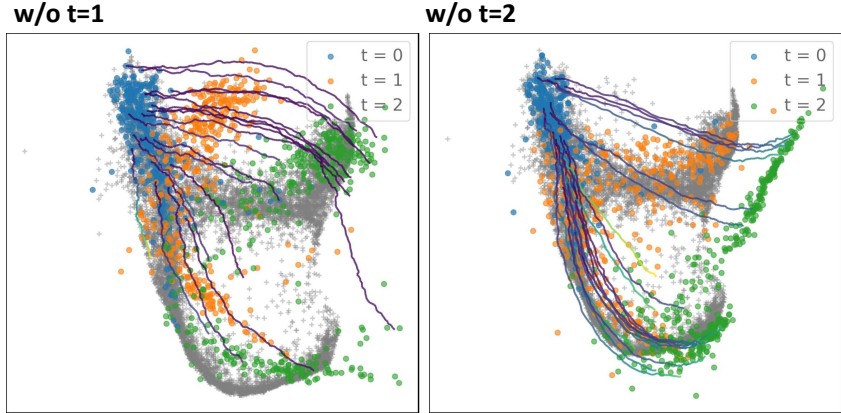

Figure 9: The results of the DeepRUOT algorithm on the 2D Mouse Blood Hematopoiesis dataset for interpolation (with t=1 removed) and extrapolation (with t=2 removed)

Table 14: On the three-gene simulated dataset, after removing the data of each time point in turn and training on the remaining data, the Wasserstein distances (i.e., $\mathcal{W}_1$ and $\mathcal{W}_2$ distances) between the model-predicted data for the missing time points and the ground truth are computed.

| Model | $t=1$ $\mathcal{W}_1$ | $\mathcal{W}_2$ | $t=2$ $\mathcal{W}_1$ | $\mathcal{W}_2$ | $t=3$ $\mathcal{W}_1$ | $\mathcal{W}_2$ | $t=4$ $\mathcal{W}_1$ | $\mathcal{W}_2$ |
|---|---|---|---|---|---|---|---|---|
| TIGON | $0.1205_{\pm0.0000}$ | $0.1679_{\pm0.0000}$ | $0.0931_{\pm0.0000}$ | $0.1919_{\pm0.0000}$ | $0.2390_{\pm0.0000}$ | $0.3369_{\pm0.0000}$ | $0.2403_{\pm0.0000}$ | $0.3616_{\pm0.0000}$ |
| RUOT | $0.0960_{\pm0.0027}$ | $0.1505_{\pm0.0018}$ | $\mathbf{0.0887}_{\pm0.0069}$ | $\mathbf{0.1501}_{\pm0.0062}$ | $0.1184_{\pm0.0058}$ | $\mathbf{0.1704}_{\pm0.0079}$ | $0.1428_{\pm0.0062}$ | $\mathbf{0.2179}_{\pm0.0135}$ |
| Var-RUOT(Ours) | $\mathbf{0.0880}_{\pm0.0036}$ | $\mathbf{0.1210}_{\pm0.0066}$ | $0.1043_{\pm0.0035}$ | $0.2293_{\pm0.0045}$ | $\mathbf{0.0943}_{\pm0.0029}$ | $0.1769_{\pm0.0092}$ | $\mathbf{0.1401}_{\pm0.0047}$ | $0.3382_{\pm0.0045}$ |

Table 15: On the EMT dataset, after removing the data of each time point in turn and training on the remaining data, the Wasserstein distances (i.e., $\mathcal{W}_1$ and $\mathcal{W}_2$ distances) between the model-predicted data for the missing time points and the ground truth are computed.

| Model | $t=1$ $\mathcal{W}_1$ | $\mathcal{W}_2$ | $t=2$ $\mathcal{W}_1$ | $\mathcal{W}_2$ | $t=3$ $\mathcal{W}_1$ | $\mathcal{W}_2$ |
|---|---|---|---|---|---|---|
| TIGON | $0.3457_{\pm0.0000}$ | $0.3560_{\pm0.0000}$ | $0.3733_{\pm0.0000}$ | $0.3849_{\pm0.0000}$ | $0.5260_{\pm0.0000}$ | $0.5424_{\pm0.0000}$ |
| RUOT | $0.3107_{\pm0.0017}$ | $0.3201_{\pm0.0016}$ | $\mathbf{0.3344}_{\pm0.0024}$ | $\mathbf{0.3445}_{\pm0.0021}$ | $0.4947_{\pm0.0019}$ | $0.5074_{\pm0.0019}$ |
| Var-RUOT(Ours) | $\mathbf{0.3018}_{\pm0.0030}$ | $\mathbf{0.3104}_{\pm0.0031}$ | $0.3375_{\pm0.0027}$ | $0.3460_{\pm0.0028}$ | $\mathbf{0.4082}_{\pm0.0027}$ | $\mathbf{0.4189}_{\pm0.0027}$ |

Table 16: On the Mouse Blood Hematopoiesis dataset, after removing the data of each time point in turn and training on the remaining data, the Wasserstein distances (i.e., $\mathcal{W}_1$ and $\mathcal{W}_2$ distances) between the model-predicted data for the missing time points and the ground truth are computed.

| Model | $t=1$ $\mathcal{W}_1$ | $\mathcal{W}_2$ | $t=2$ $\mathcal{W}_1$ | $\mathcal{W}_2$ |
|---|---|---|---|---|
| TIGON | $0.5838_{\pm0.0000}$ | $0.6726_{\pm0.0000}$ | $1.3264_{\pm0.0000}$ | $1.3928_{\pm0.0000}$ |
| RUOT | $0.6235_{\pm0.0014}$ | $0.6971_{\pm0.0012}$ | $1.0723_{\pm0.0096}$ | $1.1397_{\pm0.0120}$ |
| Var-RUOT(Ours) | $\mathbf{0.2696}_{\pm0.0054}$ | $\mathbf{0.3279}_{\pm0.0044}$ | $\mathbf{0.2594}_{\pm0.0069}$ | $\mathbf{0.3016}_{\pm0.0095}$ |

Table 17: Long-term extrapolation performance comparison on the three-gene simulation dataset. All models were trained on three-gene simulation data from the first five time points ($t=0,1,2,3,4$) and evaluated on four subsequent, unseen time points ($t=5,6,7,8$). The table reports performance metrics (e.g., $\mathcal{W}_1$ and $\mathcal{W}_2$ distances) against the ground truth.

| Method | $t=5$ $\mathcal{W}_1$ | $\mathcal{W}_2$ | $t=6$ $\mathcal{W}_1$ | $\mathcal{W}_2$ | $t=7$ $\mathcal{W}_1$ | $\mathcal{W}_2$ | $t=8$ $\mathcal{W}_1$ | $\mathcal{W}_2$ |
|---|---|---|---|---|---|---|---|---|
| TIGON | $0.1932_{\pm0.0}$ | $0.2496_{\pm0.0}$ | $0.3437_{\pm0.0}$ | $0.4011_{\pm0.0}$ | $0.5209_{\pm0.0}$ | $0.5643_{\pm0.0}$ | $0.6913_{\pm0.0}$ | $0.7155_{\pm0.0}$ |
| DeepRUOT | $0.1027_{\pm0.0048}$ | $0.1459_{\pm0.0010}$ | $0.1940_{\pm0.0069}$ | $0.2192_{\pm0.0055}$ | $0.3359_{\pm0.0088}$ | $0.3644_{\pm0.0074}$ | $0.5047_{\pm0.0076}$ | $0.5362_{\pm0.0059}$ |
| VarRUOT | $0.1057_{\pm0.0175}$ | $0.2191_{\pm0.0067}$ | $0.1806_{\pm0.0223}$ | $0.2604_{\pm0.0073}$ | $0.2890_{\pm0.0032}$ | $0.3787_{\pm0.0083}$ | $0.4260_{\pm0.0032}$ | $0.5557_{\pm0.0067}$ |

## C.4 Experiments on High Dimensional Dataset

**High Dimensional Gaussian Dataset** To evaluate the effectiveness of our method on high-dimensional datasets, we first tested it on Gaussian datasets of 50 and 100 dimensions. We learned the dynamics of the data using the standard WFR metric ($\psi(g) = \frac{1}{2}g^2$) as well as a modified growth penalty function, $\psi(g) = g^{2/15}$, which ensures $\psi''(g) < 0$. The learned trajectories and growth rates are illustrated in Fig. 11. Under both choices of $\psi(g)$, our method captures reasonable dynamics: the Gaussian distribution centered on the left shifts upward and downward, while the Gaussian distribution on the right exhibits growth without displacement. We also tested the performance of VarRUOT and other algorithms on a 150-dimensional Gaussian Mixture dataset, as shown in Table 18.

**50D Mouse Blood Hematopoiesis and Pancreatic $\beta$-cell Differentiation Dataset** We tested our method on two high-dimensional real scRNA-seq datasets including 50D Mouse Blood Hematopoiesis dataset and Pancreatic $\beta$-cell Differentiation dataset. We used UMAP to reduce the dimensionality of the datasets to 2 (only for visualization) , plotted the growth of each data point, and visualized the vector fields $\mathbf{u}(\mathbf{x}, t)$ on the reduced dimensions using the scvelo library. The results for the two datasets are shown in Fig. 12 and Fig. 13, respectively. As can be seen from the figures, the reduced

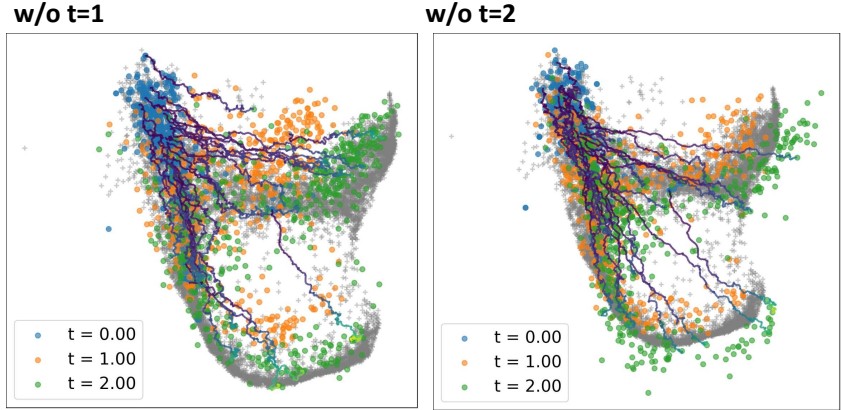

Figure 10: The results of the Var-RUOT algorithm on the 2D Mouse Blood Hematopoiesis dataset for interpolation (with t=1 removed) and extrapolation (with t=2 removed)

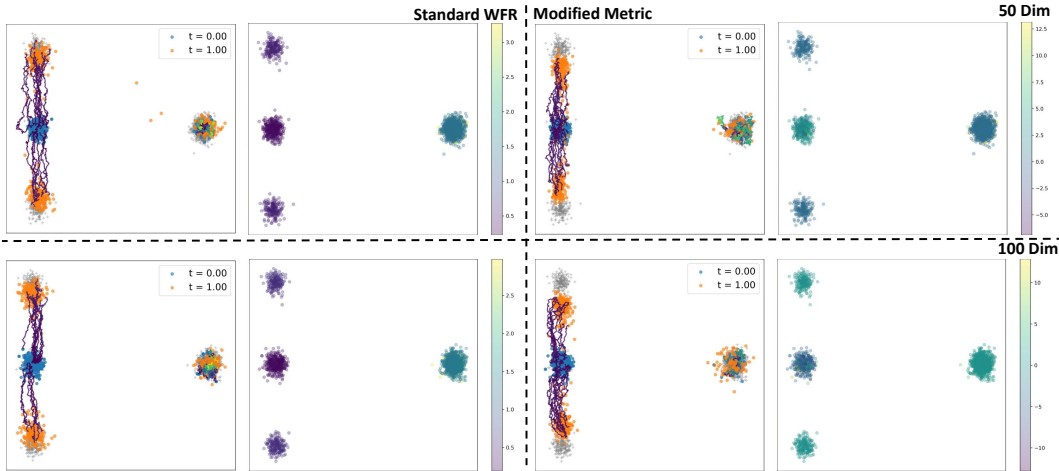

Figure 11: Trajectories and growth learned on the 50-dimensional and 100-dimensional Gaussian datasets using the Standard WFR Metric and Modified Metric.

Table 18: Wasserstein distances ($\mathcal{W}_1$ and $\mathcal{W}_2$) between the predicted distributions of each algorithm and the true distribution at various time points on 150D Gaussian Data. Each experiment was run five times to compute the mean and standard deviation.

| Model | $t = 1$ | | $t = 2$ | | Path Action |
|---|---|---|---|---|---|
| | $\mathcal{W}_1$ | $\mathcal{W}_2$ | $\mathcal{W}_1$ | $\mathcal{W}_2$ | |
| SF2M | $7.286_{\pm 0.002}$ | $7.345_{\pm 0.002}$ | $9.457_{\pm 0.001}$ | $9.558_{\pm 0.002}$ | |
| PISDE | $\mathbf{6.051}_{\pm 0.002}$ | $\mathbf{6.061}_{\pm 0.002}$ | $7.546_{\pm 0.001}$ | $7.597_{\pm 0.001}$ | |
| MIO Flow | $6.494_{\pm 0.000}$ | $6.521_{\pm 0.000}$ | $7.812_{\pm 0.000}$ | $7.866_{\pm 0.000}$ | |
| TIGON | $6.157_{\pm 0.000}$ | $6.169_{\pm 0.000}$ | $7.615_{\pm 0.000}$ | $7.665_{\pm 0.000}$ | 13.4913 |
| DeepRUOT | $6.202_{\pm 0.001}$ | $6.213_{\pm 0.001}$ | $7.648_{\pm 0.003}$ | $7.696_{\pm 0.003}$ | 13.5992 |
| Var-RUOT | $6.072_{\pm 0.008}$ | $6.089_{\pm 0.009}$ | $\mathbf{7.544}_{\pm 0.005}$ | $\mathbf{7.596}_{\pm 0.005}$ | $10.1033_{\pm 0.5677}$ |

velocity field points from points with smaller $t$ to those with larger $t$, which indicates that our model can correctly learn a vector field that transfers the distribution even in high-dimensional data.

**EB Dataset and NeurIPS Challenge Dataset**

To further quantitatively evaluate the performance of VarRUOT on high-dimensional data, we employed the 100-dimensional EB dataset and the 50-dimensional NeurIPS Challenge dataset. The

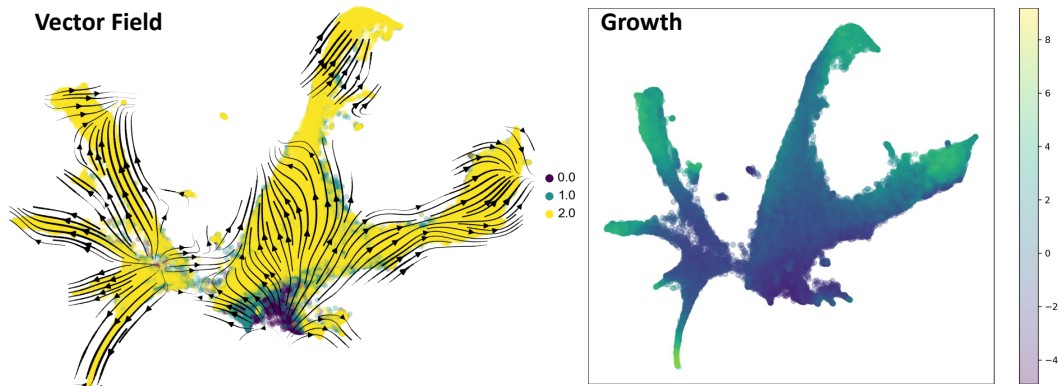

Figure 12: Learned vector field $\mathbf{u}(\mathbf{x}, t)$ and growth on the 50D Mouse Blood Hematopoiesis dataset (data reduced using UMAP and vector field stream plots generated with the scvelo library).

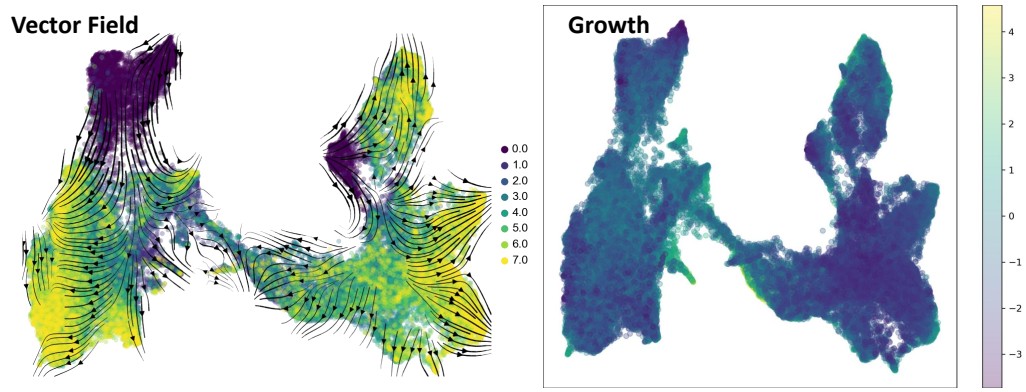

Figure 13: Learned vector field $\mathbf{u}(\mathbf{x}, t)$ and growth on the Pancreatic $\beta$-cell Differentiation dataset (data reduced using UMAP and vector field stream plots generated with the scvelo library).

quantitative results for VarRUOT against other algorithms are presented in Table 19 and Table 20, respectively. On the EB dataset, Var-RUOT demonstrates distribution reconstruction performance comparable to that of TIGON and DeepRUOT, while surpassing simulation-free approaches (OTCFM, UOTCFM) and Wasserstein Gradient Flow. For the NeurIPS Challenge dataset, Var-RUOT outperforms all other prior methods evaluated. We also report the results for VarRUOT and competing algorithms on the unnormalized 5-dimensional EB dataset in Table 21.

Table 19: Wasserstein distances ($\mathcal{W}_1$ and $\mathcal{W}_2$) between the predicted distributions of each algorithm and the true distribution at various time points on EB Data. Each experiment was run five times to compute the mean and standard deviation.

| Method | $t_1$ | | $t_2$ | | $t_3$ | | $t_4$ | |
|---|---|---|---|---|---|---|---|---|
| | $\mathcal{W}_1$ | $\mathcal{W}_2$ | $\mathcal{W}_1$ | $\mathcal{W}_2$ | $\mathcal{W}_1$ | $\mathcal{W}_2$ | $\mathcal{W}_1$ | $\mathcal{W}_2$ |
| SF2M | $9.6732_{\pm 0.0006}$ | $9.7446_{\pm 0.0005}$ | $11.0711_{\pm 0.0006}$ | $11.1649_{\pm 0.0024}$ | $12.5436_{\pm 0.0041}$ | $12.6476_{\pm 0.0047}$ | $14.7100_{\pm 0.0077}$ | $14.8127_{\pm 0.0006}$ |
| PISDE | $8.8317_{\pm 0.0007}$ | $8.9075_{\pm 0.0008}$ | $9.4283_{\pm 0.0015}$ | $9.5312_{\pm 0.0016}$ | $9.7872_{\pm 0.0013}$ | $9.8952_{\pm 0.0016}$ | $11.1102_{\pm 0.0018}$ | $11.2071_{\pm 0.0022}$ |
| MIOFLOW | $8.7182_{\pm 0.0000}$ | $8.7924_{\pm 0.0000}$ | $9.4139_{\pm 0.0000}$ | $9.5180_{\pm 0.0000}$ | $9.7547_{\pm 0.0000}$ | $9.8671_{\pm 0.0000}$ | $11.0080_{\pm 0.0000}$ | $11.1065_{\pm 0.0000}$ |
| TIGON | $8.7992_{\pm 0.0000}$ | $8.8769_{\pm 0.0000}$ | $9.6497_{\pm 0.0000}$ | $9.7533_{\pm 0.0000}$ | $10.0130_{\pm 0.0000}$ | $10.1209_{\pm 0.0000}$ | $11.3284_{\pm 0.0000}$ | $11.2452_{\pm 0.0000}$ |
| RUOT | $8.8029_{\pm 0.0023}$ | $8.8803_{\pm 0.0022}$ | $9.6518_{\pm 0.0071}$ | $9.7555_{\pm 0.0065}$ | $10.0365_{\pm 0.0126}$ | $10.1424_{\pm 0.0118}$ | $11.3555_{\pm 0.0133}$ | $11.4501_{\pm 0.0132}$ |
| OTCFM | $9.4459_{\pm 0.0000}$ | $9.5280_{\pm 0.0000}$ | $11.3786_{\pm 0.0000}$ | $11.5893_{\pm 0.0000}$ | $14.0529_{\pm 0.0000}$ | $14.9498_{\pm 0.0000}$ | $19.7489_{\pm 0.0000}$ | $19.7489_{\pm 0.0000}$ |
| UOTCFM | $9.4794_{\pm 0.0000}$ | $9.5596_{\pm 0.0000}$ | $11.6060_{\pm 0.0000}$ | $11.8336_{\pm 0.0000}$ | $14.6715_{\pm 0.0000}$ | $15.6972_{\pm 0.0000}$ | $21.7653_{\pm 0.0000}$ | $25.5683_{\pm 0.0000}$ |
| Action Matching | $12.7145_{\pm 0.0000}$ | $12.9813_{\pm 0.0000}$ | $16.7809_{\pm 0.0000}$ | $16.8206_{\pm 0.0000}$ | $13.4663_{\pm 0.0000}$ | $13.7058_{\pm 0.0000}$ | $18.1165_{\pm 0.0000}$ | $18.1657_{\pm 0.0000}$ |
| WGF | $9.9847_{\pm 0.0000}$ | $10.0482_{\pm 0.0000}$ | $11.0562_{\pm 0.0000}$ | $11.1344_{\pm 0.0000}$ | $11.9719_{\pm 0.0000}$ | $12.0508_{\pm 0.0000}$ | $13.2417_{\pm 0.0000}$ | $13.3035_{\pm 0.0000}$ |
| Var-RUOT | $8.9158_{\pm 0.0026}$ | $8.9996_{\pm 0.0006}$ | $9.7955_{\pm 0.0203}$ | $9.9651_{\pm 0.0245}$ | $10.2647_{\pm 0.0214}$ | $10.4849_{\pm 0.0316}$ | $11.4927_{\pm 0.0585}$ | $11.6323_{\pm 0.0629}$ |

Table 20: Wasserstein distances ($\mathcal{W}_1$ and $\mathcal{W}_2$) between the predicted distributions of each algorithm and the true distribution at various time points on NeurIPS Challenge Data. Each experiment was run five times to compute the mean and standard deviation.

| Method | $t_1$ | | $t_2$ | | $t_3$ | | $t_4$ | |
|---|---|---|---|---|---|---|---|---|
| | $\mathcal{W}_1$ | $\mathcal{W}_2$ | $\mathcal{W}_1$ | $\mathcal{W}_2$ | $\mathcal{W}_1$ | $\mathcal{W}_2$ | $\mathcal{W}_1$ | $\mathcal{W}_2$ |
| SF2M | $6.1876_{\pm0.0085}$ | $6.2565_{\pm0.0081}$ | $12.6170_{\pm0.0602}$ | $13.6426_{\pm0.0707}$ | $9.1830_{\pm0.0041}$ | $10.9930_{\pm0.1741}$ | $12.5831_{\pm0.6213}$ | $13.2691_{\pm0.4054}$ |
| PISDE | $5.6642_{\pm0.0078}$ | $5.7069_{\pm0.0079}$ | $5.2358_{\pm0.0073}$ | $5.2779_{\pm0.0077}$ | $6.2793_{\pm0.0134}$ | $6.4015_{\pm0.0134}$ | $8.9086_{\pm0.0053}$ | $9.0298_{\pm0.0060}$ |
| MIOFLOW | $6.4267_{\pm0.0000}$ | $6.4817_{\pm0.0000}$ | $6.2715_{\pm0.0000}$ | $6.3141_{\pm0.0000}$ | $7.1079_{\pm0.0000}$ | $7.2222_{\pm0.0000}$ | $8.9035_{\pm0.0000}$ | $9.0160_{\pm0.0000}$ |
| TIGON | $5.9481_{\pm0.0000}$ | $6.0003_{\pm0.0000}$ | $5.7599_{\pm0.0000}$ | $5.8008_{\pm0.0000}$ | $6.5539_{\pm0.0000}$ | $6.6787_{\pm0.0000}$ | $8.9573_{\pm0.0000}$ | $9.0587_{\pm0.0000}$ |
| RUOT | $6.0166_{\pm0.0020}$ | $6.0579_{\pm0.0018}$ | $5.6972_{\pm0.0048}$ | $5.7415_{\pm0.0050}$ | $6.5099_{\pm0.0034}$ | $6.6419_{\pm0.0062}$ | $8.8047_{\pm0.0069}$ | $8.9103_{\pm0.0069}$ |
| OTCFM | $5.5082_{\pm0.0000}$ | $5.5645_{\pm0.0000}$ | $5.4476_{\pm0.0000}$ | $5.5151_{\pm0.0000}$ | $7.0935_{\pm0.0000}$ | $7.3473_{\pm0.0000}$ | $10.1416_{\pm0.0000}$ | $10.7654_{\pm0.0000}$ |
| UOTCFM | $5.5526_{\pm0.0000}$ | $5.6089_{\pm0.0000}$ | $5.4349_{\pm0.0000}$ | $5.5086_{\pm0.0000}$ | $6.7482_{\pm0.0000}$ | $6.6849_{\pm0.0000}$ | $9.6782_{\pm0.0000}$ | $10.3230_{\pm0.0000}$ |
| Action Matching | $7.3701_{\pm0.0000}$ | $7.4312_{\pm0.0000}$ | $8.5156_{\pm0.0000}$ | $8.5485_{\pm0.0000}$ | $9.2842_{\pm0.0000}$ | $9.3269_{\pm0.0000}$ | $9.9665_{\pm0.0000}$ | $10.0428_{\pm0.0000}$ |
| WGF | $7.1785_{\pm0.0000}$ | $7.2411_{\pm0.0000}$ | $7.7135_{\pm0.0000}$ | $7.7479_{\pm0.0000}$ | $8.2521_{\pm0.0000}$ | $8.3044_{\pm0.0000}$ | $8.6629_{\pm0.0000}$ | $8.7749_{\pm0.0000}$ |
| Var-RUOT | $4.7961_{\pm0.0763}$ | $4.9015_{\pm0.0844}$ | $4.1816_{\pm0.0145}$ | $4.2887_{\pm0.0151}$ | $6.2312_{\pm0.0130}$ | $6.4587_{\pm0.0144}$ | $9.1058_{\pm0.0511}$ | $9.3636_{\pm0.0537}$ |

Table 21: Comparison of Wasserstein-1 distances ($\mathcal{W}_1$) between the predicted distributions and the ground truth on the unnormalized 5-dimensional EB dataset.

| | Wasserstein-1 Distance ($\mathcal{W}_1$) at time $t$ | | | |
|---|---|---|---|---|
| Model | $t=1$ | $t=2$ | $t=3$ | $t=4$ |
| MMFM | 0.477 | 0.554 | 0.781 | 0.872 |
| Metric FM | 0.449 | 0.552 | 0.583 | 0.597 |
| SF2M | 0.556 | 0.715 | 0.750 | 0.650 |
| MIOFlow | 0.442 | 0.585 | 0.651 | 0.670 |
| TIGON | 0.386 | 0.502 | 0.602 | 0.600 |
| DeepRUOT | 0.386 | 0.497 | 0.591 | 0.585 |
| UOT-FM | 0.544 | 0.670 | 0.729 | 0.852 |
| VARRUOT | 0.416 | 0.486 | 0.509 | 0.511 |

## C.5 Comparison with Tranditioal OT Solvers

To verify whether the Var-RUOT algorithm can find the correct minimum action paths for both Optimal Transport (OT) and Unbalanced Optimal Transport (UOT) problems, we constructed two normalized (balanced) and two unbalanced Gaussian Mixture datasets on a 2D plane. We then compared the action computed by the Var-RUOT method (with $\sigma = 0$ to eliminate stochasticity) against the minimum action values calculated by traditional solvers.

To compute the ground truth for the minimum action on the normalized datasets, we used the pot library. For the unbalanced datasets, we could not find an open-source solver for dynamic unbalanced optimal transport (apart from deep learning-based methods like TIGON ). Therefore, we adopted a static Wasserstein-Fisher-Rao (WFR) distance solver, designed as a variant of the Sinkhorn algorithm as described in (Wang et al. 2019), to serve as our baseline. The comparison results are presented in Table 22.

Table 22: Comparison of Path Action values on 2D Gaussian Mixture datasets.

| Dataset | Normalized | | Unbalanced | |
|---|---|---|---|---|
| | Single Gaussian | Gaussian Mixture | Single Gaussian | Gaussian Mixture |
| Path Action (Traditional Solver) | 0.5046 | 1.9062 | 1.3776 | 2.4464 |
| Path Action (Var-RUOT) | 0.4998 | 1.8754 | 1.0102 | 2.3909 |

The results show that for the two normalized datasets, Var-RUOT finds Path Action values comparable to those computed by the pot library. On the two unbalanced datasets, the Path Action found by Var-RUOT is even lower than that of the traditional solver. This outcome might be attributable to either (1) suboptimal solution of the baseline solver or (2) numerical roundoff errors. Nevertheless, our review of the literature did not identify alternative open-source baselines addressing this specific problem. Overall, these results demonstrate that Var-RUOT can achieve performance comparable to traditional OT solvers or WFR distance solvers for the task of finding the minimum action path, which validates its effectiveness.

# D    Further Discussions

## D.1    Comparison with Relevant Algorithms

Several methods have been proposed to address the problem of trajectory inference for temporal snapshots of single-cell data. We list the differences between our method and previous trajectory inference algorithms in Table 23.

Table 23: Comparison of trajectory inference methods. Properties assessed are: handling **Unbalanced Distributions** (unequal mass); modeling **Stochastic Dynamics**; addressing both **Unbalanced + Stochastic** properties simultaneously; and formulation based on the **Least Action** principle via first-order conditions, not just as a loss penalty. The '$+$' indicates the method possesses the property, while a '$-$' indicates it does not.

| Method | Unbalanced Distribution | Stochastic Dynamics | Unbalanced + Stochastic | Least Action |
|---|---|---|---|---|
| SF2M | $-$ | $+$ | $-$ | $-$ |
| PISDE | $-$ | $+$ | $-$ | $+$ |
| MIOFlow | $-$ | $-$ | $-$ | $-$ |
| Action Matching | $+$ | $+$ | $-$ | $+$ |
| TIGON | $+$ | $-$ | $-$ | $-$ |
| DeepRUOT | $+$ | $+$ | $+$ | $-$ |
| Var-RUOT (Ours) | $+$ | $+$ | $+$ | $+$ |

### Difference from DeepRUOT

Like DeepRUOT, Var-RUOT addresses the critical challenge of handling both unbalanced distributions and stochastic dynamics, which is essential for single-cell omics data.

The primary difference is architectural. DeepRUOT employs three interdependent neural networks to model the velocity field $v(x, t)$, growth rate $g(x, t)$, and log-density $\log \rho(x, t)$. It enforces physical constraints as soft penalties in the loss function. This complex setup necessitates a multi-stage pre-training procedure to ensure convergence.

In contrast, Var-RUOT uses a single network to parameterize a scalar field $\lambda_\theta(x, t)$. Both the velocity $u(x, t)$ and growth rate $g(x, t)$ are analytically derived from $\lambda_\theta(x, t)$ via first-order optimality conditions (Theorem 4.1). This simpler, end-to-end design eliminates the need for pre-training and improves training stability.

### Difference from Action Matching

The differences between Var-RUOT and Action Matching could be summarized from three perspectives. Firstly, Var-RUOT handles both unbalanced distributions and stochastic dynamics, which is crucial for single-cell omics.

Secondly, Action Matching's training requires sampling from intermediate distributions $q_t(x)$. These are typically challenging to obtain in single-cell omics, where data exists only as sparse time-point snapshots. This data limitation can potentially hinder the ability to capture complex dynamics, whereas Var-RUOT is designed for such snapshot data, and empirical results suggest Var-RUOT's desirable performance.

Lastly, the methods employ different loss formulations. Action Matching minimizes an Action Gap loss:

$$\mathcal{L}_{\mathrm{AM}} = \mathbb{E}_{q_0(x)}[s_0(x)] - \mathbb{E}_{q_1(x)}[s_1(x)] + \int_0^1 \mathbb{E}_{q_t(x)} \left[ \frac{1}{2} \|\nabla s_t(x)\|^2 + \frac{\partial s}{\partial t}(x) \right] \mathrm{d}t.$$

On the other hand, Var-RUOT directly minimizes the residual of the Hamilton-Jacobi-Bellman (HJB) equation. By directly enforcing this first-order optimality condition in its loss, Var-RUOT can find solutions with low action robustly in empirical tests.

## D.2    Convergence of the Particle Method and Loss Computation

The VarRUOT algorithm involves three loss terms: $\mathcal{L}_{\mathrm{HJB}}$, $\mathcal{L}_{\mathrm{Action}}$, and $\mathcal{L}_{\mathrm{Recon}}$. Among these, $\mathcal{L}_{\mathrm{HJB}}$ and $\mathcal{L}_{\mathrm{Action}}$ are computed using the Monte Carlo method. This involves sampling a set of particles,

integrating along their trajectories, and then averaging the results over all particles. The convergence rate of the Monte Carlo method is $\mathcal{O}\left(\frac{1}{\sqrt{N}}\right)$, where $N$ is the number of particles used.

The reconstruction loss, $\mathcal{L}_{\text{Recon}}$, is composed of two parts: $\mathcal{L}_{\text{OT}}$ and $\mathcal{L}_{\text{Mass}}$. $\mathcal{L}_{\text{OT}}$, calculated using the `geomloss` library, represents the $\mathcal{W}_2$ distance between the normalized empirical distribution and the true distribution. According to the triangle inequality, $\mathcal{W}_2(\mu^N, \nu) \leq \mathcal{W}_2(\mu^N, \mu) + \mathcal{W}_2(\mu, \nu)$, the convergence of $\mathcal{L}_{\text{OT}}$ as the number of particles increases depends on the rate at which the empirical measure $\mu^N = \frac{1}{N}\sum_{i=1}^{N}\delta(\boldsymbol{x} - \boldsymbol{x}_i)$ (with $\boldsymbol{x}_i \sim \mu$) approximates the true measure $\mu$. The expected error is bounded as follows (Fournier and Guillin 2013):

$$
\mathbb{E}[\mathcal{W}_2(\mu^N, \mu)] \leq C \begin{cases} N^{-\frac{1}{2}} & \text{if } d = 2,3 \\ N^{-\frac{1}{2}}\log(1+N) & \text{if } d = 4 \\ N^{-\frac{2}{d}} & \text{if } d > 4 \end{cases}
$$

# E   Limitations and Broader Impacts

## E.1   Limitations

The algorithm presented in this paper offers new insights for solving the RUOT problem; however, it still has several limitations. Firstly, although Var-RUOT parameterizes $\mathbf{u}$ and $g$ with a single neural network, and designs the loss function based on the necessary conditions for the minimal action solution, since neural network optimization only finds local minima, there is still no guarantee that the solution found is indeed the one with minimal action. Moreover, in practice, the HJB loss does not necessarily converge to zero (A typical training curve for the HJB loss is shown in Fig. 14). Thus, it effectively acts as a regularization term. This could be addressed by conducting a more detailed analysis on simpler versions of the RUOT problem (for instance, Gaussian-to-Gaussian transport or Dirac-to-Dirac Transport).

Furthermore, when using the modified metric, the goodness-of-fit in the distribution deteriorates, which may suggest that the $\mathbf{u}$ and $g$ satisfying the optimal necessary conditions derived via the variational method are limited in transporting the initial distribution to the terminal distribution. This might reflect a controllability issue in control theory that warrants further investigation.

Finally, the choice of $\psi(g)$ in the action is dependent on biological priors. To automate it, one could approximate $\psi$ with a neural network or derive it from microscopic or mesoscopic dynamics, such as the branching Wiener process to model cell division for a more physically grounded action (Baradat and Lavenant 2021).

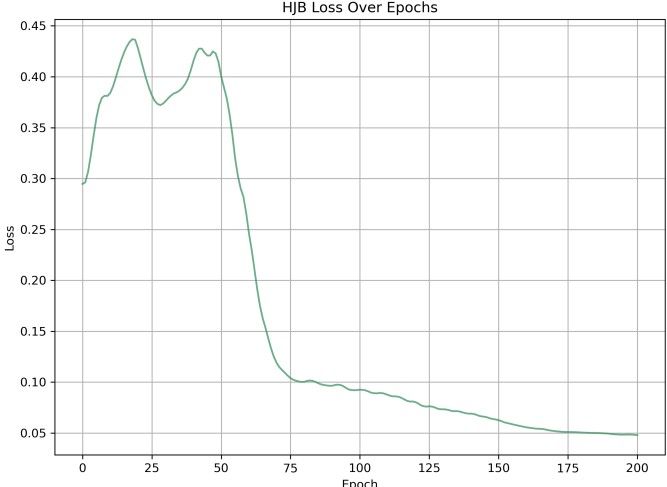

Figure 14: Training curve of HJB Loss on three-gene simulation dataset.

### E.2 Broader Impacts

Var-RUOT explicitly incorporates the first-order optimality conditions of the RUOT problem into both the parameterization process and the loss function. This approach enables our algorithm to find solutions with a smaller action while maintaining excellent distribution fitting accuracy. Compared to previous methods, Var-RUOT employs only a single network to approximate a scalar field, which results in a faster and more stable training process. Additionally, we observe that the selection of the growth penalty function $\psi(g)$ within the WFR metric is highly correlated with the underlying biological priors. Consequently, our new algorithm provides a novel perspective on the RUOT problem.

Our approach can be extended to other analogous systems. For example, in the case of simple mesoscopic particle systems where the action can be explicitly formulated such as in diffusion or chemical reaction processes, our framework can effectively infer the evolution of particle trajectories and distributions. This capability makes it applicable to tasks such as experimental data processing and interpolation. In the biological or medical field, our method can be employed to predict cellular fate and to provide quantitative diagnostic results or treatment plans for certain diseases.

It should be noted that the performance of Var-RUOT largely depends on the quality of the data. Datasets containing significant noise may lead the model to produce results with a slight bias. Moreover, the particular form of the action can have a substantial impact on the model's outcomes, potentially affecting important biological priors. These factors could present challenges for subsequent biological analyses or clinical decision-making, and care must be taken in the use and dissemination of the model-generated interpolation results to avoid data contamination.

When applying our method in biological or medical contexts, it is crucial to train the model using high-quality experimental data, select an action formulation that is well-aligned with the relevant domain-specific priors, and ensure that the results are validated by domain experts. Furthermore, there is a need to enhance the interpretability of the model and to further improve training speed through methods such as simulation-free techniques. These directions represent important avenues for our future work.

