# OpenReview forum: "Variational Regularized Unbalanced Optimal Transport: Single Network, Least Action"
_NeurIPS.cc/2025/Conference — NeurIPS 2025 poster_

### Official Review · Reviewer_YDV3 · 2025-06-22

**Clarity:** 3
**Significance:** 2
**Originality:** 2
**Rating:** 4
**Confidence:** 4

**Summary:**

The authors of the paper introduce Var-RUOT, a method to perform deep regularised unbalanced optimal transport. Var-RUOT parameterizes a scalar potential and derives the necessary conditions required by the associated field and growth penalty to minimise the isotropic time invariant RUOT problem. The potential is trained by minimising a loss function computed over simulated particles from an empirical measure. The loss has multiple components. First, a reconstruction term ensures that the model recapitulates the marginal distributions over time and preserves the total mass of individual snapshots. The second component is a Hamilton-Jacobi-Beltrami loss enforcing field optimality under the ITI-RUOT problem.
Furthermore, the least action loss is added to regularize the dynamic optimal transport field. Finally, the paper proposes a new biological interpretation of the growth penalty function, showing that it varies in the direction of the transporting field. By controlling the second derivative of the penalty function, one can encode biological processes, such as the loss of stemness. In their experiments, the authors compare their model with existing methods on the task of reconstructing time marginals of real and simulated single-cell data. Together with distribution matching metrics the paper additionally shows that the model achieves lower path action and faster convergence compared to its counterparts, offering an efficient approach to study cellular development in cellular biology.

**Questions:**

* Why are the weights initialised as 1? Is there a way to instead learn them?
* What is the ground truth for the mass objective? In other words, how do you compute M(Tk) if you only have the weights for the solution of the ODE?
* Why did the authors pick $\frac{2}{15}$ as an exponent of $\phi_2$? How does the penalty function behave for different values of the exponent?
* **Line 71**: isn’t the Schrödinger bridge always dynamic as it operates via regularisation to a reference probability path?

**Ethical Concerns:**

["NO or VERY MINOR ethics concerns only"]

**Final Justification:**

Dear AC,

Thank you very much for handling this paper.

In my final assessment, I increased the score assigned to the present submission from 3 to 4. The authors have addressed most of my concerns with substantial experiments, explanations, and analyses. I believe the paper is above the acceptance bar, as it introduces new theory while providing value for downstream applications.

The reason I did not score a 5 is that I believe some aspects are not fully polished, such as the exploration of the relevance of the new formulation of the non-canonical growth penalty and the lack of presentation quality in the qualitative results.

I remain available to provide further justification for my score.

Best regards,

The Reviewer

**Limitations:**

The limitations of the work have been sufficiently addressed in the Appendix.

**Paper Formatting Concerns:**

I did not come across any formatting constraints during my review.

**Quality:**

3

**Strengths And Weaknesses:**

## Strengths

I generally find the paper well-written, for which I commend the authors. The math is introduced gradually and is easy to digest, which is a very positive aspect in my opinion. I also think that the problem of enforcing stronger optimality constraints to dynamical OT is relevant and well-justified, specifically in the unbalanced setting, which is very useful in computational biology. I particularly enjoyed the growth penalty explanation part, as a better interpretation of its analytical form could provide end users with no expertise in OT with a biological rationale on how to set its value to optimally model the system they are considering.

## Weaknesses

At the moment, there are some concerns on my side regarding the presentation of the paper that induced me to a borderline rejection score. I would like to know more about the authors' opinions during the rebuttal phase to be able to produce a final informed assessment.

* **Selection of the growth penalty:** Though I appreciate the formalisation of the growth penalty in biological terms, the results proposed by the authors show that using more sophisticated terms than the WFR standard approach is suboptimal on the snapshot reconstruction task. Therefore, I am not sure this specific aspect of the paper is ripe enough to be used as an effective contribution to the paper. I still think this aspect is interesting, and maybe additional options for the penalty functions could be considered in further iterations of the paper, or at least be shown in a setting where it is actually useful.
* **Line 103** Here, I recommend that the authors define $\alpha$ with a short sentence.
* **Time index:** Overall, I would personally find it easier to understand if all the time-resolved variables and quantities in the paper had an index indicating time. Some of them already have this (for example, the density $\rho$), but some others miss it (as, for instance, the empirical measure $\mu$ or the weights $w$).
* **Benchmark:** One thing that I find sub-optimal in the current presentation is the way benchmarks are conducted. First, some models are presented in Tab. 1 and 2, while many of them are dropped later in Tab. 4. I personally believe that the baselines should be the same throughout to ensure transparency.
* **Missing baselines:** Moreover, nowadays, there are very powerful OT-based methods for trajectory inference in low dimensions, like OT-CFM and derivatives thereof. I think I would have given some priority to them as baselines to test whether stochasticity and unbalancedness help in trajectory reconstruction. Also, combinations of OT Flow Matching and unbalancedness were tackled in some existing works [1], which I think is worth comparing against. Along the same line, comparison with [2], which is presented as a special case of the current model, is beneficial. What I am trying to say is that in a research setting where simulation-based methods are overtaken by their simulation-free counterpart, comparison with the latter models is meaningful to show that the proposed approach is a valuable option.
* **Dataset choice.** There are widespread benchmarks of methods used to match temporal snapshots in scRNA-seq. They are based on the Embryoid body dataset from [3] and the NeurIPS benchmark in [4]. They also test slightly higher dimensionalities of the input space, which I think is relevant here in a simulation-based setting.
* **Figure presentation.** In my opinion, Fig. 3 is not very clear. I can see what the colors mean, but the paths are not clear to me. I think the presentation could be improved by making the source and the target more understandable. For example, I don't understand where one can see higher curvature (line 256) in the results from DeepRUOT. Moreover, I would add a legend explaining the difference between crosses and full dots. Moreover, in Fig. 2, Var-RUOT produces different growth patterns compared to DeepRUOT, where the growth rate first increases and then goes down again in the last time point. Is it expected? I believe this figure could benefit from more explanations. In other words, in most of the provided qualitative evidence, I cannot tell why Var-RUOT is better than DeepRUOT.
* **Wall clock time:** Very minor point. I would add the unit of measure of wall clock time somewhere in the table caption.
* **Description of performance:** Sometimes, the authors seem to interpret the scores in absolute terms. For example, in lines 248-250 or in lines 286-288. I think these scores are mostly comparative, so I would phrase it differently and say that the results demonstrate that Var-RUOT better models trajectories than the baselines, rather than the accurate recovery claim.

I wish I could be more positive in my assessment at this stage. I am looking forward to hearing more from the authors in their responses.

[1] Eyring, L., Klein, D., Uscidda, T., Palla, G., Kilbertus, N., Akata, Z. and Theis, F., 2023. Unbalancedness in neural monge maps improves unpaired domain translation. arXiv preprint arXiv:2311.15100.

[2] Neklyudov, K., Brekelmans, R., Tong, A., Atanackovic, L., Liu, Q. and Makhzani, A., 2023. A computational framework for solving wasserstein lagrangian flows. arXiv preprint arXiv:2310.10649.

[3] Moon, K.R., Van Dijk, D., Wang, Z., Gigante, S., Burkhardt, D.B., Chen, W.S., Yim, K., Elzen, A.V.D., Hirn, M.J., Coifman, R.R. and Ivanova, N.B., 2019. Visualizing structure and transitions in high-dimensional biological data. Nature biotechnology, 37(12), pp.1482-1492.

[4] Luecken, M.D., Burkhardt, D.B., Cannoodt, R., Lance, C., Agrawal, A., Aliee, H., Chen, A.T., Deconinck, L., Detweiler, A.M., Granados, A.A. and Huynh, S., 2021, August. A sandbox for prediction and integration of DNA, RNA, and proteins in single cells. In Thirty-fifth conference on neural information processing systems datasets and benchmarks track (Round 2).

---

> ### Author Rebuttal · Authors · 2025-07-31
>
> Thank you for carefully reviewing our paper and for providing valuable comments. Below please find our point-by-point response:
>
> > Summary & Strengths
>
> We appreciate your recognition of our work's strengths. Our paper derives the first-order optimality conditions for the RUOT problem and solves it by fitting a single scalar field, thereby modeling unbalanced distributions and stochastic dynamics. Compared to previous methods, our Var-RUOT **finds paths with lower action**, **achieves stable training**, and **converges faster**. We also discuss the **biological priors** associated with the RUOT penalty function. With each action guiding its corresponding dynamics, we hope our framework offers a more physical approach to modeling single-cell omics data.
>
>
>
> > [W1, Q3] On the Selection of $g^\frac{2}{15}$
>
> Thank you for your question regarding $\psi(g)$. To clarify, we use $g^{2/15}$ **as an example** to show that when $\psi'(g)>0$ and $\psi''(g)<0$, $g(\boldsymbol{x},t)$ increases along $\boldsymbol{u}(\boldsymbol{x},t)$ (Theorem 4.2). We chose $\psi_2(g)=g^{2/15}$ because:
> 1. It is the simplest form  that meets $\frac{d\psi(g)}{d|g|}>0$, $\psi(g)=\psi(-g)$, and $\psi''(g)<0$.
> 2. From the optimality condition we obtain
> $$
> g(\lambda) = \left(\frac{\lambda}{\alpha}\frac{2q+1}{2p}\right)^{\frac{2q+1}{2p-(2q+1)}} \propto \left(\frac{1}{\lambda}\right)^{\frac{2q+1}{(2q+1)-2p}}.
> $$
> For a more stable training process, we keep $g(\lambda)$ from being too steep near $\lambda=0$. So we set $p=1$ and choose a large $q$ (i.e., $q=7$).
>
> Note that $g(\lambda)$ is undefined at $\lambda=0$ due to the necessary discontinuity in $\psi'(g)$ at $g=0$. To address this, we linearly interpolate between $g(-\delta)$ and $g(\delta)$ (Appendix A.2). This singularity causes slowing convergence and degrading distribution reconstruction when using $\psi_2(g)$. A more reasonable way to deal with this problem is to fit the reciprocal of $\lambda$, which is our future work.
>
> Empirically, on the 2D Mouse Blood Hematopoiesis dataset (results are shown below), $\psi(g)=g^{2/15}$ performs similarly to $\psi(g)=g^{2/21}$, while $\psi(g)=g^{2/9}$ (with a steeper $g(\lambda)$) leads to unstable training and poorer reconstruction. Our experiments demonstrate that choosing the exponent on $g$ within a certain range is feasible. We will **update the paper to include the above analysis in the sensitivity analysis section of the Appendix**.
>
> |$\psi(g)$/$\mathcal{W}_1$ Distance|t=1|t=2|
> |-|-|-|
> |$\psi(g)=g^{(2/9)}$|0.9476|1.2197|
> |$\psi(g)=g^{(2/15)}$|0.2953| 0.1917|
> |$\psi(g)=g^{(2/21)}$|0.2858|0.2565|
>
> As noted in our Limitations, choosing $\psi(g)$ **remains open**. Future work may derive it from **physical principles** (e.g., Branching SDE, Stochastic Thermal Dynamics) or **learn it with a neural network**. **We will include these discussions in the Appendix.**
>
> > [Q1] On Weight Initialization
>
> Using particle methods, we approximate the true measure $\mu$ with the empirical measure $\mu^N$. We initialize by placing a particle at every $t=0$ data point with the uniform weight 1. This yields
> $$
> \mu^{N}(0) = \frac{1}{N}\sum_{i=1}^{N} w_i \delta(x - X_{i} ) \rightarrow \mu(0),
> $$
> which justifies the rationale to set $w_i(0)=1$.
>
> > [Q2] On the Ground Truth for Mass Matching
>
> In RUOT modeling, because cell counts vary (due to growth or death), the number of data points differs over time. If we set the total mass at $t_1=0$ as the reference number 1, then the mass at time $t_i$ as $\frac{N_i}{N_1}$ (with $N_i$ as the count at $t_i$), which serves as the ground truth for mass matching loss. **We will clarify this in our paper.**
>
> > [Q4]On the Schrodinger Bridge Problem
>
> We apologize for the confusion. The original Schrodinger Bridge is dynamic—minimizing the KL divergence over path space between a stochastic process and a reference one. In our paper, "static" means some method[(Tong, A.,et al) S2FM] first compute a static OT solution via entropy-regularized OT, then fit the velocity field using Conditional Flow Matching. **We will clarify this in the paper.**
>
>
> > [W4,5,6] Additional Baselines for Existing Datasets
>
> Thanks for the great suggestion. Follow your advice, we **supplemented three baselines on existing datasets**; in addition, we provided results for **nine baselines and Var-RUOT on two new datasets**. We first **supplemented the additional baselines** on existing datasets with:
> - Wasserstein Gradient Flow (WGF in the table) [(Neklyudov, K., et al.) Wasserstein Lagrangian Flows]
> - OTCFM [(Lipman, Y., et al.) Flow Matching]
> - UOTCFM [(Eyring, L., et al.)Unbalancedness in Neural Monge Maps]
>
> Specifically, for the 2D Mouse Blood Hematopoiesis dataset, we have added three previously unevaluated baselines: MIOFLOW, S2FM, and PISDE.
>
> > Simulation Dataset
>
> |Method/$\mathcal{W}_1$ Distance|$t_1$|$t_2$|$t_3$|$t_4$|
> |-|-|-|-|-|
> |OTCFM|0.1035|0.2078|0.2898|0.3107|
> |UOTCFM|0.1002|0.1653|0.1711|0.4129|
> |WGF|0.4983|0.8346|0.8046|0.4493|
> |Var-RUOT|0.0452|0.0385|0.0445|0.0572|
>
> > EMT Dataset
>
> |Method/$\mathcal{W}_1$ Distance|$t_1$|$t_2$|$t_3$|
> |-|-|-|-|
> |OTCFM|0.2557|0.2701|0.2799|
> |UOTCFM|0.2538|0.2696|0.2771|
> |WGF|0.3901|0.4381|0.2848|
> |Var-RUOT|0.2540|0.2670|0.2683|
>
> > 2D Mouse Dataset
>
> |Method/$\mathcal{W}_1$ Distance|$t_1$|$t_2$|
> |-|-|-|
> |MIOFLOW|0.3546|0.2772|
> |S2FM|0.1706|0.1602|
> |PISDE|0.3124|0.2531|
> |OTCFM|0.3674|0.3222|
> |UOTCFM|0.3301|0.2051|
> |WGF|0.3302|0.2051|
> |Var-RUOT|0.1200|0.1917|
>
>
> Next, we show experiments on **additional datasets**. We use a 100-dimensional, 5-time-point ($t=0,1,2,3,4$) EB dataset [(Moon, K.R., et al) Visualizing structure and transitions...] and the NeurIPS challenge dataset [(Luecken, M.D., et al.) A sandbox for prediction and integration...]. For NeurIPS dataset, we extract gene expression data from donor 28483, cluster them via K-Means by pseudo time, and order the clusters (by centroid) into $t=0,1,2,3,4$. This new dataset is then used for further tests. The results are as follows:
>
> > EB Dataset
>
> |Method/$\mathcal{W}_1$ Distance|$t_1$|$t_2$|$t_3$|$t_4$|
> |-|-|-|-|-|
> |S2FM|9.6732|11.0711|12.5436|14.7100|
> |PISDE|8.8317|9.4283|9.7872|11.1102|
> |MIOFLOW|8.7182|9.4139|9.7547|11.0080|
> |TIGON|8.7992|9.6497|10.0130|11.3284|
> |RUOT|8.8029|9.6518|10.0365|11.3555|
> |OTCFM|9.4459|11.3786|14.0529|19.7489|
> |UOTCFM|9.4794|11.6060|14.6715|21.7653|
> |ActionMatching|12.7145|16.7809|13.4663|18.1165|
> |WGF|9.9847|11.0562|11.9719|13.2417|
> |Var-RUOT|8.9158|9.7955|10.2647|11.4927|
>
> > NeurIPS Dataset
>
> |Method/$\mathcal{W}_1$ Distance|$t_1$|$t_2$|$t_3$|$t_4$|
> |-|-|-|-|-|
> |S2FM|6.1876|12.6170|9.1830|12.5831|
> |PISDE|5.6642|5.2358|6.2793|8.9086|
> |MIOFLOW|6.4267|6.2715|7.1079|8.9035|
> |TIGON|5.9481|5.7599|6.5539|8.9573|
> |RUOT|6.0166|5.6972|6.5099|8.8047|
> |OTCFM|5.5082|5.4476|7.0935|10.1416|
> |UOTCFM|5.5526|5.4349|6.7482|9.6782|
> |ActionMatching|7.3701|8.5156|9.2842|9.9665|
> |WGF|7.1785|7.7135|8.2521|8.6629|
> |Var-RUOT|4.7961|4.1816|6.2312|9.1058|
>
> And we compared the path action learned by TIGON, DeepRUOT and VarRUOT:
>
> |Method/Path Action|TIGON|DeepRUOT|VarRUOT|
> |-|-|-|-|
> |EB Dataset|4.4927|5.0571|4.1536|
> |NeurIPS Dataset|10.4250|10.9898|3.6982|
>
> In summary, the experiments show that:
> - For low-dimensional data: On the EMT dataset, Var-RUOT achieves distribution reconstruction **comparable to OTCFM and UOTCFM**, and on the three-gene simulation and 2D Mouse Blood Hematopoiesis datasets, it **outperforms the other baselines**.
> - For high-dimensional data: On the EB dataset, Var-RUOT yields results similar to TIGON and DeepRUOT, and outperforms simulation-free methods (OTCFM, UOTCFM) as well as Wasserstein Gradient Flow; on the NeurIPS dataset, it **outperforms existing methods**. Notably, compared to DeepRUOT and TIGON, Var-RUOT finds a path with **significantly lower action**.
>
> Thus, the experiments confirm that Var-RUOT remains competitive on high-dimensional data, achieving good distribution reconstruction while finding lower-action paths. **We will include these discussions on new datasets & baselines in the Appendix.**
>
>
> > [W7] On the Figures
>
> Thank you for your questions regarding our figures. For Figures 2 and 3:
> - Crosses represent real dataset points, and circles represent data generated by our algorithm.
> - Trajectory colors indicate particle mass, with brighter areas showing higher mass.
>
> We will add legends and annotations for clarity.
>
> Regarding the EMT results, Figure 3 (a) shows DeepRUOT produces curved trajectories (even passing through sparse top regions), while Figure 3 (b) shows nearly straight lines connecting the start and end points. Since straight lines minimize action, Table 2 compares the Path Action of DeepRUOT and Var-RUOT, with Var-RUOT achieving significantly lower action.
>
> The growth pattern in Var-RUOT emerges naturally from using the WFR Metric (Theorem 4.2), where the growth rate increases along the trajectory. Without a "minimum action" objective, many valid velocity fields and growth patterns could reconstruct the marginals.  Both methods correctly reconstruct the marginals, while Var-RUOT finds a lower-action solution, which aligns with our objective.
>
> We clarify that our goal in presenting these figures is **not** to show that Var-RUOT has a significantly lower reconstruction loss, but to demonstrate that both methods perform similarly in marginal reconstruction. Var-RUOT’s advantages lie in it **obtains a lower-action solution**, **simpler scalar field fitting**, **faster convergence**, and the elimination of the pretraining phase in DeepRUOT.
>
> > [W2,3,8,9]Other Writing Issues
>
> Thank you for your valuable suggestions. We will revise the paper to:
> - Define $\alpha$ at line 103 as the penalty coefficient for growth/death.
> - Clarify in lines 248–250 and 286–288 that the $\mathcal{W}_1$, $\mathcal{W}_2$ distances and Path Action are relative indicators, with Var-RUOT performing better relative to the baseline.
> - Add the unit (seconds) for Wall Clock time in Table 3.
> - Add time index to time-dependent variables.

---

> > ### Comment · Reviewer_YDV3 · 2025-08-03
> >
> > Dear authors,
> >
> > Thank you very much for your rebuttal. I am impressed and commend the amount of experimental effort you put into it. Below, I will provide some additional comments on the paper after rebuttal.
> >
> > **Value of the exponent.** I still believe that this theoretical analysis could benefit from a case where changing the exponent of $g$ provides biological advantages compared to the standard WRF approach, cause as of now it sounds a little bit like an underexplored avenue. However, I do appreciate the experiments on the variation of the parameter and the explanation. Thank you for those.
> >
> > **Experiments.** The experiments you provided are comprehensive. The only thing I wonder is whether you used the same pipeline as the compared publications to process and reconstruct the data, since the Wasserstein distance values on Cite and EB differ from the original publications, even just in terms of range.
> >
> > **Figures.** Thank you for your explanation. I now see what you mean. I would recommend making the paths more distinguished from the colours of the points, because now the qualitative analysis is a bit hard to grasp. I am still not sure that I understand the behaviour of Var-RUOT in Fig.2. The growth coefficient increases, decreases again and then increases again along the trajectory. Is this behaviour expected?
> >
> > All in all, I think the paper passes the acceptance bar for the conference, so I decided to increase my 3 to a 4. All the best to the authors for the remainder of the discussion period.

---

> > > ### Author Response · Authors · 2025-08-04
> > >
> > > Thank you again for taking the time to review our paper and read our responses. We are especially glad to hear your positive evaluation of our work. We are also very happy to further discuss some questions here.
> > >
> > > > Regarding the exponent of $g$
> > >
> > >   Thank you for your suggestion. In the revised paper, we will especially highlight the biological implications of various exponent choices as meaningful future work directions, and as a on-going work, we are indeed cooperating with wet-lab biologists now to incorporate lineage-tracing information to provide more biological insights into this problem.
> > >
> > > > Regarding the Experiments
> > >
> > > Thank you for your reminders, and we have taken the chance to carefully check all the details of implementation. In our experiments, we standardized the data so that the mean of each principal component is 0 and the standard deviation is 1. Moreover, in our analysis, the EB Dataset was reduced to 50 dimensions using PCA, which may differ from the dimensional settings in other works (e.g. 5 dimensions). These data pre-processing differences explain why our results vary from those reported in the original literature. We will detail the data processing workflow in the revised version of the paper, especially highlighting the differences. For a more comprehensive evaluation, we will also provide another table using the processing pipelines in the original publications. More importantly, we will open-source all the data processing code and the processed datasets to ensure fully reproducible results of all the experiments.
> > >
> > > > Regarding the Figures
> > >
> > >  Thank you for your nice suggestion. We will adjust the trajectory colors so that they are clearly distinguishable from the points in the figure.
> > >
> > > In addition, the outcome shown in Figure 2 is indeed the implication of Theorem 4.2 and one of the main messages conveyed by our work: if the WFR Metric is employed, the growth rate increases along the direction of the velocity field **for the data points at each time step**, which might bring such biologically "weird" phenonmenon. Because the assumed dynamical system is non-autonomous (with $t$ explicitly present in $\lambda$), this theorem does not impose constraints on the growth rate across data points from different time steps. As a result, the figure shows the growth rate first rising and then falling in order to fit the overall distribution change.  Overall, this indeed motivates us to discuss the choices of $\psi$, and implies that WFR metric needs to be carefully used in terms of biological interpretation. We will include more detailed discussion in the revised paper about this result.
> > >
> > > Once again, thank you for all your insightful and constructive comments. Your feedback has greatly helped us improve the paper, and we will incorporate your suggestions into the final version.
> > >
> > > Sincerely,
> > >
> > > The Authors

---

> ### Comment · Reviewer_YDV3 · 2025-08-04
>
> Dear authors,
>
> Thank you again for your final remarks. Everything sounds reasonable. I confirm my final score of 4.
>
> All the best!

---

### Official Review · Reviewer_5WHv · 2025-06-30

**Clarity:** 2
**Significance:** 3
**Originality:** 2
**Rating:** 4
**Confidence:** 3

**Summary:**

The authors present a new method for learning high dimensional stochastic dynamics for unnormalized distributions. The authors build on the recent RUOT framework modifying the loss to ensure the learned solution satisfies the optimality condition of least action. This in turns allows them to use a single network and parametrize only a scalar field. Additionally they propose a growth penalty function which better aligns with biological priors.

**Questions:**

- It would be great to see a table (or something like this) clearly showing the differentiate factors and or advantages of their method compared to the many other (see above).
- Recent works show that estimating action losses can be quite difficult and they proposed quadrature of these types of losses [3]. Do the authors encounter similar issues?
- How and why are the authors able to eliminate the pertaining steps of the original RUOT algorithm [1] ?
- How exactly was the g^2/15 value obtained? Was this a tuned hyper-parameter?

[1] Learning stochastic dynamics from snapshots through regularized unbalanced optimal transport. Zhang. 2024.
[3] Parametric model reduction of mean-field and stochastic systems via higher order action matching. Berman. 2024.

**Ethical Concerns:**

["NO or VERY MINOR ethics concerns only"]

**Final Justification:**

The response was very thorough and addressed my concerns.

**Limitations:**

- The Limitations section is pushed to the appendix, which in my opinion is not in the spirit of the guidelines and neurips checklist.

**Paper Formatting Concerns:**

- Large emojis in the first figure make the paper feel unserious. I recommend removing these.

**Quality:**

3

**Strengths And Weaknesses:**

**Strengths**
- The motivation of trying to satisfy the condition of  least action is well motivated.
- There is a substantial theory section.
- The numerical results are quite strong. The method really does seem to be an improvement. The faster convergence speed is promising.

**Weaknesses**
- The overall presentation is not great with regards to what the key methodological innovations are and how they fit within the existing literature. It is quite difficult to tease out the exact differences between this paper other work such as the original RUOT paper [1] or action matching style papers [2]. This is quite important because the latter works also explicitly enforce optimality conditions WRG to least action and the loss terms look very similar to what the paper calls "Action Loss".
- In particular one of the key approaches of the proposed method is to paramzeraize with a single scalar field where by the velocity is the gradient of this field and the growth function is equal to it. The title of paper might suggest that this is a new innovation, but it is in fact exactly what is done in the original action matching paper [2]. While the authors cite this paper and state "these methods typically employ separate neural networks to parameterize the velocity and growth functions, without leveraging their optimality conditions or the inherent relationship between them". To my knowledge this is incorrect with respect to [2].


[1] Learning stochastic dynamics from snapshots through regularized unbalanced optimal transport. Zhang. 2024.
[2] Action Matching: Learning Stochastic Dynamics from Samples. Neklyudov. 2023.

---

> ### Author Rebuttal · Authors · 2025-07-31
>
> Thank you for carefully reviewing our paper and raising valuable questions. Below we address your concerns in a point-to-point response:
>
> > Summary & Strengths
>
> We appreciate your recognition of our work's strengths. Our paper derives the first-order optimality conditions for the RUOT problem and solves it by fitting a single scalar field, thereby modeling unbalanced distributions and stochastic dynamics. Compared to previous methods, our Var-RUOT **finds paths with lower action**, **achieves stable trainin**, and **converges faster**. We also discuss the **biological priors** associated with the RUOT penalty function. With each action guiding its corresponding dynamics, we hope our framework offers a more physical approach to modeling single-cell omics data.
>
>
>
> > [W1, W2, Q1] Differences from Previous Methods
>
> Thank you for your question and sorry for the confusion. First, we sincerely apologize for not thoroughly checking our citation positions and misunderstandings caused by it. We fully agree that Action Matching is a very innovative work uses a single network to parameterize $s_{\theta}(\boldsymbol{x}, t)$ that inspires our research; we will review the manuscript to correct any improper citations and clarify the significance of previous methods.
>
> Following your great suggestion, we also made a table to clearly highlight the contribution of Var-RUOT that will **later be included in the revised Appendix**:
>
>
> |Method|Unbanlanced Distribution| Stochastic Dynamics | Unbalanced + Stochastic | Least Action|
> |---|---|---|---|---|
> |SF2M|-|+|-|-|
> |PISDE|-|+|-|+|
> |MIOFlow|-|-|-|-|
> |Action Matching|+|+|-|+|
> |TIGON|+|-|-|-|
> |DeepRUOT|+|+|+|-|
> |Var-RUOT|+|+|+|+|
>
>
> - **Unbalanced Distribution** indicates that the method can handle non-normalized distributions.
> - **Stochastic Dynamics** means the method supports stochastic dynamics.
> - **Unbalanced + Stochastic** shows the method can handle both simultaneously (e.g., DeepRUOT).
> - **Least Action** signifies that the method employs a first-order condition to minimize the action, rather than simply adding the action as a weighted loss.
>
> The key difference between Var-RUOT and DeepRUOT is that Var-RUOT **leverages first-order optimality conditions** in its parameterization and loss design (e.g., using only a scalar field rather than three neural networks as in DeepRUOT). Compared to DeepRUOT, our approach **finds paths with lower action** and **addresses unstable training and slow convergence**, thus eliminating the need for a pre-training phase.
>
> Var-RUOT and Action Matching differ in three key ways:
>
> Firstly, Var-RUOT can **simultaneously handle imbalanced data distributions and stochastic dynamics**, both crucial for single-cell omics.
>
>
> Secondly, Action Matching's training algorithm needs to sample from intermediate distribution $q_{t}(x)$, while in single-cell omics data, we only have snapshot data at a few time points as boundary sample. So acquiring intermediate samples from $q_{t}(x)$ may be challenging due to missing evolution data in-between, potentially hindering the ability to capture complex dynamics. Empirically, our experimental results (Tables 1, 2, 4) show that Var-RUOT achieves more satisfactory results over existing results including Action Matching.
>
> Lastly, Action Matching defines the Action Gap loss as:
>
> $$L_{\text{AM}} = E_{q_{0}(x)} [s_{0}(x)] - E_{q_{1}(x)} [s_{1}(x)] + \int_{0}^{1} E_{q_{t}(x)} \left[ \dfrac{1}{2} \| \nabla s_{t}(x) \|^{2} + \dfrac{\partial s}{\partial t}(x) \right] \mathrm{d} t$$
>
> While Var-RUOT directly adopts the HJB Loss(Line 204-209) to ensure that the computed solution $\lambda(\boldsymbol{x},t)$ satisfies the HJB equation. In other words, Var-RUOT incorporates the first-order condition into the loss, enabling it to find a solution with lower action. **We will add these discussions in the revised manuscript**.
>
>
>
>
>
> > [Q2] On the Estimation of the Action Loss
>
> Thank you for your question. In our experiments, we employed two methods to compute the integral losses for both the action and HJB terms. When using the Euler-Maruyama method, setting the step size too large (e.g., $\Delta t = 0.2$) results in high variance estimates for the action and HJB losses, leading to unstable training. Therefore, we use $\Delta t = 0.1$ in all experiments. The Stochastic Runge-Kutta method offers better convergence, allowing an integration step of $\Delta t = 0.2$. Since our approach is simulation-based, a longer step size significantly reduces training time. In practice, we thus choose the Stochastic Runge-Kutta method and select a larger step size as long as training remains stable. We will discuss this in our final version and  cite [(Berman.) High Order Action Matching] in our paper.
>
> > [Q3] Why the Related Steps in the Original RUOT Algorithm Can Be Omitted
>
> In the original DeepRUOT algorithm, three neural networks are employed to approximate $\boldsymbol{v}(\boldsymbol{x},t)$, $g(\boldsymbol{x},t)$, and $\log \rho(\boldsymbol{x},t)$, with all constraints imposed as soft constraints via loss functions. For instance, to ensure that $\boldsymbol{v}$, $g$, and $\rho$ satisfy the Fokker–Planck equation, DeepRUOT introduces the Fokker–Planck loss
>
> $$L_{\text{FP}} = \|\partial_{t} p_{\theta} + \nabla_{x} \cdot (p_{\theta}\boldsymbol{v})- g_{\theta} p_{\theta}\| + \lambda_{w} \| p_{\theta}(\boldsymbol{x},0) - p_{0} \|$$
>
> Training these three interdependent networks simultaneously is challenging and often causes convergence issues. To accelerate training by obtaining a good initialization, DeepRUOT incorporates a pretraining phase: (1) training $\boldsymbol{v}$ and $g$ using a reconstruction loss ${L}_{\text{Recons}}$, and (2) training $p$ with Conditional Flow Matching.
>
> In Var-RUOT, we use a single neural network $\lambda_\theta(\boldsymbol{x},t)$ to approximate a scalar field. The velocity field $\boldsymbol{u}(\boldsymbol{x},t)$ and growth rate $g(\boldsymbol{x},t)$ are derived analytically from $\lambda(\boldsymbol{x},t)$ using first-order optimality conditions (Theorem 4.1). As a result, Var-RUOT avoids handling multiple interdependent networks and eliminates the need for the pretraining phase used in DeepRUOT. This is indeed one of the main contribution of our work; ablation experiments in DeepRUOT demonstrate that these pretraining steps are indispensable (DeepRUOT[(Zhang.), DeepRUOT], Appendix B.5), and we have removed them in Var-RUOT. **We will add these discussions in the revised manuscript**.
>
>
> > [Q4] Why we Choose $g^{2/15}$
>
> Thanks for your question regarding $\psi(g)$. we choose $g^{2/15}$ primarily to **serve as an example** showing that when $\psi'(g) > 0$ and $\psi''(g) < 0$, at each time step $g(\boldsymbol{x},t)$ increases along the direction of the velocity field $\boldsymbol{u}(\boldsymbol{x},t)$ (Theorem 4.2). As for why we adopt $\psi_{2}(g) = g^{2/15}$ as the example, it is because: (1) It is the simplest choice that satisfies the conditions given in our paper (lines 224–229): $\dfrac{\mathrm{d}\psi(g)}{\mathrm{d}|g|} > 0$, $\psi(g) = \psi(-g)$, and $\psi''(g) < 0$. (2) For training considerations: when choosing $\psi(g)=g^{(2p)/(2q+1)}$, the first-order optimality condition allows us to derive
> $$
> g(\lambda) = \left(\dfrac{\lambda}{\alpha} \dfrac{2q+1}{2p}\right)^{\frac{2q+1}{2p - (2q+1)}} \propto \left(\dfrac{1}{\lambda}\right)^{\frac{2q+1}{(2q+1)-2p}}.
> $$
> We aim for more stable training, and since $g$ is computed directly from $\lambda$, we do not want the $g(\lambda)$ to be overly steep (especially near $\lambda=0$), which requires the exponent $\dfrac{2q+1}{(2q+1)-2p}$ to be as small as possible. This means choosing $p$ as small as possible (hence $p=1$) and choosing $q$ as large as possible (we select $q=7$). Therefore, we ultimately choose the penalty function $\psi(g)=g^{2/15}$.
>
> *Note: It is worth noting that $g(\lambda)$ is undefined at $\lambda=0$. In fact, for any function $\psi(g)$ that meets the conditions outlined in lines 224–229 along with $\psi''(g)<0$, its first derivative must be discontinuous at $g=0$, so this singularity is inevitable. To handle this issue, we select a small value $\delta$ and linearly interpolate between $g(-\delta)$ and $g(\delta)$ by redefining
> $$
> g(\lambda) = \dfrac{1}{2\delta}\left(g(\delta) - g(-\delta)\right)(\lambda + \delta) + g(-\delta), \quad \lambda \in [-\epsilon, \epsilon),
> $$
> to bypass the singularity (Appendix A.2). Another approach to deal with the singularity is to fit the reciprocal of $\lambda$, denoted as $\mu$, thereby moving the singularity at $\lambda=0$ to $\mu \rightarrow \infty$. In practice, $\mu \rightarrow \infty$ does not occur, just as the singularity where $|g| \rightarrow \infty$ is not reached. This method avoids artificially modifying the $g(\lambda)$ relationship and is more reasonable. This will be our future work.*
>
> To further explain our training considerations, we experimented with two additional parameter sets on the 2D Mouse Blood Hematopoiesis dataset. The experimental results are as follows:
>
>
> |$\psi(g)$  / $\mathcal{W}_1$ Distance|t=1|t=2|
> |---|---| ---- |
> |$\psi(g)=g^{(2/9)}(p=1, q=4)$|0.9476|1.2197|
> |$\psi(g)=g^{(2/15)}(p=1, q=7)$|0.2953|0.1917|
> |$\psi(g)=g^{(2/21)}(p=1, q=10)$|0.2858|0.2565|
>
>
>
> The experimental results indicate that choosing $\psi(g)=g^{2/15}$ produces performance comparable to $\psi(g)=g^{2/21}$, while selecting $\psi(g)=g^{2/9}$ leads to unstable training and poorer reconstruction of the distribution. Our experiments demonstrate that choosing the exponent on $g$ within a certain range is feasible. **We will update the paper to include the above analysis in the sensitivity analysis section of the Appendix.**
>
> >  On Writing Issues
>
> Thank you for your suggestions regarding our writing. We will review and revise our paper to:
> - Move the Limitations section into the main text.
> - Remove the emoji from Figure 1.

---

> > ### Comment · Reviewer_5WHv · 2025-08-02
> >
> > Thank you these responses were helpful. I feel my concerns have been addressed. Conditional on the authors making these changes, I think this work is a good edition to the conference.

---

> > > ### Author Response · Authors · 2025-08-03
> > >
> > > Thank you again for taking the time to review our paper and to read our responses. Your insightful and constructive feedback has helped us further improve our work. We will incorporate your suggestions into the revised version of the paper.
> > >
> > > Sincerely,
> > >
> > > The Authors

---

### Official Review · Reviewer_s4d8 · 2025-06-30

**Clarity:** 3
**Significance:** 3
**Originality:** 3
**Rating:** 5
**Confidence:** 4

**Summary:**

This paper introduces Var-RUOT, a new method for learning the dynamics of evolving systems using just a few snapshot observations. Traditional approaches rely on multiple neural networks and often struggle to find the most efficient, biologically realistic paths. Var-RUOT simplifies the problem by learning only a single scalar function, from which all relevant dynamics can be derived. This leads to more stable training, faster convergence, and solutions that follow the principle of least action (i.e., the most efficient evolution). The method also allows for customizing the behavior of growth or decay in the system to better reflect biological processes. Through experiments on synthetic and real biological datasets, the authors show that Var-RUOT achieves more accurate and efficient results than existing approaches.

**Questions:**

1. Your method relies on enforcing first-order necessary conditions for action minimization via the HJB loss. However, these conditions are not sufficient for global or even local optimality. Can you provide any theoretical or empirical evidence that minimizing the HJB residual leads to low-action trajectories beyond satisfying a PDE constraint pointwise?

2. The derivation assumes that the mapping \( \lambda = \alpha \psi'(g) \) can be inverted to recover \( g(x,t) \). What guarantees do you have that \( \lambda_\theta(x,t) \) stays within the image of \( \psi' \) during training? How is the domain of \( g \) enforced or regularized when \( \psi \) is non-convex or has flat regions?

3. In lower-dimensional or isotropic settings, optimal transport paths with minimal action (e.g., in the WFR or Benamou-Brenier case) are analytically or numerically solvable. Why did you not include comparisons against such ground-truth geodesics to rigorously evaluate whether Var-RUOT truly finds lower-action trajectories?

4. What is the quantitative effect of violating the HJB equation on the evolution of \( \rho(x,t) \)? Can small residual errors compound over time and distort the learned transport path, particularly in extrapolation tasks?

5. The biological interpretation of \( \psi''(g) \) controlling the direction of growth is intuitive but lacks formal backing. Why is \( \psi(g) = g^{2/15} \) considered a principled choice beyond its curvature sign? Have you validated this with any empirical or theoretical biological models?

6. You claim that learning a single scalar field simplifies the problem. However, could this choice overly constrain the model class, especially in settings where velocity and growth cannot be jointly derived from a single potential? Are there situations where this coupling leads to underfitting?

**Ethical Concerns:**

["NO or VERY MINOR ethics concerns only"]

**Final Justification:**

I believe this is a valuable paper and authors addressed my concerns thoroughly, therefore I increased my score to 5.

**Limitations:**

Yes

**Quality:**

3

**Strengths And Weaknesses:**

### Strengths

1. By reducing the RUOT problem to learning a single scalar field \lambda(x,t), the method avoids redundant parameterization of velocity and growth fields, leading to a compact, interpretable, and theoretically grounded framework.

2. Var-RUOT achieves faster convergence and more stable training compared to prior methods, as shown through detailed empirical comparisons across multiple datasets and metrics (e.g., action, Wasserstein distance, wall time).

3. The paper connects the choice of the growth penalty function \psi(g) to biological priors, enabling control over whether growth increases or decreases along inferred trajectories, offering practical relevance in single-cell dynamics applications.

###  Weaknesses

1. The paper enforces only first-order optimality conditions through the HJB residual loss, which are necessary but not sufficient for achieving true minimal action. There is no guarantee that the learned solution corresponds to a global or even local minimum of the action functional.

2. While the method relies on a weighted particle system to approximate density evolution, it lacks any quantitative analysis of convergence rates or variance bounds for the empirical measure, especially under high-dimensional stochasticity.

3. The method assumes that the mapping  \lambda = \alpha \psi'(g) is invertible across all relevant values of \lambda, but this is not always true for non-convex or saturating forms of \psi. No mechanism is introduced to ensure this invertibility is respected during training.

---

> ### Author Rebuttal · Authors · 2025-07-31
>
> Thank you for carefully reviewing our paper and for providing valuable comments. Below please find our point-by-point response:
>
> > Summary & Strengths
>
> We appreciate your recognition of our work's strengths. Our paper derives the first-order optimality conditions for the RUOT problem and solves it by fitting a single scalar field, thereby modeling unbalanced distributions and stochastic dynamics. Compared to previous methods, our Var-RUOT **finds paths with lower action**, **achieves stable training**, and **converges faster**. We also discuss the **biological priors** associated with the RUOT penalty function.
>
>
> > [W1] Regarding Var-RUOT’s Guarantee for the Minimal Action Solution
>
> Thank you for your comment. Since Var-RUOT is a deep learning–based RUOT solver, it guarantees only a local minimum of the action. Empirically, experiments (Table 1,2,4) along with all additional results in the rebuttal show that Var-RUOT achieves a lower action than DeepRUOT, and its Path Action is comparable to that of traditional OT solvers.
>
> We consider guaranteeing a global minimum an interesting direction for future work and **will discuss it in the revised manuscript**. We plan to start with analytical solutions for simple cases (e.g. Dirac-to-Dirac transport) and then extend to real data by treating complex distributions as a superposition of Dirac deltas and solving the associated velocity field via a Flow Matching–like approach.
>
>
> > [W2] Convergence Rate
>
> Thanks for your insightful question. From the perspective of numerical integration, we use the particle method to estimate $L_{\text{HJB}}$ and $L_{\text{Action}}$, which is equivalent to performing integration via Monte Carlo methods. Let the number of particles be $N$. Then, for any smooth function $f(\boldsymbol{x})$ (In the Action Loss and HJB Loss, the integrand.), the central limit theorem tells us that the error of the integral $\int f(\boldsymbol{x})\,\mathrm{d}\mu^N$ is of order $\mathcal{O}\left(\dfrac{1}{\sqrt{N}}\right)$. **We will revise the paper and include this discussion on the convergence rate.** In practical applications of the Var-RUOT algorithm, our default setting is $N=512$, which ensures that the estimates of both loss have very small variance.
>
>
> > [Q1, Q4] Regarding the effect of the HJB Loss.
>
> Thanks for the insightful question. Var-RUOT controls the penalty for $\lambda(\boldsymbol{x},t)$ violating the HJB equation by adjusting the weight $\gamma_{\text{HJB}}$: a higher weight increases the importance of the HJB loss, reducing the violation after training. Sensitivity analysis (Section C.2, Figure 7) shows that as **$\gamma_{\text{HJB}}$ increases, the Path Action decreases**, proving that a lower HJB loss leads to a smaller action solution. To address the concerns, we further provide sensitivity results for the EMT dataset and additional results for the 2D Mouse dataset:
>
>
> - EMT Dataset
>
> |$\gamma_{\text{HJB}}$|$0$| $6.25 \times 10^{-3}$|$3.125 \times 10^{-2}$| $6.25 \times 10^{-2}$|$6.25 \times 10^{-1}$|$3.125$|
> |-|-|-|-|-|-|-|
> |$\mathcal{L}_{\text{HJB}}$ |1.9016|1.3091|0.6205|0.3685|0.0223| 0.0006|
> |Path Action|0.4612|0.4584|0.3855|0.3544|0.2975|0.2808|
>
>
> - 2D Mouse Dataset
>
> |$\gamma_{\text{HJB}}$ |$0$| $6.25 \times 10^{-3}$ | $3.125 \times 10^{-2}$ | $6.25 \times 10^{-2}$ | $6.25 \times 10^{-1}$ | $3.125$ |
> |-|-|-|-|-|-|-|
> |$\mathcal{L}_{\text{HJB}}$ |1.9107|1.4605|1.0745|0.6819|0.0725|0.0074|
> |Path Action|3.3002|3.2165|3.2397|3.1491|3.1182|3.1200|
>
>
>
> These results confirm that **incorporating the HJB loss effectively finds lower action solutions**. Based on our findings, we recommend setting $\gamma_{\text{HJB}}$ between $10^{-2}$ and $10^{-1}$ to balance minimal action with data reconstruction quality. We will include these discussion in our final version.
>
>
> > [W3,Q2] Reversibility Issue of $\psi'(g)$
>
>
> Thank you for your valuable comment and apologize for the insufficient discussions in previous paper. Here we discuss the two scenarios of $\psi$ selection in more details:
>
> -  $\psi_{1}(g)=g^{2p}$, $\psi_{1}'(g)=2p\cdot g^{2p-1}$, $p\in\mathbb{Z}^{+}$. Here, $\psi_{1}'(g)$ is a bijection from $\mathbb{R}$ to $\mathbb{R}$, so $g$ is uniquely determined from $\lambda=\alpha\psi_{1}'(g)$.
>
> - $\psi_{2}(g)=g^{\frac{2p}{2q+1}}$, $\psi_{2}'(g)=\frac{2p}{2q+1}g^{\frac{2p-(2q+1)}{2q+1}}$, $p,q\in\mathbb{Z}^{+}$ with $2q+1>2p$. Here, $\psi_{2}(g)$ is a bijection from $\mathbb{R}\setminus\{0\}$ to $\mathbb{R}\setminus\{0\}$ (undefined at 0). Directly solving$\lambda=\alpha\psi_{2}'(g)$ yields
> $$
> g=\left(\frac{\alpha}{\lambda}\frac{2q+1}{2p}\right)^{\frac{2q+1}{(2q+1)-2p}}.
> $$
> At $\lambda=0$, $g(\lambda)$ is undefined. To handle this inevitable singularity, we set a small $\delta$ and linearly interpolate between $g(-\delta)$ and $g(\delta)$ for $\lambda\in[-\epsilon,\epsilon)$,i.e., $g(\lambda) = \dfrac{1}{2\delta}\left(g(\delta) - g(-\delta)\right)(\lambda + \delta) + g(-\delta)$ (Appendix A.2)
>
>
> Another approach is to fit the reciprocal of $\lambda$, denoted as $\mu$, thereby moving the singularity at $\lambda=0$ to $\mu \rightarrow \infty$. In practice, $\mu \rightarrow \infty$ does not occur, just as the singularity where $|g| \rightarrow \infty$ is not reached. This method avoids artificially modifying the $g(\lambda)$ relationship and is more reasonable. **This will be our future work. We will update the paper for more thorough disucssions on $\psi'(g)$ choice and include these detailed analysis in the appendix.**
>
> > [Q3] Comparison with Traditional OT Solvers
>
> Thank you for your valuable comment. In additional experiment, here we built two 2D normalized and unbalanced datasets to test Var-RUOT’s ability to find minimal action paths. For normalized data, we used the `pot` library to obtain ground truth; for unbalanced data, we applied a Sinkhorn-based static WFR solver [(Wang Z., et al.) Wasserstein-fisher-rao document distance]. With $\sigma=0$, Var-RUOT achieves similar—or even lower—Path Action values than traditional solvers(results are shown below), demonstrating its effectiveness. **We will include these analysis in the appendix.**
>
>
> |Dataset|Normalized|Normalized|Unbalanced|Unbalanced|
> |---|---|---|---|---|
> |-|Simple Gaussian|Gaussian Mixture|Simple Gaussian|Gaussian Mixture|
> |Path Action (Traditional Solver)|0.5046 |1.9062 |1.3776| 2.4464|
> |Path Action (Var-RUOT) |0.4998|1.8754 |1.0102| 2.3909|
>
>
> > [Q4] Extrapolation Performance
>
> We evaluated Var-RUOT’s extrapolation capability and error propagation by extending the three-gene simulation dataset to nine time points. The first five were used for training, while the remaining four tested extrapolation. We compared TIGON, DeepRUOT, and Var-RUOT. The results are shown below:
>
> |Method/$\mathcal{W}_{1}$  Distance|t=5|t=6|t=7|t=8|
> |---|---|---|---|---|
> |TIGON|0.1932|0.3437|0.5209|0.6913|
> |DeepRUOT|0.1027|0.1940|0.3359|0.5047|
> |VarRUOT|0.1057|0.1806|0.2890|0.4260|
>
> Var-RUOT outperforms TIGON and performs similarly to DeepRUOT in long-term extrapolation. However, as time increases, error grows because the model’s non-autonomous system learns $\lambda(\boldsymbol{x}, t)$ only within the training window. Beyond this range, the $t$-dependency is less accurate, leading to larger errors. Nonetheless, this non-autonomous system improves flexibility and helps prevent underfitting (see Q6). **We will include these analysis in the appendix.**
>
>
> >[Q5] Choice of $\psi(g) = g^{2/15}$
>
> Sorry for the confusion. We clarify that choosing $\psi(g)=g^{2/15}$ primarily to **showcase an example** with $\psi''(g)<0$. There are two reasons.
>
> Firstly, the simplest function family satisfying our conditions (lines 224–229)—that is, $\frac{d\psi(g)}{d|g|}>0$, $\psi(g)=\psi(-g)$, and $\psi''(g)<0$—is given by
> $$
> \psi(g)=g^{\frac{2p}{2q+1}},\quad p,q\in\mathbb{Z}^+,\; 2p<2q+1.
> $$
>
> Secondly, for training stability, the first-order condition yields
> $$
> g(\lambda)=\left(\frac{\lambda}{\alpha}\,\frac{2q+1}{2p}\right)^{\frac{2q+1}{2p-(2q+1)}} \propto \left(\frac{1}{\lambda}\right)^{\frac{2q+1}{(2q+1)-2p}}.
> $$
> To avoid a steep $g(\lambda)$ curve near $\lambda=0$, we choose the smallest $p$ (i.e., $p=1$) and a large $q$ (here, $q=7$), which leads to $\psi(g)=g^{2/15}$.
>
> Here we conduct additional experiments on the 2D Mouse Blood Hematopoiesis dataset (results are shown below) to show that while $\psi(g)=g^{2/15}$ and $\psi(g)=g^{2/21}$ perform similarly, however $\psi(g)=g^{2/9}$ yields unstable training and poorer reconstruction.
>
>
> | $\psi(g)$  / $\mathcal{W}_1$ Distance| t=1 | t=2 |
> | --- | ----| ---- |
> | $\psi(g)=g^{(2/9)}(p=1, q=4)$   | 0.9476 | 1.2197|
> | $\psi(g)=g^{(2/15)}(p=1, q=7)$  | 0.2953 | 0.1917  |
> | $\psi(g)=g^{(2/21)}(p=1, q=10)$ | 0.2858 |0.2565 |
>
>
> Overall, our experiments confirm that, within a reasonable range, different choices for the exponent on $g$ are feasible. We will include this sensitivity analysis in the appendix.
>
> > [Q6] On Whether a Single Scalar Field Causes Underfitting
>
> Thanks for the insightful question. Here we clarify that:
> - Experiments show that Var-RUOT reconstructs the marginal distributions at all time points well—both on low-dimensional data (Sections 6.1 and 6.3) and high-dimensional data (Appendix C.4), the results are comparable to DeepRUOT (which uses three networks) without underfitting. Please also refer to the responses to Reviewer kRW9 [Q1] and Reviewer YDV3 [W4,5,6], which include quantitative results of Var-RUOT on high-dimensional datasets.
> - Allowing $\lambda$ to depend on both $\boldsymbol{x}$ and $t$ enables capturing both trajectory evolution and total mass fluctuations; if $\lambda$ depended solely on $\boldsymbol{x}$, it could not fit distributions showing some specific mass changing pattern (e.g., mass that first rises then falls).
> - Moreover, the DeepRUOT method employs three networks, while they are interdependent (via a Fokker–Planck loss), which makes training unstable and requires pretraining, while Var-RUOT is simpler and more stable.

---

> ### Comment · Reviewer_s4d8 · 2025-08-06
>
> I thank the authors for their comprehensive rebuttal. All of my questions have been answered satisfactorily, and I am satisfied with the responses. Therefore, I am increasing my score to 5.

---

> > ### Author Response · Authors · 2025-08-07
> >
> > We sincerely appreciate the time and effort you dedicated to reviewing our paper and considering our responses. Your insightful and constructive feedback has been invaluable, and we are committed to incorporating your suggestions into the revised version of the manuscript.
> >
> > Sincerely,
> > The Authors

---

### Official Review · Reviewer_kRW9 · 2025-07-02

**Clarity:** 3
**Significance:** 3
**Originality:** 2
**Rating:** 5
**Confidence:** 4

**Summary:**

This paper proposes Var-RUOT, a new framework for solving the Regularized Unbalanced Optimal Transport (RUOT) problem (Def. 4.1). The authors reformulate the RUOT problem as a weighted particle system (Theorem 5.1). By applying the Hamilton–Jacobi–Bellman (HJB) equation (Theorem 4.1), they derive an action-matching loss based on the value function. In addition, they introduce an HJB loss term to enforce necessary conditions and a reconstruction loss to ensure that the distribution generated by the model remains consistent with the real data distribution. The method is evaluated on synthetic datasets (Sec. 6.1) and applied to biological problems (Sec. 6.2, 6.3).

**Questions:**

- Does the method scale to high-dimensional settings (e.g., more than 100 dimensions) where the marginals are Gaussian Mixture Models with many modes? It seems challenging to generalize this approach to such complex, high-dimensional cases.

- What hyperparameters require careful tuning? I imagine that the ratio between the loss function might be crucial. Could the authors elaborate on this aspect or provide additional ablation studies? Also, since estimating $\lambda$ relies on accurately minimizing the HJB loss, could the authors include a training curve for the HJB loss? Furthermore, have the authors considered using an L1 loss instead, as it might yield sharper estimates of the value surface? Any comments on this would be appreciated.

**Ethical Concerns:**

["NO or VERY MINOR ethics concerns only"]

**Final Justification:**

I initially thought this paper should be clearly accepted, and the authors addressed all of my concerns.

**Limitations:**

Please refer to Strength and Weakness section.

**Paper Formatting Concerns:**

.

**Quality:**

3

**Strengths And Weaknesses:**

- The paper is well-organized and mathematically sound, with clear examples and intuitive explanations.

- To the best of my knowledge, this is the first work to apply a weighted particle method in this context. This formulation provides valuable intuition and could inspire extensions to related problems. I believe this is an innovative contribution that merits attention from the NeurIPS community.

- The method demonstrates an efficient application of RUOT compared to previous works. The experimental results consistently outperform prior approaches. Moreover, the combination of toy experiments for validation and real-world biological applications highlights both the technical soundness and practical potential of the approach.


- However, there are some limitations. Estimating the value function solely through a PINN-like loss (via the HJB equation) may lead to inaccuracies, as solving PDEs is challenging and in this case requires high precision—since the control is derived from differentiating the value function. Even small errors in estimating the value surface could cause significant deviations. Additionally, while the action-matching loss is novel, it may have limited scalability, and the need to estimate second-order derivatives with respect to the state space adds further computational burden. Despite these limitations, I still believe the paper’s contributions are valuable.

---

> ### Author Rebuttal · Authors · 2025-07-31
>
> Thank you for carefully reviewing our paper and for providing valuable comments. Below please find our point-by-point response:
>
> > Summary & Strengths
>
> We appreciate your recognition of our work's strengths. Our paper derives the first-order optimality conditions for the RUOT problem and solves it by fitting a single scalar field, thereby modeling unbalanced distributions and stochastic dynamics. Compared to previous methods, our Var-RUOT **finds paths with lower action**, **achieves stable training**, and **converges faster**. We also discuss the **biological priors** associated with the RUOT penalty function. With each action guiding its corresponding dynamics, we hope our framework offers a more physical approach to modeling single-cell omics data.
>
> >[W1] Relying solely on a PINN-like HJB loss to estimate the value function can produce significant control deviations, as even small errors cause large inaccuracies due to sensitive differentiation.
>
> Thank you for your insightful question. We fully agree that since deep learning solver for the RUOT problem converts hard constraints into soft ones via the loss function, so there's no guarantee that the solution exactly satisfies the marginal constraints nor minimizes the action at every time point. Instead, the HJB loss serves as a **regularizer** that promotes a lower action value without necessarily reducing to 0. On the other hand, we point out that empirically, sensitivity analysis (lines 656–670) shows that increasing the loss weight weight $\gamma_{\text{HJB}}$ would decrease the action value, indicating the effectiveness of such term to achieve the decrease of action for the comnputed path. **We will add the relevant discussions in the revised manuscript**.
>
>
> >[W2] Compute Second-order derivatives causes computational burden.
>
> Thank you for your comments and suggestion. We fully agree that compute second-order derivatives using PyTorch autograd **by default** will requires retaining a larger graph and thus more memory and time than first-order derivatives. Meanwhile we clarify that
>
> - Unlike previous RUOT methods such as DeepRUOT that train three networks for $\boldsymbol{v}$, $\rho$, and $g$, VAR-RUOT fits a single scalar field $\lambda$, reducing the search space, improving stability, and converging faster (see Section 6.2).
>
> - Since Var-RUOT only need second-order derivatives for the $\mathcal{L}_{\text{HJB}}$ loss (to compute $\nabla_x^2 \lambda$), meaning we only require the trace of the Hessian matrix $H$. We can estimate it using the **Hutchinson method**:
>
>   $$
>   \text{trace}(H)=\mathbb{E}_{\boldsymbol{v}}[\boldsymbol{v}^{T} H \boldsymbol{v}], \quad \boldsymbol{v}\sim N(0,I)
>   $$
>
>   which computes the derivative along one direction, reducing cost (error $O(1/\sqrt{m})$, with $m$ samples). We can choose to use Hutchinson early in training for speed and later switch to full autograd for accuracy. **This feature will be made as an option in our open-source code to speedup the computation**.
>
> >[Q1] Can Var-RUOT be Generalized to high-dimensional settings?
>
> Thank you for your question on Var-RUOT's scalability to high-dimensional datasets. In previous manuscript, we validated its scalability on several datasets: 50 and 100 dimensional Gaussian Mixture datasets (lines 709–715), a 50-dimensional Mouse Blood Hematopoiesis dataset, and a 30-dimensional β-cell Differentiation dataset (lines 716–723). For the high-dimensional Gaussian Mixture dataset, Figure 13 qualitatively shows the learned particle trajectories and growth rates on gaussian datasets; Figures 14 and 15 display the learned velocity fields and growth rates for the real 50D and 30D datasets, demonstrating Var-RUOT's applicability.
>
> To quantitatively evaluate scalability, we further designed simulated data at three time points (t = 0, 1, 2) where each point is generated from a mixture of three Gaussians in **150 dimensions**. We then compared various baselines on this dataset, with results as follows:
>
>
> |Method/$\mathcal{W}_1$ Distance|t=1|t=2|PathAction|
> |---|---|---|---|
> |SF2M|7.286|9.457|-|
> |PISDE|**6.051**|7.546|-|
> |MIOFlow|6.494|7.812|-|
> |TIGON|6.157|7.615|13.4913|
> |DeepRUOT|6.202|7.648|13.5992|
> |VarRUOT|6.072|**7.544**|10.1033|
>
>
> Var-RUOT achieves comparable **reconstruction accuracy** to PISDE, slightly outperforms TIGON and DeepRUOT, and significantly outperforms other methods on the 150-dimensional Gaussian Mixture dataset. It also yields a **significantly lower path action** than TIGON/DeepRUOT, confirming its scalability in high-dimensional settings. **We will include these experiments in our revised manuscript.**
>
>
> >[Q2]Questions on hyperparameter tuning and ablation studies.
>
> Thank you for your question on hyperparameter tuning. Our method requires tuning three hyperparameters: the WFR metric parameter $\alpha$ (controlling the penalty for mass growth/decay) and two loss parameters, $\gamma_{\text{HJB}}$ and $\gamma_{\text{Action}}$. In Appendix C.2, we show that our method is robust to variations in $\alpha$ (using the 2D Mouse Blood Hematopoiesis dataset, lines 648–655, Table 7). Similarly, sensitivity experiments on $\gamma_{\text{HJB}}$ and $\gamma_{\text{Action}}$ (lines 656–670, Figures 7–8) demonstrate that increasing either parameter reduces the Path Action. However, very high values diminish the impact of the reconstruction loss that enforces marginal constraints, which degrades performance.
>
> To provide further insights, here we have conducted additional sensitivity tests on $\gamma_{\text{HJB}}$ and $\gamma_{\text{Action}}$ using the Mouse Blood Hematopoiesis dataset.
>
>
> - **Case with fixed $\gamma_{\text{Action}} = 6.25\times {10}^{-2}$:**
>
>
> |$\gamma_{\text{HJB}}$|$0$|$6.25\times10^{-3}$|$3.125\times10^{-2}$|$6.25\times10^{-2}$|$6.25\times10^{-1}$|$3.125$|
> |:-:|:-:|:-:|:-:|:-:|:-:|:-:|
> |$\mathcal{W}_{1}$|0.1573|0.1609|0.1550|0.1316|0.2109|0.2539|
> |PathAction|3.3002|3.2165|3.2397|3.1491|3.1182|3.1200|
>
>
>
> - **Case with fixed $\gamma_{\text{HJB}}= 6.25\times {10}^{-2}$:**
>
>
> |$\gamma_{\text{Action}}$|$0$|$6.25\times10^{-3}$|$3.125\times10^{-2}$|$6.25\times10^{-2}$|$6.25\times10^{-1}$|$3.125$|
> |:-:|:-:|:-:|:-:|:-:|:-:|:-:|
> |$\mathcal{W}_{1}$|0.1454|0.1344|0.1674|0.1316|0.4001|0.7914|
> |PathAction|3.4043|3.4497|3.2737|3.1491|2.4716|1.2059|
>
>
>
> The results from these additional sensitivity experiments are consistent with those presented in the original manuscript: higher values of $\gamma_{\text{HJB}}$ and $\gamma_{\text{Action}}$ lead to a lower Path Action, yet if they are set too high, the reconstruction quality deteriorates significantly. In fact, all experiments in the main text use the same parameters, $\gamma_{\text{HJB}} = \gamma_{\text{Action}} = 6.25 \times 10^{-2}$ (see Table 6). In practice, we recommend setting both parameters in the range of $10^{-2}$ to $10^{-1}$ and **we will make it clear in the open-source package**.
>
> >[Q3]Training curve for the HJB loss.
>
> Thank you for your question. Due to rebuttal restrictions, we cannot provide an image of the HJB loss curve. The HJB loss initially rises for 10–20 epochs (since $L_{\text{Recon}}$  dominates while parameters are near zero) and then gradually declines as $L_{\text{Recon}}$ reaches a similar scale as $L_{\text{HJB}}$ and $L_{\text{Action}}$. **We will include this curve in the appendix of revised manuscript.**
>
>
> >[Q4]Can we use L1 loss instead?
>
> Thank you for your valuable suggestion. In additional experiment, we have modified the HJB loss per your recommendation by replacing the quadratic penalty with a linear penalty. Then we re-ran experiments on three datasets presented in the main text for the Var-RUOT method, comparing the performance when using the original $L_{2}$ loss versus the $L_{1}$ loss.
>
> - Three Gene Simulation Dataset:
>
> |Method/$\mathcal{W}_1$ Distance|t=1|t=2|t=3|t=4|PathAct|
> |---|---|---|---|---|---|
> |Var-RUOT($L_2$Loss)|0.0452|0.0385|0.0445|0.0572|1.1105|
> |Var-RUOT($L_1$Loss)|0.0489|0.0556|0.0589|0.0664|1.1249|
>
>
>
> - EMT Dataset:
>
>
> |Method/$\mathcal{W}_1$ Distance|t=1|t=2|t=3|Path Action|
> |---|---|---|---|---|
> |Var-RUOT($L_{2}$Loss)|0.2540|0.2670|0.2683|0.3544|
> |Var-RUOT($L_{1}$Loss)|0.2631|0.2796|0.2809|0.3528
>
>
>
> - 2D Mouse Blood Hematopoiesis Dataset:
>
>
> |Method/$\mathcal{W}_1$ Distance|t=1|t=2|Path Action|
> |---|---|---|---|
> |Var-RUOT($L_{2}$Loss)|0.1200|0.1431|3.1491|
> |Var-RUOT($L_{1}$Loss)|0.1747|0.1690|3.1889|
>
>
>
> The experimental results indicate that with our chosen hyperparameter values ($\gamma_{\text{HJB}} = 6.25 \times 10^{-2}$ and $\gamma_{\text{Action}} = 6.25 \times 10^{-2}$), using either the $L_{1}$ loss or the $L_{2}$ loss for the HJB loss yields similar reconstruction performance and Path Action. **We will add the above analysis to the appendix during the revision of the paper.**

---

> > ### Comment · Reviewer_kRW9 · 2025-08-03
> >
> > Thank you for your response. My concerns are fully addressed.

---

> > > ### Author Response · Authors · 2025-08-04
> > >
> > > Thank you once again for dedicating your time to review our paper and to read our responses. Your insightful and constructive comments have significantly enhanced our work, and we will certainly integrate your suggestions into the revised version of the paper.
> > >
> > > Sincerely,
> > >
> > > The Authors

---

### Note · Authors · 2025-08-14

We thank all reviewers and the AC for their careful evaluation and constructive suggestions. Below is a summary of the points discussed and the improvements to be made in the revised manuscript:

- **HJB Loss (kRW9, s4d8):** We clarified that the HJB loss acts as a regularization term and demonstrated its effectiveness through additional sensitivity analysis.
- **The choice of $\psi(g)$ (s4d8, 5WHv, YDV3):** We explained why we chose $\psi(g)=g^{\frac{2}{15}}$, as it is the simplest form and aids training. We described our method for handling the noninvertibility of $\psi'(g)$, and supplemented our work with experiments to show that a range of exponents for $g$ is feasible.
- **Finding the path of least action (kRW9, s4d8):** We clarified that both the HJB Loss and the Action Loss direct the model to find the path of minimal action, and we compared our approach with traditional OT solvers to demonstrate its effectiveness.
- **Differences from previous methods (5WHv):** We provided a table outlining the differences between Var-RUOT and previous methods, including: (1) explicit use of the RUOT problem’s optimality conditions; (2) simultaneous consideration of unbalanced distributions and stochastic dynamics; and (3) faster training and convergence.
- **Technical details (all reviewers):** We clarified various technical points, including the computational complexity of second-order derivatives, the use of L1 Loss, the particle method's convergence rate, potential underfitting, the numerical integration for Action Loss estimation, and the initialization of particle weights and mass ground truth.
- **Scalability experiments (kRW9, YDV3):** We added high-dimensional dataset experiments—using 150-dimensional Gaussian mixture data, the EB Dataset, and the NeurIPS Dataset—to validate Var-RUOT’s scalability. We also introduced additional baselines (OTCFM, UOTCFM, and WGF) and demonstrated Var-RUOT’s long-term extrapolation capability.
- **Writing and figures (5WHv, YDV3):** We clarified issues regarding Figures 2 and 3 and revised the manuscript’s wording in accordance with the reviewers' suggestions.

During the rebuttal phase, we were delighted that all reviewers acknowledged their concerns were addressed. Our method has received interest from the AI for Science community, and we believe it will inject new vitality into fields like single-cell data modeling and neural optimal control.

---

### Decision · Program_Chairs · 2025-09-17

**Decision:**

Accept (poster)

**Comment:**

This paper presents Var-RUOT, a new approach to recover the dynamics from snapshot data, aproblem central in the computational biology literature. The core innovation is to directly incorporate the first-order optimality conditions (HJB) of the RUOT problem into the models parameterization and loss. This leads to significant simplifications over the previous methods.

The work received unanimous and positive recommendation from all four reviewers who were particularly impressed by the method's theoretically grounding, the performance, and a strong rebuttal that addressed all concerns from all reviewers. For these reasons I recommend accepting this work.